# Using altimetry observations combined with GRACE to select parameter sets of a hydrological model in data scarce regions

Petra Hulsman[1], Hessel C. Winsemius[1], Claire I. Michailovsky[2], Hubert H.G. Savenije[1], Markus Hrachowitz[1]

[1]Water Resources Section, Faculty of Civil Engineering and Geosciences, Delft University of Technology, Stevinweg 1, 2628 CN Delft, The Netherlands
[2]IHE Delft Institute for Water Education, Westvest 7, 2611 AX Delft, The Netherlands

*Correspondence to*: Petra Hulsman (p.hulsman@tudelft.nl)

**Abstract.** Limited availability of ground measurements in the vast majority of river basins world-wide increases the value of alternative data sources such as satellite observations in hydrological modelling. This study investigates the potential of using remotely sensed river water level, i.e. altimetry observations, from multiple satellite missions to identify parameter sets for a hydrological model in the semi-arid Luangwa River Basin in Zambia. A distributed process-based rainfall runoff model with sub-grid process heterogeneity was developed and run on a daily timescale for the time period 2002 to 2016. As a benchmark, feasible model parameter sets were identified using traditional model calibration with observed river discharge data. For the parameter identification using remote sensing, data from the Gravity Recovery and Climate Experiment (GRACE) were used in a first step to restrict the feasible parameter sets based on the seasonal fluctuations in total water storage. Next, three alternative ways of further restricting feasible model parameter sets using satellite altimetry time-series from 18 different locations along the river were compared. In the calibrated benchmark case, daily river flows were reproduced relatively well with an optimum Nash-Sutcliffe efficiency of $E_{NS,Q} = 0.78$ (5/95[th] percentiles of all feasible solutions $E_{NS,Q,5/95} = 0.61 - 0.75$). When using only GRACE observations to restrict the parameter space, assuming no discharge observations are available, an optimum of $E_{NS,Q} = -1.4$ ($E_{NS,Q,5/95} = -2.3 - 0.38$) with respect to discharge was obtained. The direct use of altimetry based river levels frequently led to over-estimated flows and poorly identified feasible parameter sets ($E_{NS,Q,5/95} = -2.9 - 0.10$). Similarly, converting modelled discharge into water levels using rating curves in the form of power relationships with two additional free calibration parameters per virtual station resulted in an over-estimation of the discharge and poorly identified feasible parameter sets ($E_{NS,Q,5/95} = -2.6 - 0.25$). However, accounting for river geometry proved to be highly effective. This included using river cross-section and gradient information extracted from global high-resolution terrain data available on Google Earth, and applying the Strickler-Manning equation to convert modelled discharge into water levels. Many parameter sets identified with this method reproduced the hydrograph and multiple other signatures of discharge reasonably well with an optimum of $E_{NS,Q} = 0.60$ ($E_{NS,Q,5/95} = -0.31 - 0.50$). It was further shown that more accurate river cross-section data improved the water level simulations, modelled rating curve and discharge simulations during intermediate and low flows at the basin outlet where detailed on-site cross-section information was available. Also, increasing the number of virtual stations used for parameter selection in the calibration period considerably improved the model performance in a spatial split sample validation. The results provide robust evidence that in the absence of directly observed discharge data for larger rivers in data scarce regions, altimetry data from multiple virtual stations combined with GRACE observations have the potential to fill this gap when combined with readily

available estimates of river geometry, thereby allowing a step towards more reliable hydrological modelling in poorly gauged or ungauged basins.

## 1 Introduction

Reliable models of water movement and distribution in terrestrial systems require sufficient good quality hydro-meteorological data throughout the modelling process. However, the development of robust models is challenged by the limited availability of ground measurements in the vast majority of river basins world-wide (Hrachowitz et al., 2013). Therefore, modellers increasingly resort to alternative data sources such as satellite data (Lakshmi, 2004; Winsemius et al., 2008; Sun et al., 2018; Pechlivanidis and Arheimer, 2015; Demirel et al., 2018; Zink et al., 2018; Rakovec et al., 2016; Nijzink et al., 2018; Dembélé et al., 2020).

In the absence of directly observed river discharge data, various types of remotely sensed variables provide valuable information for the calibration and evaluation of hydrological models. These include, for instance, remotely sensed time series of river width (Sun et al., 2012; Sun et al., 2015), flood extent (Montanari et al., 2009; Revilla-Romero et al., 2015), or river and lake water levels from altimetry (Getirana et al., 2009; Getirana, 2010; Sun et al., 2012; Garambois et al., 2017; Pereira-Cardenal et al., 2011; Velpuri et al., 2012).

Satellite altimetry observations provide estimates of the water level relative to a reference ellipsoid. For these observations, a radar signal is emitted from the satellite in the nadir direction and reflected back by the earth surface. The time difference between sending and receiving this signal is then used to estimate the distance between the satellite and the earth surface. As the position of the satellite is known at very high accuracy, this distance can then be used to infer the surface level relative to a reference ellipsoid (Łyszkowicz and Bernatowicz, 2017; Calmant et al., 2009). Satellite altimetry is sensed and recorded along the satellite's track. Altimetry based water levels can therefore only be observed where these tracks intersect with open-water surfaces; for rivers, these points are typically referred to as "virtual stations" (de Oliveira Campos et al., 2001; Birkett, 1998; Schneider et al., 2017; Jiang et al., 2017; Seyler et al., 2013). Depending on the satellite mission, the equatorial inter-track distance can vary between 75 km and 315 km, the along-track distance between 173 m and 374 m, and the temporal resolution between 10 days and 35 days (Schwatke et al., 2015; CNES, Accessed 2018; ESA, 2018; Łyszkowicz and Bernatowicz, 2017). Due to this rather coarse resolution, the application of remotely sensed altimetry data is at this moment limited to large lakes or rivers of more than approximately 200 m wide (Getirana et al., 2009; de Oliveira Campos et al., 2001; Biancamaria et al., 2017). Use of altimetry for hydrological models so far also remains rather rare due to the relatively low temporal resolution of the data, with applications typically limited to monthly or longer modelling time steps (Birkett, 1998).

In some previous studies, altimetry data were used to estimate river discharge at virtual stations in combination with routing models (Michailovsky and Bauer-Gottwein, 2014; Michailovsky et al., 2013) or stochastic models (Tourian et al., 2017). Other studies either directly related river altimetry to modelled discharge (Getirana et al., 2009; Getirana and Peters-Lidard, 2013; Leon et al., 2006; Paris et al., 2016) or they relied on rating curves developed with water level data from either in-situ measurements (Michailovsky et al., 2012; Tarpanelli et al., 2013; Papa et al., 2012; Tarpanelli et al., 2017) or, alternatively, from altimetry data (Kouraev et al., 2004). In typical applications, radar altimetry data from one single or only a few virtual stations were used for model calibration, validation or data assimilation. These data were mostly obtained from a single satellite mission, either TOPEX/Poseidson or Envisat (Sun et al., 2012; Getirana, 2010; Liu et al., 2015; Pedinotti et al., 2012;

Fleischmann et al., 2018; Michailovsky et al., 2013; Bauer-Gottwein et al., 2015). In previous studies, hydrological models have been calibrated or validated successfully with respect to (satellite based) river water levels for example by 1) applying the Spearman Rank Correlation coefficient (Seibert and Vis, 2016; Jian et al., 2017), or by converting modelled discharge to stream levels using 2) rating curves whose parameters are free calibration parameters in the modelling process (Sun et al., 2012; Sikorska and Renard, 2017) or 3) the Strickler-Manning equation to directly estimate water levels over the hydraulic properties of the river (Liu et al., 2015; Hulsman et al., 2018).

In the Zambezi river basin, altimetry data has been used in previous studies for hydrological modelling (Michailovsky and Bauer-Gottwein, 2014; Michailovsky et al., 2012). These studies used the altimetry data from the Envisat satellite in an assimilation procedure to update states in a Muskingum routing scheme. Including the altimetry data improved the model performance, especially when the model initially performed poorly due to high model complexity or input data uncertainties.

Despite these recent advances in using river altimetry in hydrological studies, exploitation of its potential is still limited. Various previous studies have argued and provided evidence based on observed discharge data that, in a special case of multi-criteria calibration, the simultaneous model calibration to flow in multiple sub-basins of a river basin, can be beneficial for a more robust selection of parameter sets and thus for a more reliable representation of hydrological processes and their spatial patterns (e.g. Ajami et al., 2004; Clark et al., 2016; Hrachowitz and Clark, 2017; Hasan and Pradhanang, 2017; Santhi et al., 2008). Hence, there may be considerable value in simultaneously using altimetry data not only from one single satellite mission but in combining data from multiple missions, which has not yet been systematically explored. While promising calibration results using data from Envisat were found by Getirana (2010) in tropical and Liu et al. (2015) in snow-dominated regions, altimetry data from multiple sources has not yet been used to calibrate hydrological models in semi-arid regions.

As altimetry observations only describe water level dynamics, it does not provide direct information on the discharge amount. In an attempt to reduce the uncertainty in modelled discharge arising from the missing information on flow amounts, data from the Gravity Recovery and Climate Experiment (GRACE), which provides estimates of the total monthly water storage anomalies, were used to support model calibration. With GRACE, discharge can be constrained through improved simulation of the rainfall partitioning into runoff and evaporation as illustrated in previous studies (Rakovec et al., 2016; Bai et al., 2018).

Therefore, the overarching objective of this study is to explore the combined information content (cf. Beven, 2008) of river altimetry data from multiple satellite missions and GRACE observations to identify feasible parameter sets for the calibration of hydrological models of large river systems in a semi-arid, data scarce region. More specifically, in a step-wise approach we use GRACE observations together with altimetry data from multiple virtual stations to identify model parameters following three different strategies and we compare model performances to a traditional calibration approach based on in-situ observed river discharge. These three strategies compare altimetry observations to 1) modelled discharge by applying the Spearman Rank Correlation coefficient, and to modelled stream levels by converting modelled discharge using 2) rating curves whose parameters were treated as free model calibration parameters and 3) the Strickler-Manning equation to infer water levels directly from hydraulic properties of the river. These three strategies are tested on a distributed process-based rainfall-runoff model with sub-grid process heterogeneity for the Luangwa Basin. More

specifically, we test the following research hypotheses: 1) the use of altimetry data combined with GRACE observations allows a meaningful selection of feasible model parameter sets to reproduce river discharge depending on the applied parameter identification strategy, and 2) the combined application of multiple virtual stations from multiple satellite missions improves the model's ability to reproduce observed hydrological dynamics.

## 2 Site description

The study area is the Luangwa River in Zambia, a tributary of the Zambezi River (Figure 1). It has a basin area of 159,000 km$^2$ which is about 10% of the Zambezi River Basin. The Luangwa Basin is poorly gauged, mostly unregulated and sparsely populated with about 1.8 million inhabitants in 2005 (The World Bank, 2010). The mean annual precipitation is around 970 mm yr$^{-1}$, potential evaporation is around 1555 mm yr$^{-1}$ and river runoff reaches about 100 mm yr$^{-1}$ (The World Bank, 2010). The main land cover consists of broadleaf deciduous forest (55%), shrub land (25%) and savanna grassland (16%) (GlobCover, 2009). The irrigated area in the basin is limited to about 180 km$^2$, i.e. roughly 0.1% of the basin area with an annual water use of about 0.7 mm yr$^{-1}$ which amounts to < 0.001% of the annual basin water balance (The World Bank, 2010). The landscape varies between low lying flat areas along the river to large escarpments mostly in the North West of the basin and highlands with an elevation difference up to 1850 m (see Figure 1B and Section 3.2 for more information on the landscape classification). During the dry season, the river meanders between sandy banks while during the wet season from November to May it can cover flood plains several kilometres wide.

The Luangwa drains into the Zambezi downstream of the Kariba Dam and upstream of the Cahora Bassa Dam. The operation of both dams is crucial for hydropower production, and flood and drought protection, but is very difficult due to the lack of information from poorly gauged tributaries such as the Luangwa (SADC, 2008; Schleiss and Matos, 2016; The World Bank, 2010). As a result, the local population has suffered from severe floods and droughts (ZAMCOM et al., 2015; Beilfuss and dos Santos, 2001; Hanlon, 2001; SADC, 2008; Schumann et al., 2016).

### 2.1 Data availability

### 2.1.1 In-situ discharge and water level observations

In the Luangwa basin, historical in-situ daily discharge and water level observations were available from the Zambian Water Resources Management Authority at the Great East Road Bridge gauging station, located at 30$^o$ 13' E and 14$^o$ 58' S (Figure 1) about 75 km upstream of the confluence with the Zambezi. In this study, all complete hydrological years of discharge data within the time period 2002 to 2016 were used; these are the years 2004, 2006 and 2008.

### 2.1.2 Gridded data products

Besides the above in-situ observations, gridded data products were used in this study for topographic description, model forcing (precipitation and temperature), and model parameter selection/calibration (total water storage anomalies), as shown in Table 1. The temperature data was used to estimate the potential evaporation according to the Hargreaves method (Hargreaves and Samani, 1985; Hargreaves and Allen, 2003).

Gravity Recovery and Climate Experiment (GRACE) observations describe the monthly total water storage anomalies which includes all terrestrial water stores present in the groundwater, soil moisture and surface water. Two identical satellites observe the variations in the Earth's gravity field to detect regional mass changes which are dominated by variations in the terrestrial water storage once atmospheric effects have been accounted for (Landerer and Swenson, 2012; Swenson, 2012). In this study, processed GRACE observations of Release 05 generated by CSR (Centre for Space Research), GFZ (GeoForschungsZentrum Potsdam) and JPL (Jet Propulsion Laboratory) were downloaded from the GRACE Tellus website (https://grace.jpl.nasa.gov/); the average of all three sources were used. The raw data were previously processed by CSR, GFZ and JPL to remove atmospheric mass changes using ECMWF (European Centre for Medium-Range Weather Forecasts) atmospheric pressure fields, systematic errors causing north-south-oriented stripes and high frequency noise using a 300 km wide Gaussian filter via spatial smoothening (Swenson and Wahr, 2006; Landerer and Swenson, 2012; Wahr et al., 1998). Processed GRACE observations describe terrestrial water storage anomalies in "equivalent water thickness" in [cm] relative to the 2004 – 2009 time-mean baseline. In other words, the water storage anomaly is the water storage minus the long-term mean (Landerer and Swenson, 2012).

All gridded information was rescaled to the model resolution of $0.1^{\circ}$. The temperature and GRACE data were rescaled by dividing each cell of the satellite product into multiple cells such that the model resolution is obtained, retaining the original value. The precipitation was rescaled by taking the average of all cells located within each model cell.

**Table 1: Gridded data products used in this study**

|  | Time period | Time resolution | Spatial resolution | Product name | Source |
|---|---|---|---|---|---|
| **Digital elevation map** | NA | NA | $0.02^{\circ}$ | GMTED | (Danielson and Gesch, 2011) |
| **Precipitation** | 2002 – 2016 | Daily | $0.05^{\circ}$ | CHIRPS | (Funk et al., 2014) |
| **Temperature** | 2002 – 2016 | Monthly | $0.5^{\circ}$ | CRU | (University of East Anglia Climatic Research Unit et al., 2017) |
| **Total water storage** | 2002 – 2016 | Monthly | $1^{\circ}$ | GRACE | (Swenson, 2012; Swenson and Wahr, 2006; Landerer and Swenson, 2012) |

**2.1.3 Altimetry data**

The altimetry data used in this study was obtained from the following sources: the Database for Hydrological Time Series of Inland Waters (DAHITI; https://dahiti.dgfi.tum.de/en/) (Schwatke et al., 2015), HydroSat (http://hydrosat.gis.uni-stuttgart.de/php/index.php) (Tourian et al., 2013), Laboratoire d'Etudes en Géophysique et Océanographie Spatiales (LEGOS; http://www.legos.obs-mip.fr/soa/hydrologie/hydroweb/; see supplements for more information), and the Earth and Planetary Remote Sensing Lab (EAPRS; http://www.cse.dmu.ac.uk/EAPRS/). In total, altimetry data was obtained for 18 virtual stations in the Luangwa basin (Figure 1A) for the time period 2002 – 2016 from the satellite missions Jason 1 – 3, Envisat and Saral (Table 2, Figure S2).

**Table 2: Overview of the altimetry data in the Luangwa River Basin used in this study**

| Nr. | Longitude | Latitude | Time period | Nr. of days with data | Source | Mission | Space Agency | Temporal resolution | Equatorial inter-track distance | Along-track distance | Literature |
|---|---|---|---|---|---|---|---|---|---|---|---|
| 1 | 30.2823° | -14.8664° | 2008-2016 | 246 | DAHITI | Jason 2, 3 | NASA/CNES | 10 days | 315 km | 294 m | (Schwatke et al., 2015; CNES, Accessed 2018) |
| 2 | 30.0864° | -14.366° | 2008-2015 | 92 | DAHITI | Jason 2, 3 | | | | | |
| 3 | 32.1715° | -12.4123° | 2008-2016 | 248 | DAHITI | Jason 2, 3 | | | | | |
| 4 | 31.1868° | -13.5927° | 2002-2016 | 104 | DAHITI | Envisat, Saral | ESA (Envisat), ISRO/CNES (Saral) | 35 days | 80 km (Envisat), 75 km (Saral) | 374 m (Envisat), 173 m (Saral) | (Schwatke et al., 2015; ESA, 2018; CNES, Accessed 2018) |
| 5 | 31.6984° | -13.2039° | 2002-2016 | 82 | DAHITI | Envisat, Saral | | | | | |
| 6 | 32.2998° | -12.2007° | 2002-2016 | 100 | DAHITI | Envisat, Saral | | | | | |
| 7 | 32.2805° | -12.1157° | 2002-2016 | 103 | DAHITI | Envisat, Saral | | | | | |
| 8 | 32.831° | -11.3674° | 2002-2016 | 105 | DAHITI | Envisat, Saral | | | | | |
| 9 | 30.2704° | -14.8809° | 2008-2015 | 247 | HydroSat | Jason 2 | NASA/CNES | 10 days | 315 km | 294 m | (Tourian et al., 2016; Tourian et al., 2013) |
| 10 | 31.78405° | -13.0995° | 2002-2010 | 65 | EAPRS | Envisat | ESA | 35 days | 80 km | 374 m | (Michailovsky et al., 2012; ESA, 2018) |
| 11 | 31.71099° | -13.1943° | 2002-2010 | 93 | EAPRS | Envisat | | | | | |
| 12 | 30.2740° | -14.8763° | 2008-2015 | 231 | LEGOS | Jason 3 | NASA/CNES | 10 days | 315 km | 294 m | (Frappart et al., 2015; CNES, Accessed 2018) |
| 13 | 32.15843° | -12.412° | 2016-2016 | 28 | LEGOS | Jason 3 | | | | | |
| 14 | 32.15989° | -12.4127° | 2002-2009 | 137 | LEGOS | Jason 1 | | | | | |
| 15 | 30.2740° | -14.8763° | 2008-2016 | 271 | LEGOS | Jason 2 | | | | | |
| 16 | 32.16056° | -12.4125° | 2008-2016 | 283 | LEGOS | Jason 2 | | | | | |
| 17 | 31.80001° | -13.0909° | 2013-2016 | 35 | LEGOS | Saral | ISRO/CNES | 35 days | 75 km | 173 m | |
| 18 | 30.61577° | -14.1852° | 2013-2016 | 24 | LEGOS | Saral | | | | | |

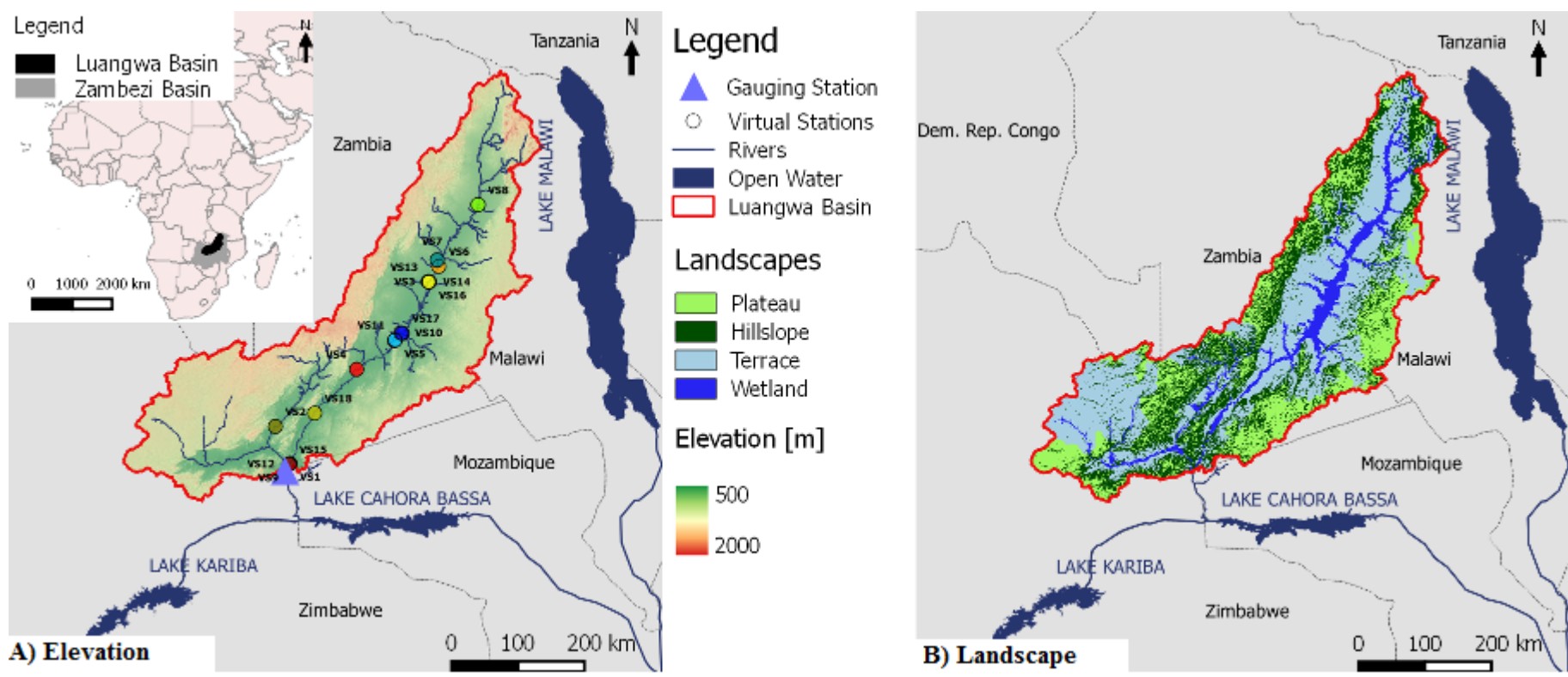

**Figure 1: A) Elevation map of the Luangwa River Basin in Zambia including the Great East Road Bridges river gauging station and the locations of the 18 virtual stations (VS 1 – VS 18, the red dot is VS 4) with altimetry data used in this study; their colours correspond to those in Figure 4. B) Map of the Luangwa River Basin with the main landscape types (see Section 3.2).**

### 2.1.4 River geometry information

In the Luangwa Basin, very limited detailed in-situ information was available on the river geometry such as cross-section and slope. For that reason, this information was extracted from global high-resolution terrain data available on Google Earth as done successfully in previous studies for other purposes (Pandya et al., 2017; Zhou and Wang, 2015). This was done for each virtual station and the basin outlet. Google Earth only provides river geometry information above the river water level. As the Luangwa is a perennial river, parts of the cross-section remain submerged throughout the year and are thus unknown. To limit uncertainties arising from this issue, the cross-section geometry for each virtual station was extracted from the Google Earth image with the lowest water levels at each individual virtual station. The dates of these images in general fall in the dry season, with flows at the Great East Road Bridges gauging station on the respective days ranging from 1% to 4% relative to the maximum discharge (see Supplementary Table S3 for the dates of the satellite images and the associated flows at the Great East Road Bridges gauging station). The database underlying the global terrain images in Google Earth originate from multiple, merged data sources with varying spatial resolutions. For the Luangwa Basin these include the Shuttle Radar Topography Mission (SRTM) with a spatial resolution of 30 m, Landsat 8 with a spatial resolution of 15 m and the Satellite Pour l'Observation de la Terre 4/5 (SPOT) with a spatial resolution of 2.5 m to 20 m (Smith and Sandwell, 2003; Irons et al., 2012; Drusch et al., 2012).

In addition to Google Earth data, the submerged part of the channel cross-section was surveyed in the field on April 27$^{th}$ 2018 near the Great East Road Bridges river gauging station at the coordinates 30$^o$ 13' E and 15$^o$ 00' S (Abas, 2018) with an Acoustic Doppler Current Profiler (ADCP).

### 3 Hydrological model development

### 3.1 General approach

The potential of river altimetry for model calibration was tested with a process-based hydrological model for the Luangwa river basin. This model relied on distributed forcing allowing for spatially explicit distributed water storage calculations. The model was run on a daily time scale for the time period 2002 to 2016. To reach the objective of this study, the following distinct parameter identification strategies were compared in a stepwise approach: (1) traditional model calibration to observed river flow as benchmark; (2) identification of parameter sets reproducing the seasonal water storage anomalies based on GRACE data only; (3a) Altimetry Strategy 1: identification of parameter sets directly based on remotely sensed water levels combined with GRACE data; (3b) Altimetry Strategy 2: identification of parameter sets based on remotely sensed water levels by converting modelled discharges into water levels using calibrated rating curves combined with GRACE data; (3c) Altimetry Strategy 3: identification of parameter sets based on remotely sensed water levels by converting modelled discharges into water levels using the Strickler-Manning equation and including river geometry information (cross-section and gradient) extracted from Google Earth combined with GRACE data; (4a) Water level Strategy 1: identification of parameter sets based on daily river water level at the catchment outlet only using the Strickler-Manning equation and including river geometry information extracted from Google Earth combined with GRACE data; and (4b) Water level Strategy 2: identification of parameter sets based on daily river water level at the catchment outlet only using the Strickler-Manning equation and including river geometry

information obtained from a detailed field survey with an Acoustic Doppler Current Profiler (ADCP) combined
with GRACE data. Note that (1) is completely independent of (2) to (4) where no discharge data was used for the
identification of parameter sets.

### 3.2 Hydrological model structure

In this study, a process-based rainfall-runoff with distributed water accounting and sub-grid process
heterogeneity was developed (Ajami et al., 2004; Euser et al., 2015). The river basin was discretized into a grid
with a spatial resolution of 10 x 10 km$^2$. Each model grid cell was characterized by the same model structure and
parameter sets but forced by spatially distributed, gridded input data (Table 1). Runoff was then calculated in
parallel for each cell separately. Subsequently, a routing scheme was applied to estimate the aggregated flow in
each grid cell at each time step.

Adopting the FLEX-Topo modelling concept (Savenije, 2010) and extending it to a gridded implementation,
each grid cell was further discretised into functionally distinct hydrological response units (HRU) as
demonstrated by Nijzink et al. (2016). Each point within a grid cell was assigned to a response class based on its
position in the landscape as defined by its local slope and "Height-above-the-nearest-drainage" (HAND; Rennó
et al., 2008; Gharari et al., 2011). Similar to previous studies (e.g. Gao et al., 2016; Nijzink et al., 2016), the
response units plateau, hillslope, terrace and wetland were distinguished. Reflecting earlier work (e.g. Gharari et
al., 2011), all locations with slope of > 4% were assumed to be hillslope. Locations with lower slopes were then
either defined as wetland (HAND < 11m), terrace (11m ≤ HAND < 275m) or plateau (HAND ≥ 275m), see
Figure 2. Following this classification wetlands make up $p_{HRU}$ = 8%, terraces $p_{HRU}$ = 41%, hillslopes $p_{HRU}$ = 28%
and plateaus $p_{HRU}$ = 23% of the total Luangwa River Basin area as mapped in Figure 1B.

Each response class consisted of a series of storage components that are linked by fluxes. The flow generated
from each grid cell at any given time step is then computed as the area-weighted flow from the individual
response units plus a contribution from the common groundwater component which connects the response units
(Figure 2). Finally, the outflow from each modelling cell was routed to downstream cells to obtain the
accumulated flow in each grid cell at any given time step. For this purpose, the mean flow length of each model
gird cell to the outlet was derived based on the flow direction extracted from the digital elevation model. The
flow velocity, which was assumed to be constant in space and time, was calibrated. With this information on the
flow path length and velocity, the accumulated flow in each grid cell was calculated at the end of each time step.
The relevant model equations are given in Table 3. This concept was previously successfully applied in a wide
range of environments (Gao et al., 2014; Gharari et al., 2014; Fovet et al., 2015; Nijzink et al., 2016; Prenner et
al., 2018).

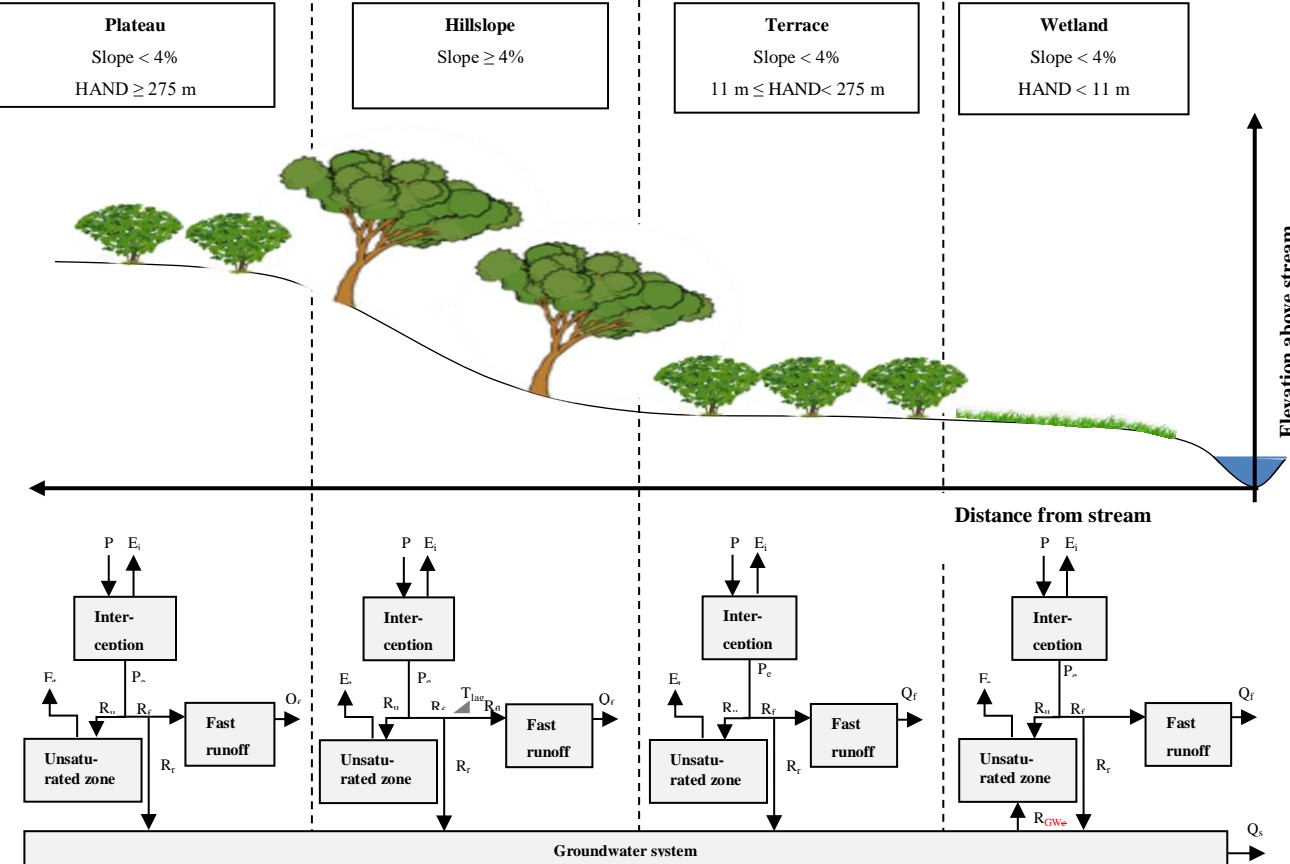

**Figure 2: Sketch of the hydrological response units including the thresholds used in this analysis for the slope and HAND (Height Above Nearest Drainage) and including their corresponding model structures. This spatial sub-grid discretization was applied to each grid cell. Symbol explanation: precipitation ($P$), effective precipitation ($P_e$), interception evaporation ($E_i$), plant transpiration ($E_a$), infiltration into the unsaturated root zone ($R_u$), drainage to fast runoff component ($R_f$), delayed fast runoff ($R_{fl}$), lag time ($T_{lag}$), groundwater recharge ($R_r$), upwelling groundwater flux ($R_{GW}$), fast runoff ($Q_f$), groundwater/slow runoff ($Q_s$).**

**Table 3: Equations applied in the hydrological model. Fluxes [mm d$^{-1}$]: precipitation ($P$), effective precipitation ($P_e$), potential evaporation ($E_p$), interception evaporation ($E_i$), plant transpiration ($E_t$), infiltration into the unsaturated zone ($R_u$), drainage to fast runoff component ($R_f$), delayed fast runoff ($R_{fl}$), groundwater recharge ($R_r$), upwelling groundwater ($R_{GW}$), fast runoff ($Q_f$), groundwater/slow runoff ($Q_s$), total runoff ($Q_m$). Storages [mm]: storage in interception reservoir ($S_i$), storage in unsaturated root zone ($S_u$), storage in groundwater/slow reservoir ($S_s$), storage in fast reservoir ($S_f$). Parameters: interception capacity ($I_{max}$) [mm], maximum upwelling groundwater ($C_{max}$) [mm d$^{-1}$], maximum root zone storage capacity ($S_{umax}$) [mm], splitter ($W$) [-], shape parameter ($\beta$) [-], transpiration coefficient ($C_e$) [-], time lag ($T_{lag}$) [d], reservoir time scales [d] of fast ($K_f$) and slow ($K_s$) reservoirs, areal weights ($p_{HRU}$) [-], time step ($\Delta t$) [d]. Model parameters are shown in bold letters in the table below. The equations were applied to each hydrological response unit (HRU) unless indicated differently.**

| Reservoir system | Water balance equation | Process functions |
|---|---|---|
| **Interception** | $\frac{\Delta S_i}{\Delta t} = P - P_e - E_i \approx 0$ | $E_i = \min\left(E_p, \min\left(P, \frac{\boldsymbol{I_{max}}}{\Delta t}\right)\right)$ <br> $P_e = P - E_i$ |
| **Unsaturated zone** | Plateau/Hillslope/Terrace: <br> $\frac{\Delta S_u}{\Delta t} = R_u - E_t$ <br><br> Wetland: <br> $\frac{\Delta S_u}{\Delta t} = R_u - E_t + R_{GW}$ | $E_t = \min\left((E_p - E_i), \min\left(\frac{S_u}{\Delta t}, (E_p - E_i) \cdot \frac{S_u}{\boldsymbol{S_{u,max}}} \cdot \frac{1}{\boldsymbol{C_e}}\right)\right)$ <br><br> $R_{GW} = \min\left(\left(1 - \frac{S_u}{\boldsymbol{S_{u,max}}}\right) \cdot \boldsymbol{C_{max}}, \frac{\frac{S_s}{\Delta t}}{\boldsymbol{p_{HRU}}}\right)$ <br><br> if $S_u + R_c \cdot \Delta t > \boldsymbol{S_{u,max}}$ : $R_c = \frac{\boldsymbol{S_{u,max}} - S_u}{\Delta t}$ <br><br> Plateau/Terrace/Wetland: <br> $R_u = P_e$ <br> Hillslope: <br> $R_u = (1 - C) \cdot P_e$ <br><br> $C = 1 - \left(1 - \frac{S_u}{\boldsymbol{S_{u,max}}}\right)^{\boldsymbol{\beta}}$ |
| **Fast runoff** | $\frac{\Delta S_f}{\Delta t} = R_{fl} - Q_f$ | $Q_f = \frac{S_f}{\boldsymbol{K_f}}$ <br> Terrace/Wetland: <br> $R_f = \frac{\max(0, S_u - \boldsymbol{S_{umax}})}{\Delta t}$ <br> $R_{fl} = R_f$ <br> Hillslope: <br> $R_f = (1 - \boldsymbol{W}) \cdot C \cdot P_e$ <br> $R_{fl} = R_f * f(\boldsymbol{T_{lag}})$ |
| **Groundwater** | $\frac{\Delta S_s}{\Delta t} = R_{r_{tot}} - R_{GW_{tot}} - Q_s$ | $R_r = \boldsymbol{W} \cdot C \cdot P_e$ <br><br> $R_{r_{tot}} = \sum_{HRU} \boldsymbol{p_{HRU}} \cdot R_r$ <br><br> $R_{GW_{tot}} = \sum_{HRU} \boldsymbol{p_{HRU}} \cdot R_{GW}$ <br><br> $Q_s = \frac{S_s}{\boldsymbol{K_s}}$ |
| **Total runoff** | $Q_m = Q_s + Q_{f_{tot}}$ | $Q_{f_{tot}} = \sum_{HRU} \boldsymbol{p_{HRU}} \cdot Q_f$ |
| **Supporting literature** | (Gharari et al., 2014; Gao et al., 2014; Euser et al., 2015) | |

### 3.3 Parameter selection procedures

To evaluate the information content and thus the utility of altimetry data for the selection of feasible model
parameter sets, a step-wise procedure as specified in detail below was applied (Table 5). Note that given data
scarcity and the related issues of epistemic uncertainties (Beven and Westerberg, 2011; McMillan and
Westerberg, 2015) and equifinality (Beven, 2006; Savenije, 2001) we did not aim to identify the "optimal"
parameter set in what is frequently considered a traditional calibration approach. In most hydrological
applications the available data have limited strength for rigorous model tests (Clark et al., 2015; Gupta et al.,
2008; Jakeman and Hornberger, 1993). Thus, to reduce the risk of rejecting good parameters when they should
have been accepted (Beven, 2010; Hrachowitz and Clark, 2017), we rather attempted to identify and discard the
most implausible parameter sets (Freer et al., 1996) that violate our theoretical understanding of the system or
that are inconsistent with the available data (Knutti, 2008). This allowed us to iteratively constrain the feasible
parameter space and thus the uncertainty around the modelled hydrograph (Hrachowitz et al., 2014). To do so, a
Monte-Carlo sampling strategy with uniform prior parameter distributions was applied to generate $5 \cdot 10^4$ model
realizations. This random set of solutions was in the following steps used as baseline and iteratively constrained
by identifying parameter sets that do not satisfy pre-specified criteria (see below), depending on the data type
and source used.

### 3.3.1 Benchmark: Parameter selection based on observed discharge data

As benchmark, and following a traditional calibration procedure, the model was calibrated with observed daily
discharge based on the Nash-Sutcliffe efficiency ($E_{NS,Q}$, Eq.1 in Table 4) using all complete hydrological years
within the time period 2002 to 2016 (Nash and Sutcliffe, 1970); these are the years starting in the fall of 2004,
2006 and 2008.

To limit the solutions to relatively robust representations of the system while allowing for data and model
uncertainty (e.g. Beven, 2006; Beven and Westerberg, 2011) only parameter sets that resulted in $E_{NS,Q} \geq 0.6$ were
retained as feasible. The hydrological model consisted of 18 free calibration parameters (Table 5, Figure S1)
whose uniform prior distributions are given in Table S1 in the supplementary material with associated parameter
constrains as summarised in Table S2.

**Table 4: Equations used to calculate the model performance**

| Name | Objective function | Symbol explanation | Equation nr. |
|---|---|---|---|
| **Nash-Sutcliffe** | $E_{\mathrm{NS},\theta} = 1 - \dfrac{\sum_t (\theta_{\mathrm{mod}}(t) - \theta_{\mathrm{obs}}(t))^2}{\sum_t (\theta_{\mathrm{obs}}(t) - \overline{\theta_{\mathrm{obs}}})^2}$ | $\theta$: variable | (1) |
| **Spearman-Rank correlation coefficient** | $E_{\mathrm{R,WL}} = \dfrac{\mathrm{cov}(r_{\mathrm{Q_{mod}}}, r_{\mathrm{WL_{obs}}})}{\sigma(r_{\mathrm{Q_{mod}}}) * \sigma(r_{\mathrm{WL_{obs}}})}$ | $r_{\mathrm{Q,mod}}$: ranks of the modelled discharge $r_{\mathrm{WL,obs}}$: rank of the observed water levels | (2) |
| **Relative error** | $E_{\mathrm{R},\theta} = 1 - \dfrac{|\theta_{\mathrm{mod}} - \theta_{\mathrm{obs}}|}{\theta_{\mathrm{obs}}}$ | $\theta$: variable | (3) |
| **Euclidian distance over multiple virtual stations** | $D_{\mathrm{E},\beta,\gamma} = 1 - \sqrt{\left(\sum_i w_i * \left(1 - E_{\beta,\gamma}\right)^2\right)}$ | $w_i$: relative weight of virtual station $i$ $\beta$: model performance metric $\gamma$: parameter selection method | (4) |
| **Euclidian distance over multiple signatures** | $D_{\mathrm{E}} = 1 - \sqrt{\dfrac{1}{(N+M)}\left(\sum_n \left(1 - E_{\mathrm{NS},\theta_n}\right)^2 + \sum_m \left(1 - E_{\mathrm{R},\theta_m}\right)^2\right)}$ | $\theta$: signature $n$: signatures evaluated with Eq.1 with maximum $N$ $m$: signatures evaluated with Eq.3 with maximum $M$ | (5) |

### 3.3.2 Parameter selection based on the seasonal water storage (GRACE)

In a next step we assumed that discharge records in the Luangwa Basin were absent. The starting assumption thus had to be that all model realizations, i.e. all sampled parameter sets, were equally likely to allow feasible representations of the hydrological system. In a stepwise approach, confronting these realizations with different types of data, we sequentially identified and discarded solutions that were least likely to provide meaningful system representations, thereby gradually narrowing down the feasible parameter space.

We first identified and discarded solutions that were least likely to preserve observed the seasonal water storage ($S_{\mathrm{tot}}$) fluctuations. To do so, the monthly modelled total water storage ($S_{\mathrm{tot,mod}} = S_{\mathrm{i}} + S_{\mathrm{u}} + S_{\mathrm{f}} + S_{\mathrm{s}}$) relative to the 2004-2009 time-mean baseline in each grid cell was compared to water storage anomalies observed with GRACE where this same time-mean baseline was used (Tang et al., 2017; Fang et al., 2016; Forootan et al., 2019; Khaki and Awange, 2019).

The model's skill to reproduce the seasonal water storage, i.e. $S_{\mathrm{tot}}$, was assessed using the Nash-Sutcliffe efficiency $E_{\mathrm{NS,Stot}}$ (Eq.1). Note that $E_{\mathrm{NS,Stot,j}}$ was computed at first from the time series of $S_{\mathrm{tot}}$ in each grid cell $j$ which were then averaged to obtain $E_{\mathrm{NS,Stot}}$. If no additional data were available, a hypothetic modeller relying on $E_{\mathrm{NS,Stot}}$ to calibrate a model, may choose only the solution with the highest $E_{\mathrm{NS,Stot}}$ or allow for some uncertainty. To mimic this traditional approach but to balance it with a sufficient number of feasible solutions to be kept for the subsequent steps we here identified and discarded the poorest performing 75% of all solutions in terms of $E_{\mathrm{NS,Stot}}$ as unfeasible for the subsequent modelling steps.

### 3.3.3 Parameter selection based on satellite altimetry data

Next, the remaining feasible parameter sets were used to evaluate their potential to reproduce time series of observed altimetry applying three distinct parameter selection and model evaluation strategies. Assuming again the situation of an ungauged basin (i.e. no time-series of river flow available), we kept for each strategy as feasible the respective 1% best performing parameter sets according to the specific performance metric associated to that strategy.

**Altimetry Strategy 1: Direct comparison of altimetry data to modelled discharge**

In the simplest approach, we directly used altimetry data to correlate observed water levels with modelled discharge based on the Spearman rank correlation coefficient ($E_{R,WL}$; Spearman, 1904) using Eq.2 (Table 4). This strategy, hereafter referred to with subscript WL, i.e. water level, requires the assumption that the relationship between water level and discharge is monotonic. The Spearman rank correlation was applied successfully in previous studies to calibrate a rainfall-runoff model to water level time series (Seibert and Vis, 2016). As there were multiple virtual stations with water level data available in this study, the $E_{R,WL}$ was computed at each location simultaneously. The individual values $E_{R,WL}$ were weighted based on the record length of the corresponding virtual stations and then combined into the Euclidean distance as aggregate metric $D_{E,R,WL}$ with Eq.4.

**Table 5: Overview of the parameter identification strategies applied in this study**

| Strategy | Calibration data | Objective function | Parameter group | Calibration parameters | Comments | Q – h conversion | Benefits (+) & limitations (-) |
|---|---|---|---|---|---|---|---|
| **Discharge (reference)** | Discharge (at basin outlet) | $E_{NS,Q}$ (Eq.1) | Entire basin<br>Plateau & Terrace<br>Hillslope<br>Wetland<br>River profile | $K_s$, $C_e$<br>$I_{max}$, $S_{umax}$, $K_f$, $W$<br>$I_{max}$, $S_{umax}$, $K_f$, $W$, $\beta$, $T_{lag}$<br>$I_{max}$, $S_{umax}$, $K_f$, $W$, $C_{max}$<br>$v$<br>Total: 18 | Traditional model calibration on observed flow data<br>Combination of 8 different flow signatures | - | - |
| **Seasonal water storage** | GRACE | $E_{NS,Stot}$ (Eq.1) | Entire basin<br>Plateau & Terrace<br>Hillslope<br>Wetland<br>River profile | $K_s$, $C_e$<br>$I_{max}$, $S_{umax}$, $K_f$, $W$<br>$I_{max}$, $S_{umax}$, $K_f$, $W$, $\beta$, $T_{lag}$<br>$I_{max}$, $S_{umax}$, $K_f$, $W$, $C_{max}$<br>$v$<br>Total: 18 | No discharge data used | - | - |
| **Altimetry Strategy 1** | Altimetry (at 18 virtual stations) & GRACE | Altimetry: $D_{E,R,WL}$ (Eq.2,4)<br>GRACE: $E_{NS,Stot}$ (Eq.1) | Entire basin<br>Plateau & Terrace<br>Hillslope<br>Wetland<br>River profile | $K_s$, $C_e$<br>$I_{max}$, $S_{umax}$, $K_f$, $W$<br>$I_{max}$, $S_{umax}$, $K_f$, $W$, $\beta$, $T_{lag}$<br>$I_{max}$, $S_{umax}$, $K_f$, $W$, $C_{max}$<br>$v$<br>Total: 18 | No discharge data used<br>Combination of 18 virtual stations<br>Combined with GRACE | - | + No extra parameters or data needed<br>+ Assumption: monotonic relation between discharge and river water level<br>- Focus on dynamics only, not volume |
| **Altimetry Strategy 2** | Altimetry (at 18 virtual stations) & GRACE | Altimetry: $D_{E,NS,RC}$ (Eq.1,4)<br>GRACE: $E_{NS,Stot}$ (Eq.1) | Entire basin<br>Plateau & Terrace<br>Hillslope<br>Wetland<br>River profile | $K_s$, $C_e$<br>$I_{max}$, $S_{umax}$, $K_f$, $W$<br>$I_{max}$, $S_{umax}$, $K_f$, $W$, $\beta$, $T_{lag}$<br>$I_{max}$, $S_{umax}$, $K_f$, $W$, $C_{max}$<br>$v$, $a_1$, $a_2$, $a_3$, $a_4$, $b_1$, $b_1$, $b_3$, $b_4$<br>Total: 26 | No discharge data used<br>Combination of 18 virtual stations<br>Combined with GRACE | Calibrated Rating curve | + No extra data needed<br>- Two extra parameters per cross-section |
| **Altimetry Strategy 3** | Altimetry (at 18 virtual stations) & GRACE | Altimetry: $D_{E,NS,SM}$ (Eq.1,4)<br>GRACE: $E_{NS,Stot}$ (Eq.1) | Entire basin<br>Plateau & Terrace<br>Hillslope<br>Wetland<br>River profile | $K_s$, $C_e$<br>$I_{max}$, $S_{umax}$, $K_f$, $W$<br>$I_{max}$, $S_{umax}$, $K_f$, $W$, $\beta$, $T_{lag}$<br>$I_{max}$, $S_{umax}$, $K_f$, $W$, $C_{max}$<br>$v$, $k$<br>Total: 18 | No discharge data used<br>Combination of 18 virtual stations<br>Combined with GRACE | Strickler-Manning | + Only 1 extra parameter<br>- Cross-section data needed<br>- Assumption: constant roughness in space and time |
| **Water level Strategy 1** | Water level (at basin outlet) & GRACE | Altimetry: $E_{NS,SM,GE}$ (Eq.1)<br>GRACE: $E_{NS,Stot}$ (Eq.1) | Entire basin<br>Plateau & Terrace<br>Hillslope<br>Wetland<br>River profile | $K_s$, $C_e$<br>$I_{max}$, $S_{umax}$, $K_f$, $W$<br>$I_{max}$, $S_{umax}$, $K_f$, $W$, $\beta$, $T_{lag}$<br>$I_{max}$, $S_{umax}$, $K_f$, $W$, $C_{max}$<br>$v$, $k$<br>Total: 19 | No discharge data used<br>Combined with GRACE | Strickler-Manning | + Only 1 extra parameter<br>- Cross-section data needed<br>- Assumption: constant roughness in space and time |
| **Water level Strategy 2** | Water level (at basin outlet) & GRACE | Altimetry: $E_{NS,SM,ADCP}$ (Eq.1)<br>GRACE: $E_{NS,Stot}$ (Eq.1) | Entire basin<br>Plateau & Terrace<br>Hillslope<br>Wetland<br>River profile | $K_s$, $C_e$<br>$I_{max}$, $S_{umax}$, $K_f$, $W$<br>$I_{max}$, $S_{umax}$, $K_f$, $W$, $\beta$, $T_{lag}$<br>$I_{max}$, $S_{umax}$, $K_f$, $W$, $C_{max}$<br>$v$, $k$<br>Total: 19 | No discharge data used<br>Combined with GRACE | Strickler-Manning | + Only 1 extra parameter<br>- Cross-section data needed<br>- Assumption: constant roughness in space and time |

**Altimetry Strategy 2: Rating curves**

In the second strategy, as successfully applied in previous studies (Getirana and Peters-Lidard, 2013; Jian et al., 2017), model parameters were selected based on the models' ability to reproduce water levels by converting the modelled discharge to water levels, assuming these two are related through a rating curve in the form of a power function (Rantz, 1982):

$$Q = a * (h - h_0)^b \qquad\qquad (6)$$

Where $h$ is the water level, $h_0$ a reference water level, and $a$ and $b$ are two additional free calibration parameters,
determining the shape of the function and lumping the combined influences of different river cross-section characteristics, such as geometry or roughness. Note, that here for each virtual station $h_0$ is the elevation that corresponds to the water level of the Google Earth image with the lowest flow available, corresponding to the assumption of no-flow at that time. This strategy is hereafter referred to as with subscript RC, i.e. rating curve. As river-cross sections vary in space, each of the 18 virtual stations would require an individual set of these
parameters $a$ and $b$. To limit the number of additional calibration parameters, we here classified the river-cross sections of the 18 virtual stations into 4 groups (Figure 1A and Figure 3). For cross-sections within each class, i.e. geometrically similar, the same values for $a$ and $b$ were used, resulting in 4 sets of $a$ and $b$ and thus a total of 8 additional calibration parameters. The river cross-sections were extracted from global high-resolution terrain data available on Google Earth (see Section 2.1.4). The modelled river water levels were evaluated against the
observed water levels at each virtual station using the Nash-Sutcliffe efficiency $E_{NS,RC}$ (equivalent to Eq.1 in Table 4), weighted based on the record length of the corresponding virtual stations and then combined into the Euclidean distance $D_{E,NS,RC}$ as an aggregated performance metric (Eq.4).

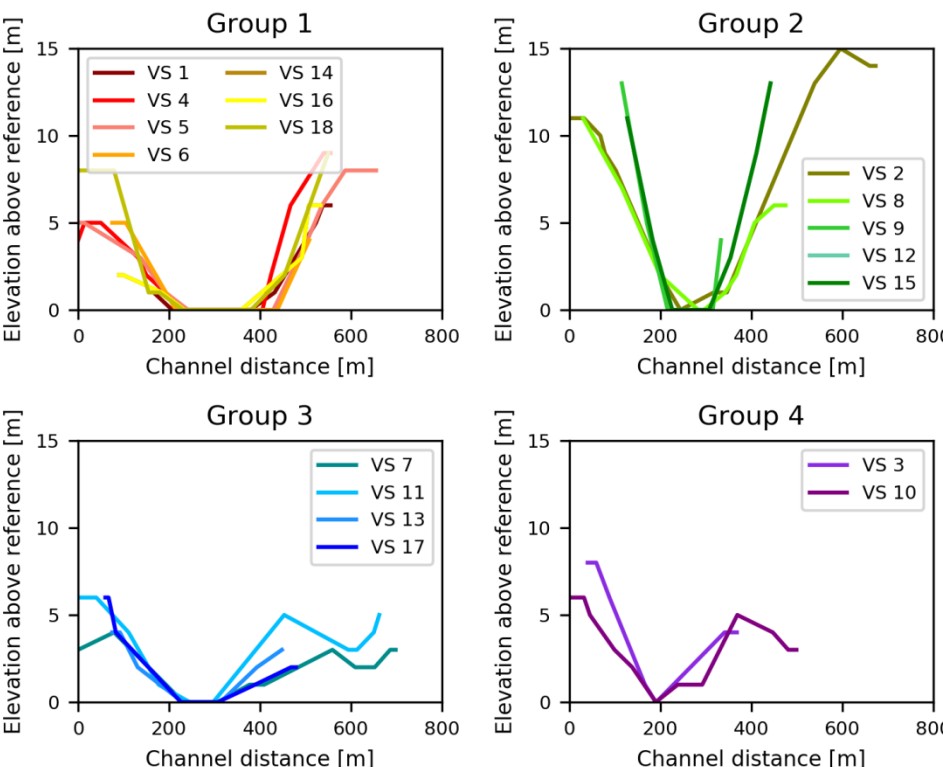

**Figure 3: River profiles at 18 virtual stations (VS) divided into four groups. The reference level is equal to the lowest**
**water level in the river profile for each location separately.**

**Altimetry Strategy 3: Strickler-Manning equation**

As a third strategy, we converted the modelled discharge to river water levels using the Strickler-Manning equation (Manning, 1891):

$$Q = k * i^{\frac{1}{2}} * A * R^{\frac{2}{3}} \tag{7}$$

Where $k$ is a roughness parameter, here treated as free calibration parameter and assumed constant for all virtual stations, $i$ is the mean channel slope, extracted here over a distance of 10 km, while $A$ and $R$ are the river cross-section area and hydraulic radius. Assuming trapezoidal cross-sections (see Figure 4 as illustrative example), $A$ and $R$ were calculated for each cross section according to:

$$A = B * d + \frac{1}{2} * d^2 * (i_1 + i_2) \tag{8}$$

$$R = \frac{A}{B + d * \left((1 + i_1^2)^{\frac{1}{2}} + (1 + i_2^2)^{\frac{1}{2}}\right)} \tag{9}$$

$$d = h - h_0 \tag{10}$$

Where $B$ is the assumed river bed width, $i_1$ and $i_2$ are the river bank slopes, $d$ the water depth, $h$ the water level
and $h_0$ the reference water level, here assumed to be the lowest observed river water level to limit the number of calibration parameters. In contrast to previous studies that use a similar approach but relied on locally observed river-cross sections (Michailovsky et al., 2012; Hulsman et al., 2018; Liu et al., 2015), here both, the river bed geometries (Figure 3) at and the channel slopes upstream of the 18 virtual stations were computed using high-resolution terrain data retrieved from Google Earth (see Section 2.1.4). Similar data sources were already used in
previous studies to extract the river geometry (e.g. Michailovsky et al., 2012; Pramanik et al., 2010; Gichamo et al., 2012). The reader is referred to Table S3 in the supplementary material for the values of the variables for each virtual station. This strategy is hereafter referred to as with subscript SM, i.e. Strickler-Manning.

Equivalent to above, the modelled river water levels were then evaluated against the observed water levels at each virtual station using the Nash-Sutcliffe efficiency $E_{NS,SM}$ (equivalent to Eq.1), weighted based on the record
length of the corresponding virtual stations and then combined into the Euclidean distance $D_{E,NS,SM}$ as an aggregated performance metric (Eq.4).

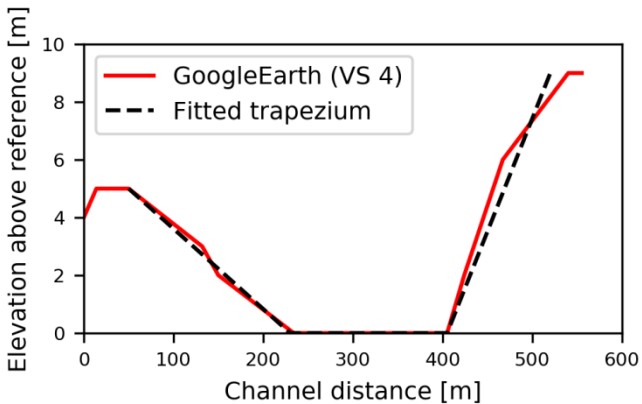

**Figure 4: Example of approximating a trapezoidal cross-section (black) into the Google Earth based cross-section data (red) for virtual station "VS 4" (see also Figure 1A and Figure 3). The reference level is equal to the lowest water**
**level in the river profile.**

### 3.3.4 Parameter selection based on daily river water level at the basin outlet

For the previous parameter identification strategy (Altimetry Strategy 3), river geometry information was extracted from high-resolution terrain data retrieved from Google Earth which have a low accuracy. Unfortunately, more accurate cross-section information from in-situ surveys was only available at the Great East Road Bridge gauging station, i.e. the basin outlet, where, in turn, no altimetry observations were available. That is why water level time series were used to illustrate the influence of the cross-section accuracy.

As shown in Figure 5, the Google Earth based above-water cross-section at the basin outlet corresponded in general well to the field survey considering that satellite images have limited spatial resolution. However, the in-situ measurement also illustrated the relevance of the submerged part of the channel cross-section at that location on the day the image was taken (June 2$^{nd}$ 2008).

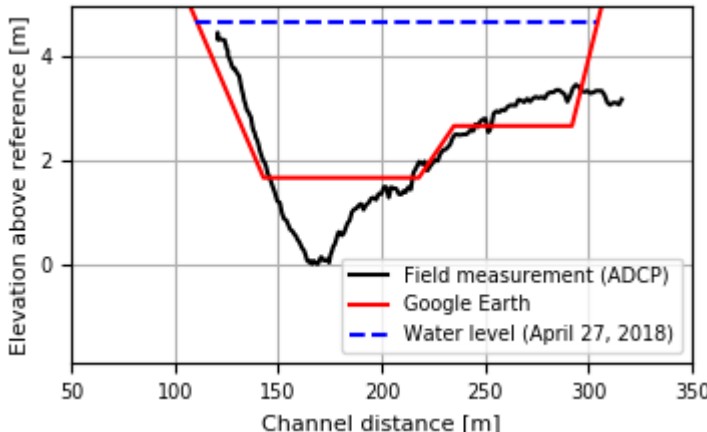

**Figure 5: River cross-section at Luangwa Bridge obtained from Google Earth and detailed field survey including the river water level on June 2$^{nd}$ 2008. Field measurements were done with an Acoustic Doppler Current Profiler (ADCP) on April 27$^{th}$ 2018 at the coordinates 30$^{o}$ 13' E and 15$^{o}$ 00' S; the satellite image was taken on June 2$^{nd}$ 2008. The reference level is equal to the lowest elevation level measured with the ADCP.**

**Water level Strategy 1: River geometry information extracted from Google Earth**

First, cross-section information was extracted from global high-resolution terrain data available on Google Earth (subscript GE) and used with the Strickler-Manning equation (Eq.7) to convert the modelled discharge to water levels. This was combined with GRACE observations to restrict the parameter space in an equivalent way as in Section 3.3.3. The model performance with respect to river water levels was calculated with the Nash-Sutcliffe efficiency $E_{NS,SM,GE}$ (Eq.1).

**Water level Strategy 2: River geometry information obtained from a detailed field survey**

Second, cross-section information obtained from a detailed field survey with an ADCP (subscript ADCP) was used with the Strickler-Manning equation (Eq.7) to convert the modelled discharge to water levels. This was combined with GRACE observations to restrict the parameter space in an equivalent way as in Section 3.3.3. The model performance with respect to river water levels was calculated with the Nash-Sutcliffe efficiency $E_{NS,SM,ADCP}$ (Eq.1).

## 3.4 Model evaluation

For each calibration strategy, the performance of all model realizations was evaluated post-calibration with respect to discharge using seven additional hydrological signatures (e.g. Sawicz et al., 2011; Euser et al., 2013) to assess the skill of the model to reproduce the overall response of the system and thus the robustness of the selected parameters (Hrachowitz et al., 2014). The signatures included the logarithm of the daily flow time series (hereafter referred to with the subscript logQ), the flow duration curve (FDC), its logarithm (logFDC), the mean seasonal runoff coefficient during dry periods (April - September; RCdry), the mean seasonal runoff coefficient during the wet periods (October - March; RCwet), the autocorrelation function of daily flow (AC) and the rising limb density of the hydrograph (RLD). An overview of these signatures can be found in Table 6, and more detailed explanations in Euser et al. (2013) and references therein. As performance measures for the model to reproduce the individual observed signatures the Nash-Sutcliffe efficiency ($E_{NS,logQ}$, $E_{NS,FDC}$, $E_{NS,logFDC}$, $E_{NS,AC}$; equivalent to Eq.1 in Table 4) and a metric based on the relative error ($E_{R,RCdry}$, $E_{R,RCwet}$, $E_{R,RLD}$; equivalent to Eq.3) were used (Euser et al., 2013). The signatures where combined, with equal weights, into one objective function, which was formulated based on the Euclidian distance $D_E$ (Eq.5) so that a value of 1 indicates a "perfect" model (Schoups et al., 2005):

**Table 6: Overview of flow signatures used in this study**

| Flow signature | Explanation | Function | Model performance equation |
|---|---|---|---|
| Q | Daily flow time series | - | $E_{NS,Q} = 1 - \frac{\sum_t (Q_{mod,t} - Q_{obs,t})^2}{\sum_t (Q_{obs,t} - \overline{Q_{obs}})^2}$ |
| logQ | Logarithm of daily flow time series | - | $E_{NS,logQ} = 1 - \frac{\sum_t (Q_{log,mod,t} - Q_{log,obs,t})^2}{\sum_t (Q_{log,obs,t} - \overline{Q_{log,obs}})^2}$ |
| FDC | Flow duration curve | - | $E_{NS,FDC} = 1 - \frac{\sum_t (Q_{sort,mod,t} - Q_{sort,obs,t})^2}{\sum_t (Q_{sort,obs,t} - \overline{Q_{sort,obs}})^2}$ |
| logFDC | Logarithm of flow duration curve | - | $E_{NS,logFDC} = 1 - \frac{\sum_t (Q_{log,sort,mod,t} - Q_{log,sort,obs,t})^2}{\sum_t (Q_{log,sort,obs,t} - \overline{Q_{log,sort,obs}})^2}$ |
| RCdry | Runoff coefficient during dry periods | $RC_{dry} = \frac{Q_{dry}}{P_{dry}}$ | $E_{R,RCdry} = 1 - \frac{|RC_{dry,mod} - RC_{dry,obs}|}{RC_{dry,obs}}$ |
| RCwet | Runoff coefficient during wet periods | $RC_{wet} = \frac{Q_{wet}}{P_{wet}}$ | $E_{R,RCwet} = 1 - \frac{|RC_{wet,mod} - RC_{wet,obs}|}{RC_{wet,obs}}$ |
| AC | Autocorrelation function | $AC_t = \frac{\sum_i (Q_i - \bar{Q}) * (Q_{i+t} - \bar{Q})}{\sum (Q_i - \bar{Q})^2}$ | $E_{NS,AC} = 1 - \frac{\sum_t (AC_{mod,t} - AC_{obs,t})^2}{\sum_t (AC_{obs,t} - \overline{AC_{obs}})^2}$ |
| RLD | Rising limb density | $RLD = \frac{N_{peaks}}{T_r}$ | $E_{R,RLD} = 1 - \frac{|RLD_{mod} - RLD_{obs}|}{RLD_{obs}}$ |

## 4 Results and discussion

### 4.1 Parameter selection and model performance

The complete set of all model realizations unsurprisingly results in a wide range of model solutions (Figure 6A), with $E_{NS,Q}$ ranging from -6.4 to 0.78 and with the combined performance metric of all signatures $D_E$ ranging from -334 to 0.79 (Figure 7). With respect to the individual flow signatures, the model performance varied such that the largest range was found in $E_{NS,Q}$ and smallest in $E_{NS,AC}$ as visualised in Figure 7 and tabulated in Table

S4. Although containing relatively good solutions, this full set of all realizations clearly also contained many parameter sets that considerably over- and/or underestimate flows.

### 4.1.1 Benchmark: Parameter selection based on observed discharge data

For the benchmark case, applying the traditional model calibration approach using discharge data, this parameter selection and calibration strategy results in a reasonable model performance, in which the seasonal but also the daily flow dynamics and magnitudes are in general well captured as shown in Figure 6B. For some years, a number of solutions overestimate flows in the wet season and underestimate flows during the dry season, when the river becomes a small meandering stream with almost annual morphological changes which is difficult to meaningfully observe. The best performing solution has a calibration objective function $E_{NS,Q,opt} = 0.78$ (5/95$^{th}$ percentiles of all feasible solutions $E_{NS,Q,5/95} = 0.61 - 0.75$; Figure 7 and Table 7). For the post-calibration evaluation of all retained solutions, it was observed that most signatures are well reproduced by the majority of solutions, except for the dry season runoff coefficient ($RC_{dry}$; Figure 7 and Table S4). This resulted in aggregated model performances, combining all signatures, of $D_{E,5/95} = 0.55 - 0.76$ with the above identified best performing solution (i.e. $E_{NS,Q,opt}$) reaching a value of $D_{E,opt} = 0.60$.

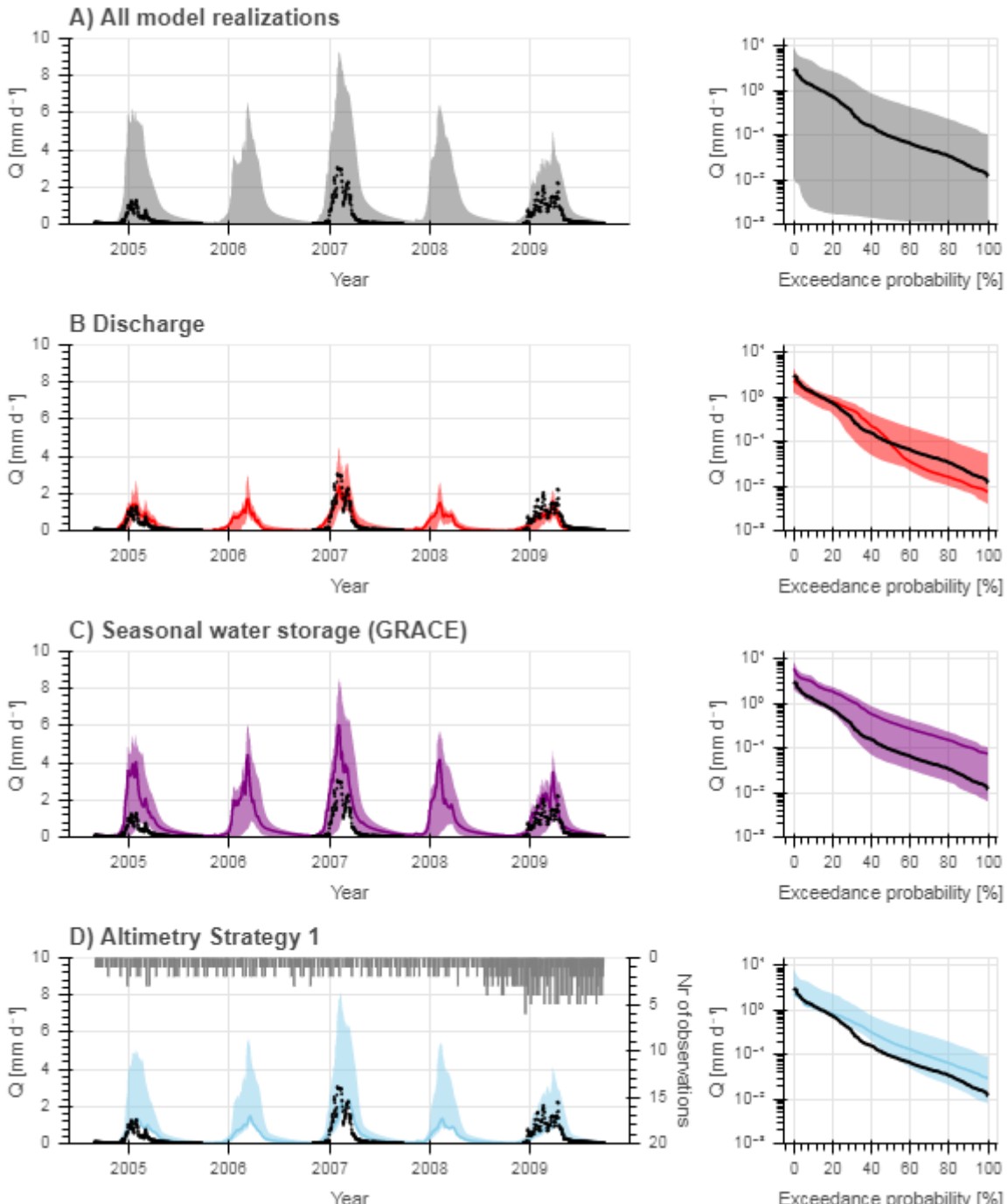

Figure 6: Range of model solutions. The left panel shows the hydrograph and the right panel the flow duration curve of the recorded (black) and modelled discharge: the line indicates the solution with the highest calibration objective function ($E_{NS}$ or $D_E$) and the shaded area the envelope of the solutions retained as feasible. A) All model solutions included; solutions retained as feasible based on B) discharge (i.e. "traditional calibration"; $E_{NS,Q}$), C) GRACE ($E_{NS,Stot}$), and D) Altimetry Strategy 1 only ($D_{E,R,WL}$). The grey bars in the left subplot D indicate the number of altimetry observations available for each day.

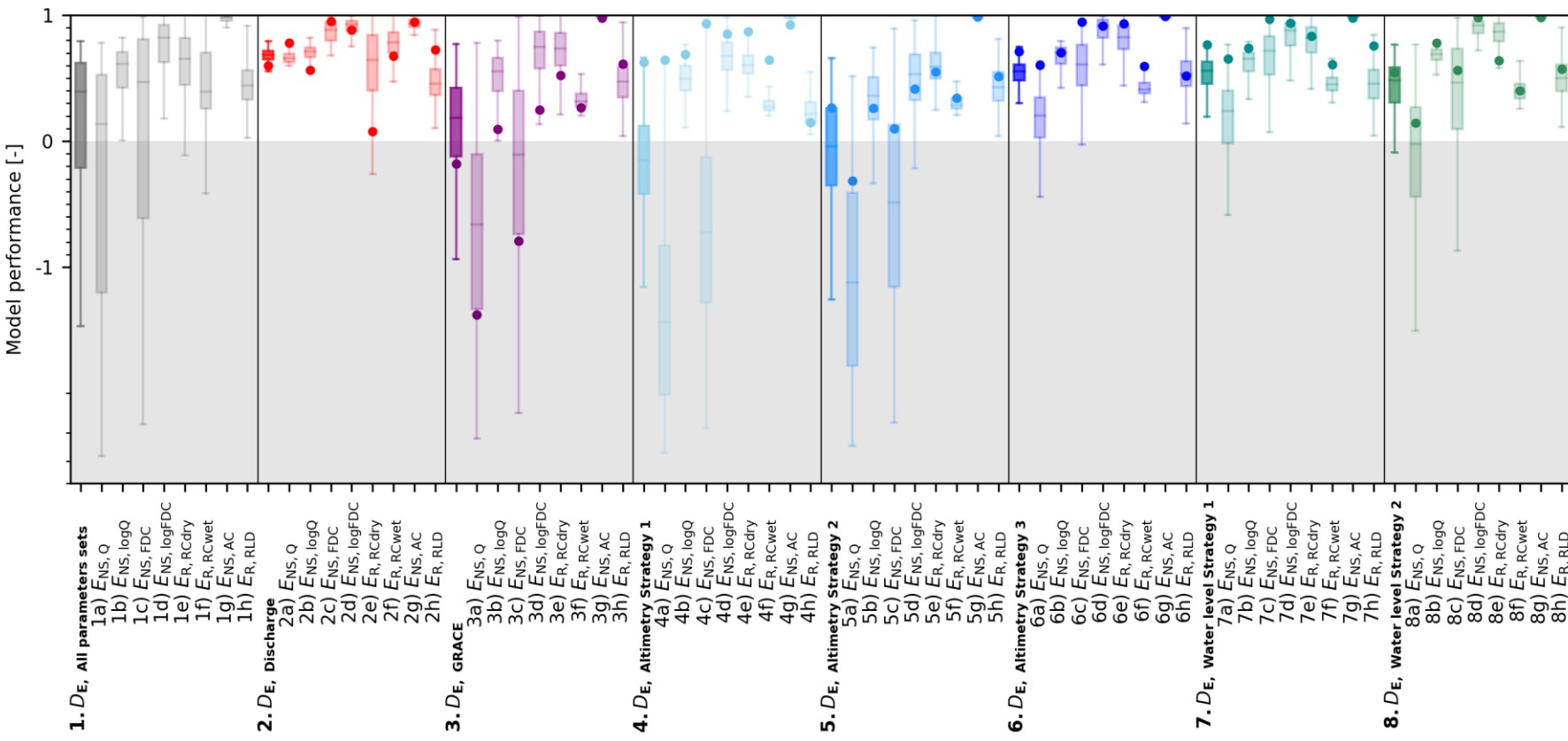

**Figure 7: Comparison of different data sources to identify feasible parameter sets. Data sources applied: 1) All random parameters (no data), 2) Discharge, 3) GRACE, 4) Altimetry data combined with GRACE (Altimetry Strategy 1), 5) Altimetry data using the rating curves combined with GRACE (Altimetry Strategy 2), and 6) Altimetry data using the Strickler – Manning equation combined with GRACE (Altimetry Strategy 3), and 7) Daily river water level combined with GRACE using the Strickler – Manning equation and cross-section information retrieved from Google Earth (Water level Strategy 1), or 8) obtained from a detailed field survey with an Acoustic Doppler Current Profiler (ADCP, Water level Strategy 2). The boxplots visualise the spread in the overall model performance $D_E$ with respect to discharge and the following individual signatures: a) daily discharge ($E_{NS,Q}$), b) its logarithm ($E_{NS,logQ}$), c) flow duration curve ($E_{NS,FDC}$), d) its logarithm ($E_{NS,logFDC}$), e) average runoff coefficient during the dry season ($E_{R,RCdry}$), f) average seasonal runoff coefficient during the wet season ($E_{R,RCwet}$), g) autocorrelation function ($E_{NS,AC}$), and h) rising limb density ($E_{R,RLD}$). The dots visualise the model performance when selecting the parameter set with the highest model efficiency according to each parameter identification strategy.**

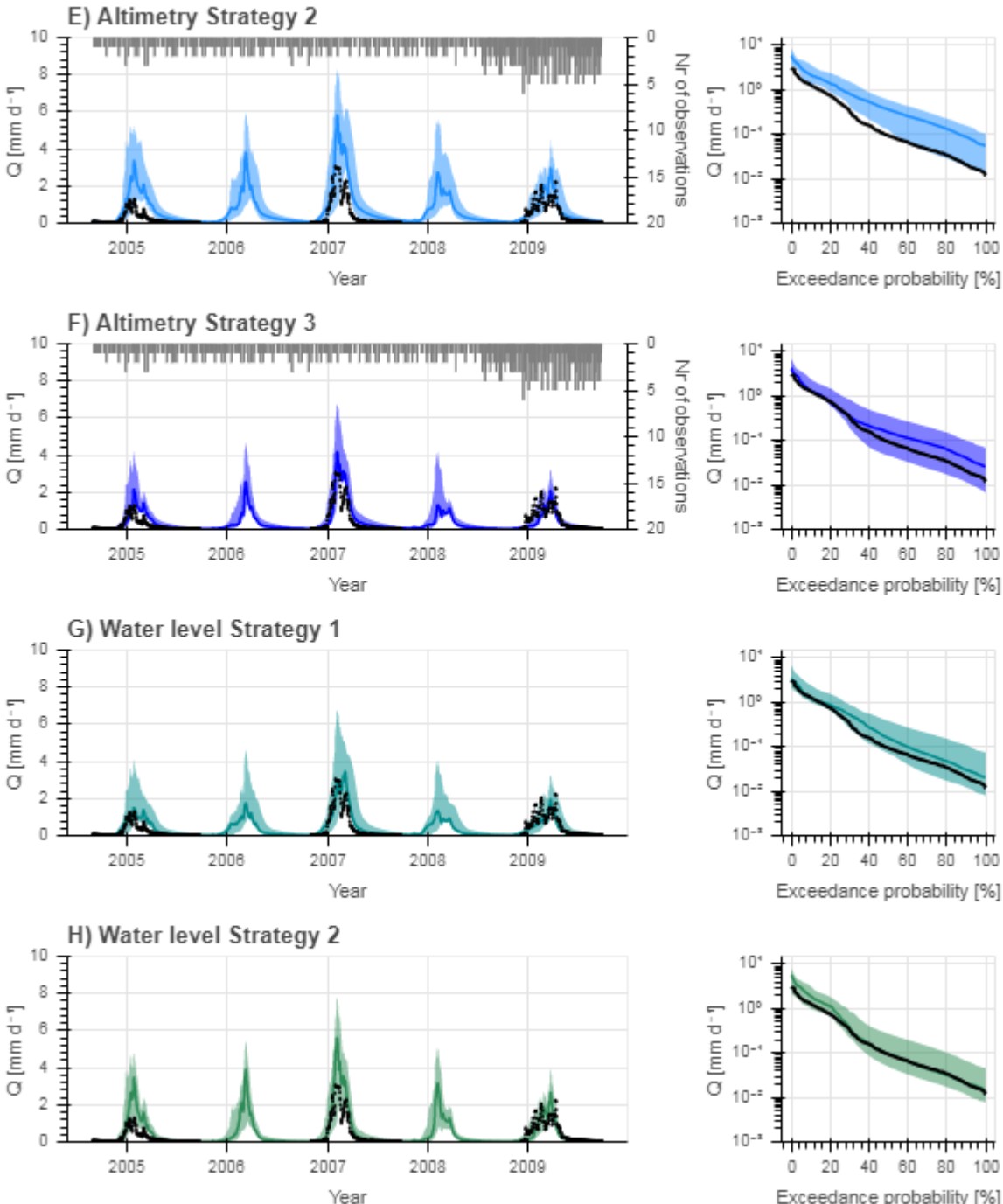

**Figure 8: Range of model solutions.** The left panel shows the hydrograph and the right panel the flow duration curve of the recorded (black) and modelled discharge: the line indicates the solution with the highest calibration objective function ($E_{NS}$ or $D_E$) and the shaded area the envelope of the solutions retained as feasible. Solutions retained as feasible based on E) Altimetry Strategy 2 using rating curves for the discharge – water level conversion ($D_{E,NS,RC}$), F) 
Altimetry Strategy 3 using the Strickler-Manning equation for the discharge – water level conversion ($D_{E,NS,SM}$), and G) Daily in-situ water level using the Strickler Manning equation for the discharge – water level conversion with cross-section information retrieved from Google Earth (Water level strategy 1; $E_{NS,SM,GE}$) or H) obtained from a detailed field survey with an Acoustic Doppler Current Profiler (ADCP; Water level strategy 2; $E_{NS,SM,ADCP}$). The grey bars in the left subplots E and F indicate the number of altimetry observations available for each day.

**Table 7: Summary of the model results: elimination of unfeasible parameter sets and detection of optimal parameter set according to each parameter identification strategy including the corresponding model performance range ($E_{NS,Q}$, $D_E$) indicating the model's skill to reproduce the discharge during the benchmark time period. For each strategy, the model efficiency for the optimal parameter set is summarised together with the corresponding performance metrics with respect to discharge ($E_{NS,Q,opt}$, $D_{E,opt}$). For all parameter sets retained as feasible, the maximum ($E_{NS,Q,max}$, $D_{E,max}$) and 5/95 percentiles ($E_{NS,Q,5/95}$, $D_{E,5/95}$) of all performance metrics with respect to discharge are summarised. Data sources used for the parameter set selection: 1) All parameter sets (no data), 2) Discharge, 3) GRACE, 4) Altimetry combined with GRACE (Altimetry Strategy 1), 5) Altimetry data using rating curves combined with GRACE (Altimetry Strategy 2), 6) Altimetry data using the Strickler – Manning equation combined with GRACE (Altimetry Strategy 3), and 7) Daily river water level combined with GRACE using the Strickler – Manning equation and cross-section information retrieved from Google Earth (Water level Strategy 1), or 8) obtained from a detailed field survey with an Acoustic Doppler Current Profiler (ADCP, Water level Strategy 2).**

| | Optimal parameter set | | Feasible parameter sets | |
| --- | --- | --- | --- | --- |
| | Model efficiency | $E_{NS,Q,opt}$ ($D_{E,opt}$) | $E_{NS,Q,max}$ ($E_{NS,Q,5/95}$) | $D_{E,max}$ ($D_{E,5/95}$) |
| **1) All parameters sets** | - | - | 0.78 (-3.8 − 0.68) | 0.79 (-1.4 − 0.71) |
| **2) Discharge** | $E_{NS,Q,opt}$ = 0.78 | 0.78 (0.60) | 0.78 (0.61 − 0.75) | 0.79 (0.55 − 0.76) |
| **3) Seasonal water storage (GRACE)** | $E_{NS,Stot,opt}$ = 0.56 | -1.4 (-0.18) | 0.78 (-2.3 − 0.38) | 0.77 (-0.58 − 0.62) |
| **4) Altimetry Strategy 1: Compare altimetry to discharge** | $D_{E,R,WL,opt}$ = 0.76 | 0.65 (0.63) | 0.65 (-2.9 − 0.10) | 0.66 (-0.83 − 0.50) |
| **5) Altimetry Strategy 2: Rating curves** | $D_{E,NS,RC,opt}$ = -0.50 | -0.31 (0.27) | 0.51 (-2.6 − 0.25) | 0.66 (-0.72 − 0.56) |
| **6) Altimetry Strategy 3: Strickler-Manning equation** | $D_{E,NS,SM,opt}$ = -1.4 | 0.60 (0.71) | 0.63 (-0.31 − 0.50) | 0.75 (0.36 − 0.67) |
| **7) Water level Strategy 1: satellite based cross-section** | $E_{NS,SM,GE,opt}$ = -1.8 | 0.65 (0.77) | 0.77 (-0.48 − 0.60) | 0.77 (0.28 − 0.70) |
| **8) Water level Strategy 2: in-situ cross-section** | $E_{NS,SM,ADCP,opt}$ = 0.79 | 0.14 (0.55) | 0.77 (-1.1 − 0.50) | 0.77 (0.03 − 0.67) |

### 4.1.2 Parameter selection based on the seasonal water storage (GRACE)

Starting from the set of all model realizations (Figures 6A and 7), and assuming no discharge observations are available, we identified and discarded parameter sets as unfeasible when they did not meet the previously defined criteria to reproduce the seasonal water storage ($E_{NS,Stot}$; see Section 3.3.2). The range of random model realizations with respect to the total water storage is visualised in Figure 9. The sub-set of solutions retained as feasible resulted in a significant reduction in the uncertainty around the modelled variables, which is illustrated by the narrower 5/95[th] percentiles of the solutions compared to the set of all realizations, as shown in Figure 6C. The feasible solutions with respect to the GRACE reached $E_{NS,Stot,opt}$ = 0.56 ($E_{NS,Stot,5/95}$ = 0.45 − 0.52) (Figure 7, Table 7). These parameter sets were then used to evaluate the model for the years 2004, 2006, 2008 used in the benchmark case. While the flow dynamics are captured relatively well, many of the retained solutions considerably overestimated flows across all seasons (Figure 6C) resulting in a decreased performance with respect to the individual flow signatures, only the dry runoff coefficient ($E_{R,RCdry}$) improved significantly compared to the benchmark as shown in Table S4 and Figure 7. The parameter set associated with the best performing model with respect to GRACE ($E_{NS,Stot,opt}$) resulted for the benchmark period in a $E_{NS,Q}$ = -1.4 ($E_{NS,Q,5/95}$ = -2.3 − 0.38) and the corresponding $D_{E,opt}$ = -0.18 ($D_{E,5/95}$ = -0.58 − 0.62) with respect to discharge (Figure 7, Table 7). As illustrated in Figure 7 and Figure 6C, many parameter sets that resulted in implausible representations of the seasonal signals were eliminated. However, as also indicated by the rather modest values of $E_{NS,Q}$ and $D_E$ with respect to discharge, the data source used here obviously contained only limited information to avoid the over predictions of flow during all wet seasons. The sequence of applying first GRACE

and then altimetry, or the reverse, did not affect the identification of feasible parameter sets when using altimetry
        data as shown in Figure S8. However, it did affect the selection of the "best" parameter set.

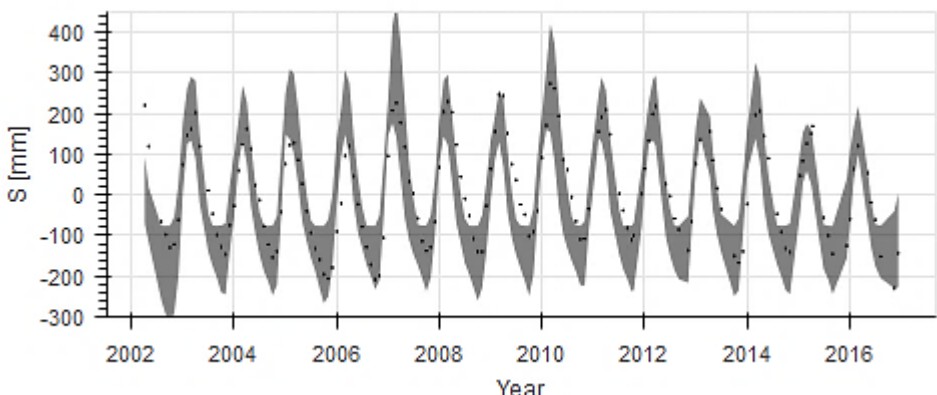

**Figure 9: Range of random model realizations with respect to the total water storage (grey) including the observation**
**according to GRACE (black)**

### 4.1.3 Parameter selection based on satellite altimetry data

**Altimetry Strategy 1: Directly compare altimetry data to modelled discharge**

The first approach, Altimetry Strategy 1, resulted in an overestimation of in particular intermediate and low
flows as shown in Figure 6D. The feasible solutions reached an optimum of $D_{E,R,WL,opt} = 0.76$ ($D_{E,R,WL,5/95} = 0.74$

$- 0.75$) with respect to altimetry observations. Focusing on the model's skill to reproduce the observed discharge
using these feasible parameter sets for the benchmark period, the parameter set associated with the best
performing model with respect to altimetry ($D_{E,R,WL,opt}$) resulted in a $E_{NS,Q} = 0.65$ ($E_{NS,Q,5/95} = -2.9 - 0.10$) and
$D_E = 0.63$ ($D_{E,5/95} = -0.83 - 0.50$) with respect to discharge (Figure 7, Table 7). Hence, the parameter set with the
highest model performance with respect to altimetry, did not perform best with respect to discharge as shown in

Table 7 and Figure S7. While the optimum model performance with respect to discharge was similar to the
benchmark, the very wide range in the 5/95[th] percentiles of the solutions indicated that this strategy has only
limited potential to identify implausible parameter sets. This was also the case with respect to the individual flow
signatures as shown in Figure 7 and Table S4.

**Altimetry Strategy 2: Rating curves**

The second approach, Altimetry Strategy 2, also resulted in an overestimation of the flows (Figure 8E). The
feasible solutions reached an optimum of $D_{E,NS,RC,opt} = -0.50$ ($D_{E,NS,RC,5/95} = -1.0 - -0.77$) with respect to altimetry
observations. As example, Figure S6A visualises the simulated and observed river water level at Virtual Station
4 (Figure 1) where the model significantly underestimated the stream levels. Focusing on the model's skill to

reproduce the discharge using these parameter sets for the benchmark period, the parameter set associated with
the best performing model with respect to altimetry ($D_{E,NS,RC,opt}$) resulted in $E_{NS,Q} = -0.31$ ($E_{NS,Q,5/95} = -2.6 -$
$0.25$) and $D_E = 0.27$ ($D_{E,5/95} = -0.72 - 0.56$) with respect to discharge (Figure 7, Table 7). Hence similar to
Altimetry Strategy 1, the best parameter set with respect to altimetry, did not perform best with respect to
discharge (see Table 7 and Figure S7). The optimum model performance with respect to discharge was worse

compared to the benchmark, and the wide range in the 5/95[th] percentiles of the solutions indicated this strategy
poorly identified the feasible parameter sets. This was also the case with respect to the individual flow signatures

as shown in Figure 7 and Table S4. Only the dry runoff coefficient ($E_{RCdry}$) improved significantly compared to the benchmark.

**Altimetry Strategy 3: Strickler-Manning equation**

The third approach, Altimetry Strategy 3, resulted in improved flow predictions compared to the other two strategies using altimetry data (Figure 8F). Even though the feasible solutions exhibit a very poor ability to reproduce the altimetry data, with an optimum of $D_{E,NS,SM,opt}$ = -1.4 ($D_{E,NS,SM,5/95}$ = -3.8 − -1.8), the model's skill to reproduce the discharge for the benchmark period using these parameter sets, significantly increased compared to the two alternative strategies. As example, Figure S6B visualises the simulated and observed river water level at Virtual Station 4 (Figure 1) where the model simulated the stream levels relatively well. The parameter set associated with the best performing model with respect to altimetry ($D_{E,NS,SM,opt}$) resulted in $E_{NS,Q}$ = 0.60 ($E_{NS,Q,5/95}$= -0.31 − 0.50) and $D_E$ = 0.71 ($D_{E,5/95}$ = 0.36 − 0.67) with respect to discharge (Figure 7, Table 7). While the optimum model performance with respect to discharge was worse compared to the benchmark, the 5/95[th] percentiles of the solutions were significantly constrained by the removal of many implausible parameter sets. This was valid for the performance with respect to the individual flow signatures ($E_{NS,θ}$ and $E_{R,θ}$) and overall flow response ($D_E$) as shown in Figure 7 and Table S4. This indicated that, although the model performance with respect to altimetry observations was low, this strategy contains valuable information to considerably constrain the feasible solution space.

**4.1.4 Parameter selection based on daily river water level at the basin outlet**

**Water level Strategy 1: River geometry information extracted from Google Earth**

The parameter identification strategy "Water level Strategy 1", using cross-section information extracted from Google Earth, resulted in a poor simulation of the river water level (Figure 10A) with an optimal objective function value with respect to river water levels of $E_{NS,SM,GE,opt}$ = -1.8 ($E_{NS,SM,GE,5/95}$ = -6.8 − -3.1). Focusing on the model's skill to reproduce the discharge using these feasible parameter sets for the benchmark period, the parameter set associated with the best performing model with respect to river water levels ($E_{NS,SM,GE,opt}$) resulted in $E_{NS,Q,GE}$ = 0.65 ($E_{NS,Q,5/95,GE}$ = -0.48 − 0.60) and $D_{E,GE}$ = 0.77 ($D_{E,GE,5/95}$ = 0.28 − 0.70) with respect to discharge (Figure 7, Table 7). The model performance with respect to the remaining signatures as visualised in Figure 7 are tabulated in Table S4. As shown in Figure 8G, the discharge was overestimated in particular during intermediate and low flows.

**Water level Strategy 2: River geometry information obtained from a detailed field survey**

The parameter identification strategy "Water level Strategy 2", using cross-section information obtained from a detailed field survey, resulted in improved river water level simulations (compare Figure 10A and B) with an optimal objective function value with respect to river water levels of $E_{NS,SM,ADCP,opt}$ = 0.79 ($E_{NS,SM,ADCP,5/95}$ = 0.60 − 0.74). The parameter set associated with the best performing model with respect to river water levels ($E_{NS,SM,ADCP,opt}$) resulted in $E_{NS,Q,ADCP}$ = 0.14 ($E_{NS,Q,5/95,ADCP}$ = -1.1 − 0.50) and in $D_{E,ADCP}$ = 0.55 ($D_{E,ADCP,5/95}$ = 0.03 − 0.67) with respect to discharge (Figure 7, Table 7); the model performance with respect to the remaining signatures as visualised in Figure 7 are tabulated in Table S4.

Compared to using river geometry information extracted from Google Earth (Water level Strategy 1), the overall model performance with respect to discharge did not increase since the parameter space was already restricted using GRACE data. However, the modelled flow duration curve during intermediate and low flows (compare Figure 8G with H) and rating curve (Figure 11) improved significantly when using more accurate geometry information obtained from a detailed field survey covering the cross-section that is submerged most of the year

which is thus unlikely to be captured by satellite based observations. Note, that the in-situ cross-section information was limited to the submerged part during the time of measurement. The remaining part (water levels > 5 m) was extrapolated which is likely to explain the larger discrepancies during high flows visible in the flow duration curve (Figure 8H).

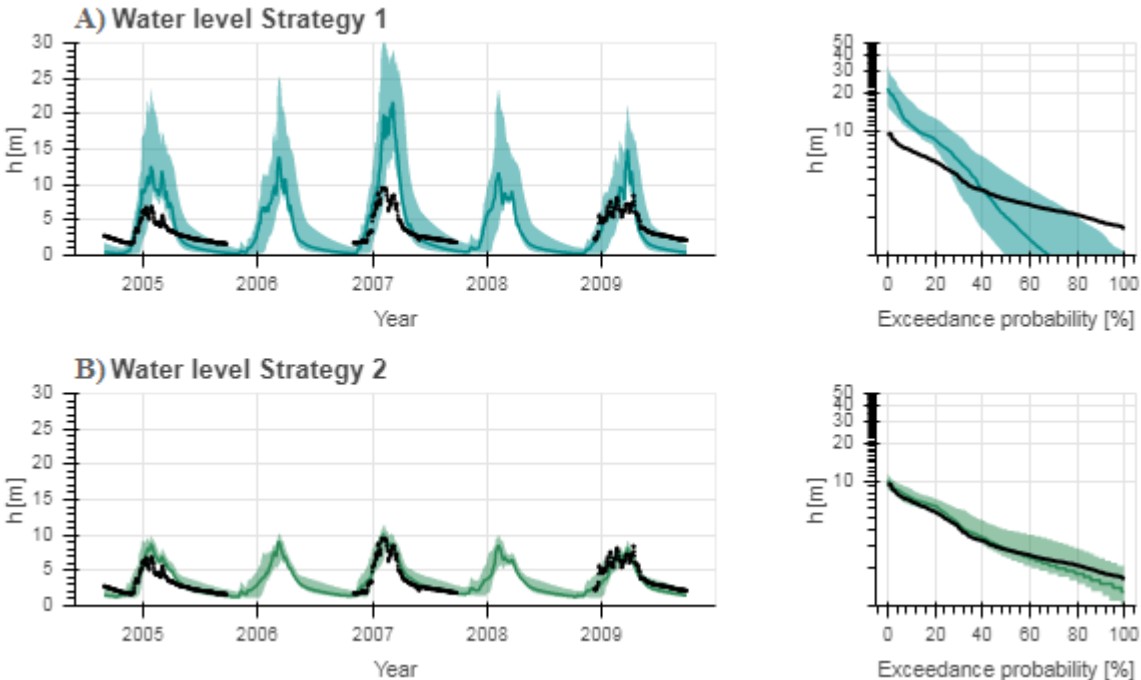

**Figure 10: Range of model solutions. The left panel shows the hydrograph and the right panel the flow duration curve of the recorded (black) and modelled discharge: the line indicates the solution with the highest calibration objective function ($E_{NS}$) and the shaded area the envelope of the solutions retained as feasible. Solutions were retained as feasible based on daily water level time series at the basin outlet using the Strickler-Manning equation for the**

**discharge – water level conversion; the cross-section was A) extracted from Google Earth (Water level Strategy 1), or B) obtained from a detailed field survey with an Acoustic Doppler Current Profiler (ADCP, Water level Strategy 2).**

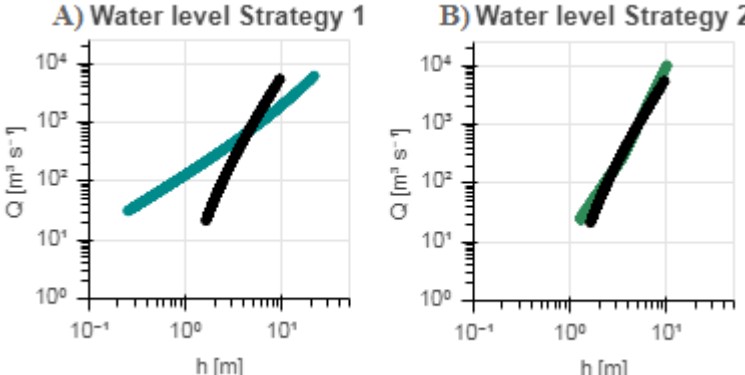

**Figure 11: Discharge - water level graphs for the recorded (black) and modelled discharge and stream levels with the optimal model performance ($E_{NS}$) using the Strickler Manning equation for the discharge – stream level conversion with cross-section information A) extracted from Google Earth (Water level Strategy 1), or B) obtained from a detailed field survey with an Acoustic Doppler Current Profiler (ADCP, Water level Strategy 2).**

### 4.2 Number of virtual stations used for model calibration and evaluation

In this study, altimetry data was available at 18 virtual stations. However, would the model performance change if more or less virtual stations were used? To answer this question, $n$ random stations were selected for model calibration; the remaining stations were used for cross-validation (Klemeš, 1986; Gharari et al., 2013; Garavaglia et al., 2017). This was repeated to cover all combinations of $n$ stations and for $n = 1, 2 \ldots 17$. When applying Strategy 3 using altimetry data with the Strickler-Manning equation, this analysis revealed that when increasing the number of calibration stations, the model calibration performance $D_{E,NS,SM}$ gradually decreased, but the ability to meaningfully reproduce the remaining observations which were not used for calibration increased significantly (Figure 12). Similar results were obtained for Strategies 1 and 2 (compare Figure 12 with Supplementary Figures S3 and S4). Also the model performance with respect to discharge increased when using more virtual stations with an optimum at 7 – 15 stations depending on the calibration strategy (Figure S5). This provides evidence that in spite of reduced calibration performance, the simultaneous use of multiple virtual stations can contribute towards more plausible selections of model parameter sets and thus increase the model realism.

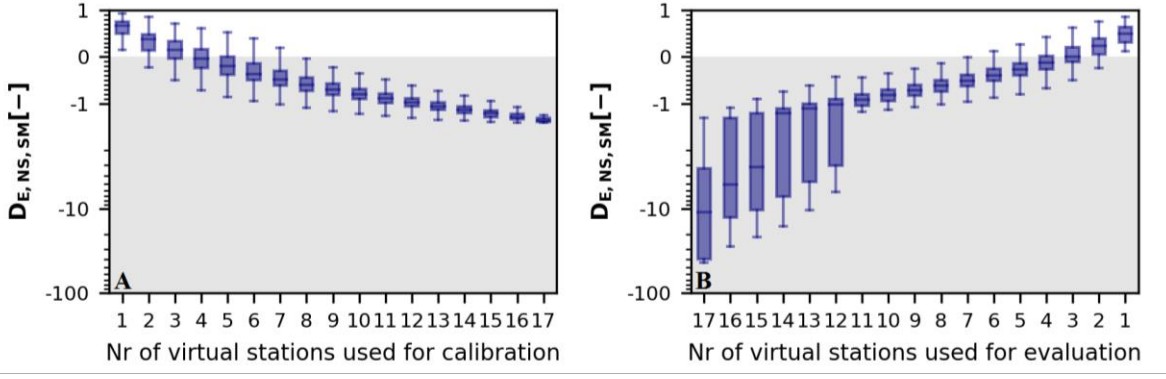

**Figure 12: Influence of the number of virtual stations used for A) model calibration and B) evaluation on the model performance $D_{E,NS,SM}$ applying Altimetry Strategy 3.**

## 4.3 Uncertainties and limitations

In the absence of discharge data for hydrological model calibration as commonly the case in poorly or ungauged regions, freely and globally available remotely sensed stream water levels could provide the opportunity to fill this gap as illustrated in this study, as well as in previous studies (e.g. Michailovsky and Bauer-Gottwein, 2014; Pereira-Cardenal et al., 2011; Sun et al., 2012). However, there are several limitations to the approach proposed in this study using altimetry for model calibration.

First, river altimetry data are prone to large uncertainties which increase for smaller river widths as a result of backscatter effects of the surrounding topography (Sulistioadi et al., 2015; Biancamaria et al., 2017; Domeneghetti et al., 2015). Too small rivers could even be missed altogether. In this study, the Luangwa river becomes a small meandering stream in the dry season resulting in larger altimetry uncertainties. Unfortunately, this uncertainty could not be estimated for the virtual stations used in this study due to data limitations. However, in previous studies in the Zambezi Basin, the RMSE relative to in-situ stream levels ranged between 0.32 m and 0.72 m using Envisat (Michailovsky et al., 2012). Improving altimetry observations such that the uncertainties decrease would improve the identification of feasible parameter sets and simulation of stream levels and flow. However, comparison results between the three altimetry based calibration strategies are not expected to change since the same altimetry data were used. In other words, Altimetry Strategy 3 is still expected to perform best when decreasing the uncertainties in the altimetry observations. Second, large uncertainties in the forcing data (precipitation and temperature) with respect to the spatial-temporal variations should not be ignored. This could compromise comparison results between modelled river water levels and altimetry within the basin since it has a low temporal resolution (10 or 35 days). Bias in the precipitation data affects storage calculations and hence the identification of feasible parameter sets based on GRACE (Le Coz and van de Giesen, 2019). This could explain why the flows were frequently overestimated when using GRACE only. In addition, precipitation bias could be compensated through calibration parameters introduced for the discharge – water level conversion. Therefore, such parameters should be constrained as much as possible. There are also data uncertainties in the cross-sections and river gradients extracted from high-resolution terrain data available on Google Earth due to its limited spatial resolution, but more importantly since no information is available below the water surface.

Further, GRACE observations are prone to uncertainties as a result of data (post-) processing including for example data smoothening (Landerer and Swenson, 2012; Blazquez et al., 2018; Riegger et al., 2012) causing leakage between neighbouring cells of 1° ($\approx$ 111 km) which are thus not completely independent from each other. Additionally, GRACE observations are more accurate for large areas. Depending on the applied processing scheme, the error is about 2 cm for basins with an area of around 63 000 km$^2$ (Landerer and Swenson, 2012; Vishwakarma et al., 2018). Also note that due to the coarse temporal resolution, monthly averaged GRACE observations are dominated by slow changing processes such as the groundwater and soil moisture system and seasonal variations reflected in all storage components. In addition, open water bodies or wetlands could affect GRACE observations if they are located in or near the basin, for example within a radius of about 300 km which is the distance often used for data smoothening. In this study, several open water bodies or wetlands were located ≤300 km of the Luangwa basin such as Lake Malawi, Kafue Flats, Cahora Bassa reservoir, Kariba reservoir, Bangweulu and Tanganyika. These open water bodies and wetlands had a limited impact on the GRACE observations due to limited fluctuations or different temporal variation as illustrated in Figure 13 for the Cahora Bassa reservoir. These uncertainties in the GRACE observations could influence the

identification of plausible parameter sets. For example feasible parameter sets could be discarded incorrectly which could distort results obtained by calibrating with respect to altimetry and GRACE simultaneously. However, the comparison between the three altimetry based calibration strategies is not expected to change since the same GRACE data were used. In other words, Altimetry Strategy 3 is still expected to perform best when considering these uncertainties.

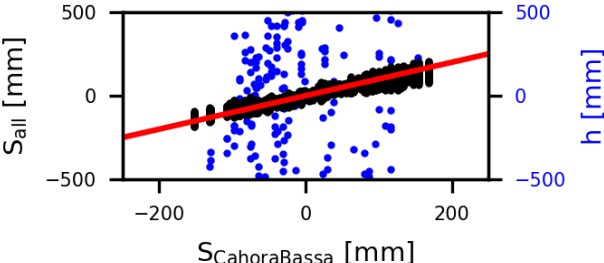

**Figure 13: Temporal correlation of the GRACE observations for the cell in which the virtual station for Cahora Bassa is located (horizontal axis) and for A) all cells within an area surrounding the virtual station with a radius of 3 degree (GRACE area of influence, vertical axis, black), and B) the altimetry observation at Cahora Bassa (vertical axis, blue). The 1:1 line is visualised in red. The relatively strong temporal correlation between the GRACE cells could be a result of the strong seasonality in this area.**

Uncertainties were not only introduced by the data, but also as a result of assumptions and simplifications. First, the reference level $h_0$ was assumed to be equal to the lowest river water level observed to limit the number of calibration parameters (Altimetry Strategy 2 and 3, Water level Strategy 1 and 2). However, uncertainties in the altimetry observations as explained previously influence $h_0$ estimates which results in a bias between the observed and simulated stream levels in Altimetry Strategies 2 and 3. Second, the roughness was assumed to be constant in time, over the entire cross-section and for all virtual stations throughout the basin \*-(Altimetry Strategy 3). However, this roughness can vary between $15 - 50$ m$^{1/3}$/s for natural rivers (Vatanchi and Maghrebi, 2019; Chow, 1959) changing the simulated stream levels between $42\% - 75\%$ in the Luangwa Basin with the low flows being the most sensitive. Third, all 18 virtual stations were grouped based on their cross-section similarity to limit the number of calibration parameters (Altimetry Strategy 2), but differences within each group remain such that the calibration parameters related to the rating curve varies slightly for each virtual station within a group. Fourth, the assumption of a constant flow velocity in space and time affects the timing of the simulated flow and stream levels influencing the comparison between model results and altimetry observations (all strategies).

Another limitation is the missing flow volume information when directly using (satellite based) river water levels for model calibration, using the Spearman Rank Correlations as model performance metric (Altimetry Strategy 1; Seibert and Vis, 2016). This resulted here in an overestimation of intermediate and low flows due to the non-linear relation between stream levels and flows. In contrast, when converting the discharge to stream water levels, flow volume information was included at the cost of introducing additional calibration parameters (Altimetry Strategy 2 and 3), thereby increasing the degrees-of-freedom and thus the potential for parameter equifinality in the model (Beven, 2006; Sikorska and Renard, 2017; Sun et al., 2012).

Furthermore, it was assumed the Nash-Sutcliffe efficiency contained sufficient valuable information to describe the model performance with respect to river water level and total water storage when identifying feasible parameter sets. This performance measure is sensitive to the sample size, outliers, bias and time-offset (McCuen

Richard et al., 2006). Unfortunately, simulated discharge and stream levels are prone to bias uncertainties as a result of spatiotemporal bias in the rainfall (Le Coz and van de Giesen, 2019). In addition, altimetry observations have a limited sample size for several virtual stations (see Table 2) and are prone to bias due to uncertainties in the reference level $h_0$ as mentioned before. Moreover, a time-offset in the simulated flow can occur as a result of rainfall uncertainties. As comparison, the model performance with respect to altimetry only reached up to $D_{E,NS,SM}$ = -1.3 for Altimetry Strategy 3, while it reached up to $E_{NS,SM,GE}$ = 0.61 with respect to daily in-situ stream levels for Water level Strategy 1. Therefore, additional study is recommended to confirm this assumption and to assess which performance metric(s) would be most suitable. The model performance with respect to discharge was evaluated with respect to multiple hydrological signatures simultaneously (see Table 6) to assess the model's skill to reproduce the internal dynamics of the system. Even though a few of these signatures have some overlapping information content (McMillan et al., 2017), each signature also contains at least some additional information not included in the other signatures. In general, the ambition is to represent a hydrological system as good as possible in a model which critically required that the model exhibits sufficient ability to simultaneously reproduce multiple flow signatures (Gupta et al., 2008; Euser et al., 2013; Hrachowitz et al., 2014).

### 4.4 Comparison with previous studies

Previous studies have successfully used river altimetry data to calibrate and evaluate rainfall-runoff models using a few virtual stations (Sun et al., 2012; Getirana, 2010; Getirana et al., 2010; Liu et al., 2015). In these studies, the modelled discharge was converted to stream levels by means of a hydraulic model or empirical relations. Our results support several previous findings and added a number of new ones.

Similar to previous studies, the rainfall-runoff model reproduced river flow relatively well when calibrating on remotely sensed stream water levels preferably at several virtual stations simultaneously, but discharge based calibration results performed significantly better (Getirana, 2010). Thus, while river altimetry data cannot fully substitute discharge observations, they at least provide an alternative data source that holds information value where no reliable discharge data are available. In addition, our results suggest that in spite of the typically limited temporal resolution of altimetry observations, these data, when using multiple virtual stations simultaneously, provide enough information to select meaningful model parameter sets (Seibert and Beven, 2009; Getirana, 2010).

Strikingly, only limited studies combined altimetry with GRACE observations in the calibration procedure (Kittel et al., 2018). As altimetry observations only describe water level variations with no information on the flow amounts, GRACE provides additional valuable information to constrain the river discharge by improving the rainfall runoff partitioning as demonstrated in previous studies (Rakovec et al., 2016; Bai et al., 2018; Dembélé et al., 2020). Combining both data sources in the calibration procedure allowed for a more accurate identification of feasible parameter sets. The model performance range with respect to discharge improved from $D_{E,5/95}$ = -8.4 − 0.77 when using only altimetry to $D_{E,5/95}$ = 0.19 − 0.75 when combining GRACE and altimetry for Altimetry Strategy 3 (see Figure S8).

In contrast to previous studies, altimetry data originated from five different satellite missions rather than a single one. As a result, altimetry data was available at 18 locations for the time period 2002 to 2016. This gave the opportunity to analyse the effect of combining different numbers of stations for calibration and evaluation. This

study illustrated that better predictions can be achieved when using more virtual stations for calibration. Furthermore, this study demonstrated that in particular the combination of altimetry with information on river geometry (cross section, gradient) proved beneficial for the selection of feasible parameter sets within relatively narrow bounds comparable to the benchmark using discharge. Using more accurate cross-section information obtained from a detailed field survey rather than Google Earth based estimates, improved the water level simulations, modelled rating curve and discharge simulations during intermediate and low flows significantly for which on-site cross-section data was available. That is why it is recommended to acquire accurate cross-section information on locations concurring with altimetry overpasses (not done is this study).

**4.5 Opportunities for future studies**

For future studies, it will be interesting to improve Altimetry Strategy 3 using additional data sources. For instance, the combination of altimetry observations with river width estimates derived from Landsat or Sentinel-1/2 (Pekel et al., 2016; Hou et al., 2018) may bear some potential as the combination of the two different hydraulic variables complements each other and increases the temporal sampling (Huang et al., 2018; Tarpanelli et al., 2017; Sichangi et al., 2016). For instance, during high flows river width estimates can be more accurate than altimetry observations especially when floodplains are inundated and small water level changes cause large river width changes. Alternatively, the altimetry observations used here could be combined with river surface water level slope estimates based on CryoSat observations which provide water level information at lower temporal resolution (every 369 days), but higher spatial resolution (equatorial inter-track distance of 7.5 km) (Schneider et al., 2017; Jiang et al., 2017). This allows for the estimation of the energy gradient based on stream levels as required in the Strickler-Manning equation, instead of the bed slope based on topography, which proved to be a good first estimate in absence of more reliable data. In addition, CryoSat observations are available annually such that there can be more overlap with altimetry observations in contrast to topography data. In addition, with the upcoming SWOT (Surface Water Ocean Topography) mission, more accurate altimetry observations should be available as well as river slope observations and width. The repeat cycle will be 21 days and across-track resolution between 10 m and 60 m increasing the number of observation points available within a specific area (Biancamaria et al., 2016; Langhorst et al., 2019; Oubanas et al., 2018). As a result, hydrological models can be calibrated with respect to river altimetry and width simultaneously at multiple locations even for small river basin improving the identification of plausible parameters sets and hence the model realism as illustrated in Section 4.2. It will also be very valuable to improve cross-section estimates with respect to the submerged part of the cross-section as already explored in previous studies (Domeneghetti, 2016) or to use drone observations to obtain more accurate cross-section information and estimates of the river slope and roughness (Entwistle and Heritage, 2019). By improving the river profile description, the simulated stream levels become more accurate which is crucial when using this time series for model calibration. As illustrated with Water level Strategies 1 and 2, improving the cross-section resulted in a more accurate rating curve (Figure 11), stream level simulation (see Figure 10), and discharge simulation (Figure 8). Clearly, it will be interesting to analyze and disentangle different individual sources of uncertainty related to the discharge – water level conversion from the hydrological model in a more data rich region (Renard et al., 2010). Unfortunately, this was not possible in this study due to the scarcely available in-situ observations in the Luangwa. As concluded by Renard et al. (2010),

reliable estimates of the data uncertainty are required to disaggregate multiple sources on uncertainty in rainfall-runoff modelling successfully.

**5 Summary and conclusion**

This study investigated the potential value of river altimetry observations from multiple satellite missions to identify feasible parameters for a hydrological model of the semi-arid and poorly gauged Luangwa River Basin. A distributed process-based rainfall-runoff model with sub-grid process heterogeneity was developed on a daily timescale for the time period 2002 to 2016. Various parameter identification strategies were implemented step-wise to assess the potential of satellite altimetry data for model calibration. As a benchmark, when identifying parameter sets with the traditional model calibration strategy using discharge data, the model was able to simulate the flows relatively well ($E_{NS,Q}$ = 0.78, $E_{NS,Q,5/95}$ = 0.61 – 0.75). When assuming no discharge observations are available, the feasible parameter sets were restricted with GRACE data only resulting in an optimum of $E_{NS,Q}$ = -1.4 ($E_{NS,Q,5/95}$ = -2.3 – 0.38) with respect to discharge. Combining GRACE with altimetry data only from 18 virtual stations focusing on the water level dynamics resulted in frequently overestimated flows and poorly identified feasible parameter sets (Altimetry Strategy 1, $E_{NS,Q,5/95}$ = -2.9 – 0.10). This was also the case when converting modelled discharge to water levels using rating curves (Altimetry Strategy 2, $E_{NS,Q,5/95}$ = -2.6 – 0.25). The identification of the feasible parameter sets improved when including river geometry information, more specifically cross-section and river gradient extracted from Google Earth, in the discharge-water level conversion using the Strickler-Manning equation (Altimetry Strategy 3, $E_{NS,Q}$ = 0.60, $E_{NS,Q,5/95}$ = -0.31 – 0.50). Moreover, it was shown that more accurate cross-section data improved the water level simulations, modelled rating curve and discharge simulations during intermediate and low flows for which on-site cross-section information was available; the Nash-Sutcliffe efficiency with respect to river water levels increased from $E_{NS,SM,GE}$ = -1.8 ($E_{NS,SM,GE,5/95}$ = -6.8 – -3.1) using river geometry information extracted from Google Earth (Water level Strategy 1) to $E_{NS,SM,ADCP}$ = 0.79 ($E_{NS,SM,ADCP,5/95}$ = 0.6 – 0.74) using river geometry information obtained from a detailed field survey (Water level Strategy 2). The model performance also improved when increasing the number of virtual stations used for parameter selection. Therefore, in the absence of reliable discharge data as commonly the case in poorly or ungauged basins, altimetry data from multiple virtual stations combined with GRACE observations have the potential to fill this gap if combined with river geometry estimates.

**Acknowledgement**

This research is supported by the TU Delft | Global Initiative, a program of the Delft University of Technology to boost Science and Technology for Global Development. This study would not have been possible without the help of those who provided us with the data. Local hydro-meteorological data was provided by WARMA (Water Resources Management Authority in Zambia), ZMD (Zambia Meteorological Department), GRDC (Global Runoff Data Centre) and NOAA (National Oceanic and Atmospheric Administration). Remotely sensed river water levels were obtained from DAHITI, HydroSat, EARPS and LEGOS.

**Literature**

Abas, I.: Remote river rating in Zambia: A case study in the Luangwa river basin, Master of Science, Civil Engineering and Geosciences, Delft University of Technology, 2018.

Ajami, N. K., Gupta, H., Wagener, T., and Sorooshian, S.: Calibration of a semi-distributed hydrologic model for streamflow estimation along a river system, Journal of Hydrology, 298, 112-135, 10.1016/j.jhydrol.2004.03.033, 2004.

Bai, P., Liu, X., and Liu, C.: Improving hydrological simulations by incorporating GRACE data for model calibration, Journal of Hydrology, 557, 291-304, https://doi.org/10.1016/j.jhydrol.2017.12.025, 2018.

Bauer-Gottwein, P., Jensen, I. H., Guzinski, R., Bredtoft, G. K. T., Hansen, S., and Michailovsky, C. I.: Operational river discharge forecasting in poorly gauged basins: The Kavango River basin case study, Hydrology and Earth System Sciences, 19, 1469-1485, 10.5194/hess-19-1469-2015, 2015.

Beilfuss, R., and dos Santos, D.: Patterns of Hydrological Change in the Zambezi Delta, Mozambique, in: Working Paper #2 Program for the Sustainable Management of Cahora Bassa Dam and the Lower Zambezi Valley, International Crane Foundation, Sofala, Mozambique, 2001.

Beven, K.: A manifesto for the equifinality thesis, Journal of Hydrology, 320, 18-36, https://doi.org/10.1016/j.jhydrol.2005.07.007, 2006.

Beven, K.: On doing better hydrological science, Hydrological Processes, 22, 3549-3553, 10.1002/hyp.7108, 2008.

Beven, K., and Westerberg, I.: On red herrings and real herrings: disinformation and information in hydrological inference, Hydrological Processes, 25, 1676-1680, 10.1002/hyp.7963, 2011.

Beven, K. J.: Preferential flows and travel time distributions: defining adequate hypothesis tests for hydrological
process models, Hydrological Processes, 24, 1537-1547, 10.1002/hyp.7718, 2010.

Biancamaria, S., Lettenmaier, D. P., and Pavelsky, T. M.: The SWOT Mission and Its Capabilities for Land Hydrology, Surveys in Geophysics, 37, 307-337, 10.1007/s10712-015-9346-y, 2016.

Biancamaria, S., Frappart, F., Leleu, A. S., Marieu, V., Blumstein, D., Desjonquères, J.-D., Boy, F., Sottolichio, A., and Valle-Levinson, A.: Satellite radar altimetry water elevations performance over a 200m wide river:
Evaluation over the Garonne River, Advances in Space Research, 59, 128-146, https://doi.org/10.1016/j.asr.2016.10.008, 2017.

Birkett, C. M.: Contribution of the TOPEX NASA Radar Altimeter to the global monitoring of large rivers and wetlands, Water Resources Research, 34, 1223-1239, 10.1029/98WR00124, 1998.

Blazquez, A., Meyssignac, B., Lemoine, J. M., Berthier, E., Ribes, A., and Cazenave, A.: Exploring the
uncertainty in GRACE estimates of the mass redistributions at the Earth surface: implications for the global water and sea level budgets, Geophysical Journal International, 215, 415-430, 10.1093/gji/ggy293, 2018.

Calmant, S., Seyler, F., and Cretaux, J.: Monitoring Continental Surface Waters by Satellite Altimetry, 247-269 pp., 2009.

Chow, V. T.: Open-channel hydraulics, McGraw-Hill, New York, 1959.

Clark, M. P., Nijssen, B., Lundquist, J. D., Kavetski, D., Rupp, D. E., Woods, R. A., Freer, J. E., Gutmann, E. D., Wood, A. W., Gochis, D. J., Rasmussen, R. M., Tarboton, D. G., Mahat, V., Flerchinger, G. N., and Marks, D. G.: A unified approach for process-based hydrologic modeling: 2. Model implementation and case studies, Water Resources Research, 51, 2515-2542, 10.1002/2015WR017200, 2015.

Clark, M. P., Schaefli, B., Schymanski, S. J., Samaniego, L., Luce, C. H., Jackson, B. M., Freer, J. E., Arnold, J.
R., Moore, R. D., Istanbulluoglu, E., and Ceola, S.: Improving the theoretical underpinnings of process-based hydrologic models, Water Resources Research, 52, 2350-2365, 10.1002/2015WR017910, 2016.

AVISO+ Satellite Altimetry Data: www.aviso.altimetry.fr, access: Jan 2018, Accessed 2018.

Danielson, J. J., and Gesch, D. B.: Global multi-resolution terrain elevation data 2010 (GMTED2010), Report 2011-1073, 2011.

de Oliveira Campos, I., Mercier, F., Maheu, C., Cochonneau, G., Kosuth, P., Blitzkow, D., and Cazenave, A.: Temporal variations of river basin waters from Topex/Poseidon satellite altimetry. Application to the Amazon basin, Comptes Rendus de l'Académie des Sciences - Series IIA - Earth and Planetary Science, 333, 633-643, http://dx.doi.org/10.1016/S1251-8050(01)01688-3, 2001.

Dembélé, M., Hrachowitz, M., Savenije, H. H. G., Mariéthoz, G., and Schaefli, B.: Improving the Predictive
Skill of a Distributed Hydrological Model by Calibration on Spatial Patterns With Multiple Satellite Data Sets, Water Resources Research, 56, e2019WR026085, 10.1029/2019WR026085, 2020.

Demirel, M., Mai, J., Mendiguren González, G., Koch, J., Samaniego, L., and Stisen, S.: Combining satellite data and appropriate objective functions for improved spatial pattern performance of a distributed hydrologic model, 2018.

Domeneghetti, A., Castellarin, A., Tarpanelli, A., and Moramarco, T.: Investigating the uncertainty of satellite altimetry products for hydrodynamic modelling, Hydrological Processes, 29, 4908-4918, 10.1002/hyp.10507, 2015.

Domeneghetti, A.: On the use of SRTM and altimetry data for flood modeling in data-sparse regions, Water Resources Research, 52, 2901-2918, 10.1002/2015WR017967, 2016.

Drusch, M., Del Bello, U., Carlier, S., Colin, O., Fernandez, V., Gascon, F., Hoersch, B., Isola, C., Laberinti, P., Martimort, P., Meygret, A., Spoto, F., Sy, O., Marchese, F., and Bargellini, P.: Sentinel-2: ESA's Optical High-Resolution Mission for GMES Operational Services, Remote Sensing of Environment, 120, 25-36, https://doi.org/10.1016/j.rse.2011.11.026, 2012.

Entwistle, N. S., and Heritage, G. L.: Small unmanned aerial model accuracy for photogrammetrical fluvial
bathymetric survey, Journal of Applied Remote Sensing, 13, 1-19, 19, 2019.

Satellite Missions Database: https://directory.eoportal.org/web/eoportal/satellite-missions, access: Jan 2018, 2018.

Euser, T., Winsemius, H. C., Hrachowitz, M., Fenicia, F., Uhlenbrook, S., and Savenije, H. H. G.: A framework to assess the realism of model structures using hydrological signatures, Hydrology and Earth System Sciences,
17, 1893-1912, 10.5194/hess-17-1893-2013, 2013.

Euser, T., Hrachowitz, M., Winsemius, H. C., and Savenije, H. H. G.: The effect of forcing and landscape distribution on performance and consistency of model structures, Hydrological Processes, 29, 3727-3743, 10.1002/hyp.10445, 2015.

Fang, K., Shen, C., Fisher, J. B., and Niu, J.: Improving Budyko curve-based estimates of long-term water
partitioning using hydrologic signatures from GRACE, Water Resources Research, 52, 5537-5554, 10.1002/2016WR018748, 2016.

Fleischmann, A., Siqueira, V., Paris, A., Collischonn, W., Paiva, R., Pontes, P., Crétaux, J. F., Bergé-Nguyen, M., Biancamaria, S., Gosset, M., Calmant, S., and Tanimoun, B.: Modelling hydrologic and hydrodynamic processes in basins with large semi-arid wetlands, Journal of Hydrology, 561, 943-959,
10.1016/j.jhydrol.2018.04.041, 2018.

Forootan, E., Khaki, M., Schumacher, M., Wulfmeyer, V., Mehrnegar, N., van Dijk, A. I. J. M., Brocca, L., Farzaneh, S., Akinluyi, F., Ramillien, G., Shum, C. K., Awange, J., and Mostafaie, A.: Understanding the global hydrological droughts of 2003–2016 and their relationships with teleconnections, Science of the Total Environment, 650, 2587-2604, 10.1016/j.scitotenv.2018.09.231, 2019.

Fovet, O., Ruiz, L., Hrachowitz, M., Faucheux, M., and Gascuel-Odoux, C.: Hydrological hysteresis and its value for assessing process consistency in catchment conceptual models, Hydrology and Earth System Sciences, 19, 105-123, 10.5194/hess-19-105-2015, 2015.

Frappart, F., Papa, F., Marieu, V., Malbeteau, Y., Jordy, F., Calmant, S., Durand, F., and Bala, S.: Preliminary Assessment of SARAL/AltiKa Observations over the Ganges-Brahmaputra and Irrawaddy Rivers, Marine
Geodesy, 38, 568-580, 10.1080/01490419.2014.990591, 2015.

Freer, J., Beven, K., and Ambroise, B.: Bayesian Estimation of Uncertainty in Runoff Prediction and the Value of Data: An Application of the GLUE Approach, Water Resources Research, 32, 2161-2173, 10.1029/95WR03723, 1996.

Funk, C. C., Peterson, P. J., Landsfeld, M. F., Pedreros, D. H., Verdin, J. P., Rowland, J. D., Romero, B. E.,
Husak, G. J., Michaelsen, J. C., and Verdin, A. P.: A quasi-global precipitation time series for drought monitoring:       U.S.       Geological       Survey,       Data       Series       832,       4,       ftp://chg-ftpout.geog.ucsb.edu/pub/org/chg/products/CHIRPS-2.0/docs/USGS-DS832.CHIRPS.pdf, 2014.

Gao, H., Hrachowitz, M., Fenicia, F., Gharari, S., and Savenije, H. H. G.: Testing the realism of a topography-driven model (FLEX-Topo) in the nested catchments of the Upper Heihe, China, Hydrol. Earth Syst. Sci., 18,
1895-1915, 10.5194/hess-18-1895-2014, 2014.

Gao, H., Hrachowitz, M., Sriwongsitanon, N., Fenicia, F., Gharari, S., and Savenije, H. H. G.: Accounting for the influence of vegetation and landscape improves model transferability in a tropical savannah region, Water Resources Research, 52, 7999-8022, 10.1002/2016WR019574, 2016.

Garambois, P.-A., Calmant, S., Roux, H., Paris, A., Monnier, J., Finaud-Guyot, P., Samine Montazem, A., and
Santos da Silva, J.: Hydraulic visibility: Using satellite altimetry to parameterize a hydraulic model of an ungauged reach of a braided river, Hydrological Processes, 31, 756-767, 10.1002/hyp.11033, 2017.

Garavaglia, F., Le Lay, M., Gottardi, F., Garçon, R., Gailhard, J., Paquet, E., and Mathevet, T.: Impact of model structure on flow simulation and hydrological realism: from a lumped to a semi-distributed approach, Hydrol. Earth Syst. Sci., 21, 3937-3952, 10.5194/hess-21-3937-2017, 2017.

Getirana, A. C. V., Bonnet, M.-P., Calmant, S., Roux, E., Rotunno Filho, O. C., and Mansur, W. J.: Hydrological monitoring of poorly gauged basins based on rainfall–runoff modeling and spatial altimetry, Journal of Hydrology, 379, 205-219, http://dx.doi.org/10.1016/j.jhydrol.2009.09.049, 2009.

Getirana, A. C. V.: Integrating spatial altimetry data into the automatic calibration of hydrological models, Journal of Hydrology, 387, 244-255, http://dx.doi.org/10.1016/j.jhydrol.2010.04.013, 2010.

Getirana, A. C. V., Bonnet, M. P., Rotunno Filho, O. C., Collischonn, W., Guyot, J. L., Seyler, F., and Mansur, W. J.: Hydrological modelling and water balance of the Negro River basin: evaluation based on in situ and spatial altimetry data, Hydrological Processes, 24, 3219-3236, 10.1002/hyp.7747, 2010.

Getirana, A. C. V., and Peters-Lidard, C.: Estimating water discharge from large radar altimetry datasets, Hydrol. Earth Syst. Sci., 17, 923-933, 10.5194/hess-17-923-2013, 2013.

Gharari, S., Hrachowitz, M., Fenicia, F., and Savenije, H. H. G.: Hydrological landscape classification: investigating the performance of HAND based landscape classifications ina central European meso-scale catchment, Hydrol. Earth Syst. Sci., 15, 3275-3291, 2011.

Gharari, S., Hrachowitz, M., Fenicia, F., and Savenije, H. H. G.: An approach to identify time consistent model parameters: sub-period calibration, Hydrol. Earth Syst. Sci., 17, 149-161, 10.5194/hess-17-149-2013, 2013.

Gharari, S., Hrachowitz, M., Fenicia, F., Gao, H., and Savenije, H. H. G.: Using expert knowledge to increase realism in environmental system models can dramatically reduce the need for calibration, Hydrol. Earth Syst. Sci., 18, 4839-4859, 10.5194/hess-18-4839-2014, 2014.

Gichamo, T. Z., Popescu, I., Jonoski, A., and Solomatine, D.: River cross-section extraction from the ASTER global DEM for flood modeling, Environmental Modelling & Software, 31, 37-46, https://doi.org/10.1016/j.envsoft.2011.12.003, 2012.

GlobCover, 2009.

Google Earth, 2018.

Gupta, H. V., Wagener, T., and Liu, Y.: Reconciling theory with observations: elements of a diagnostic approach to model evaluation, Hydrological Processes, 22, 3802-3813, 10.1002/hyp.6989, 2008.

Floods displace thousands in Mozambique: https://www.theguardian.com/world/2001/mar/28/mozambique.unitednations, access: Jan 2017, 2001.

Hargreaves, G. H., and Samani, Z. A.: Reference Crop Evapotranspiration from Temperature, Applied Engineering in Agriculture, 1, 96-99, https://doi.org/10.13031/2013.26773, 1985.

Hargreaves, G. H., and Allen, R. G.: History and evaluation of hargreaves evapotranspiration equation, Journal of Irrigation and Drainage Engineering, 129, 53-63, 10.1061/(ASCE)0733-9437(2003)129:1(53), 2003.

Hasan, M. A., and Pradhanang, S. M.: Estimation of flow regime for a spatially varied Himalayan watershed using improved multi-site calibration of the Soil and Water Assessment Tool (SWAT) model, Environmental Earth Sciences, 76, 787, 10.1007/s12665-017-7134-3, 2017.

Hou, J., van Dijk, A. I. J. M., Renzullo, L. J., and Vertessy, R. A.: Using modelled discharge to develop satellite-based river gauging: a case study for the Amazon Basin, Hydrol. Earth Syst. Sci., 22, 6435-6448, 10.5194/hess-22-6435-2018, 2018.

Hrachowitz, M., Savenije, H. H. G., Blöschl, G., McDonnell, J. J., Sivapalan, M., Pomeroy, J. W., Arheimer, B., Blume, T., Clark, M. P., Ehret, U., Fenicia, F., Freer, J. E., Gelfan, A., Gupta, H. V., Hughes, D. A., Hut, R. W., Montanari, A., Pande, S., Tetzlaff, D., Troch, P. A., Uhlenbrook, S., Wagener, T., Winsemius, H. C., Woods, R. A., Zehe, E., and Cudennec, C.: A decade of Predictions in Ungauged Basins (PUB)—a review, Hydrological Sciences Journal, 58, 1198-1255, 10.1080/02626667.2013.803183, 2013.

Hrachowitz, M., Fovet, O., Ruiz, L., Euser, T., Gharari, S., Nijzink, R., Freer, J., Savenije, H. H. G., and Gascuel-Odoux, C.: Process consistency in models: The importance of system signatures, expert knowledge, and process complexity, Water Resources Research, 50, 7445-7469, 10.1002/2014WR015484, 2014.

Hrachowitz, M., and Clark, M. P.: HESS Opinions: The complementary merits of competing modelling philosophies in hydrology, Hydrol. Earth Syst. Sci., 21, 3953-3973, 10.5194/hess-21-3953-2017, 2017.

Huang, Q., Long, D., Du, M., Zeng, C., Qiao, G., Li, X., Hou, A., and Hong, Y.: Discharge estimation in high-mountain regions with improved methods using multisource remote sensing: A case study of the Upper Brahmaputra River, Remote Sensing of Environment, 219, 115-134, https://doi.org/10.1016/j.rse.2018.10.008, 2018.

Hulsman, P., Bogaard, T. A., and Savenije, H. H. G.: Rainfall-runoff modelling using river-stage time series in the absence of reliable discharge information: a case study in the semi-arid Mara River basin, Hydrol. Earth Syst. Sci., 22, 5081-5095, 10.5194/hess-22-5081-2018, 2018.

Irons, J. R., Dwyer, J. L., and Barsi, J. A.: The next Landsat satellite: The Landsat Data Continuity Mission, Remote Sensing of Environment, 122, 11-21, https://doi.org/10.1016/j.rse.2011.08.026, 2012.

Jakeman, A. J., and Hornberger, G. M.: How much complexity is warranted in a rainfall-runoff model?, Water Resources Research, 29, 2637-2649, 10.1029/93WR00877, 1993.

Jian, J., Ryu, D., Costelloe, J. F., and Su, C.-H.: Towards hydrological model calibration using river level measurements, Journal of Hydrology: Regional Studies, 10, 95-109, https://doi.org/10.1016/j.ejrh.2016.12.085, 2017.

Jiang, L., Schneider, R., Andersen, O. B., and Bauer-Gottwein, P.: CryoSat-2 altimetry applications over rivers and lakes, Water (Switzerland), 9, 10.3390/w9030211, 2017.

Khaki, M., and Awange, J.: The application of multi-mission satellite data assimilation for studying water storage changes over South America, Science of the Total Environment, 647, 1557-1572, 10.1016/j.scitotenv.2018.08.079, 2019.

Kittel, C. M. M., Nielsen, K., Tøttrup, C., and Bauer-Gottwein, P.: Informing a hydrological model of the Ogooué with multi-mission remote sensing data, Hydrol. Earth Syst. Sci., 22, 1453-1472, 10.5194/hess-22-1453-2018, 2018.

KlemeŠ, V.: Operational testing of hydrological simulation models, Hydrological Sciences Journal, 31, 13-24, 10.1080/02626668609491024, 1986.

Knutti, R.: Should we believe model predictions of future climate change?, Philosophical Transactions of the Royal Society A: Mathematical, Physical and Engineering Sciences, 366, 4647-4664, 10.1098/rsta.2008.0169, 2008.

Kouraev, A. V., Zakharova, E. A., Samain, O., Mognard, N. M., and Cazenave, A.: Ob' river discharge from TOPEX/Poseidon satellite altimetry (1992-2002), Remote Sensing of Environment, 93, 238-245, 10.1016/j.rse.2004.07.007, 2004.

Lakshmi, V.: The role of satellite remote sensing in the Prediction of Ungauged Basins, Hydrological Processes, 18, 1029-1034, 10.1002/hyp.5520, 2004.

Landerer, F. W., and Swenson, S. C.: Accuracy of scaled GRACE terrestrial water storage estimates, Water Resources Research, 48, 11, doi:10.1029/2011WR011453, 2012.

Langhorst, T., Pavelsky, T. M., Frasson, R. P. d. M., Wei, R., Domeneghetti, A., Altenau, E. H., Durand, M. T., Minear, J. T., Wegmann, K. W., and Fuller, M. R.: Anticipated Improvements to River Surface Elevation Profiles From the Surface Water and Ocean Topography Mission, Frontiers in Earth Science, 7, 102, 2019.

Le Coz, C., and van de Giesen, N.: Comparison of rainfall products over sub-Sahara Africa, Journal of Hydrometeorology, 10.1175/JHM-D-18-0256.1, 2019.

Leon, J. G., Calmant, S., Seyler, F., Bonnet, M. P., Cauhopé, M., Frappart, F., Filizola, N., and Fraizy, P.: Rating curves and estimation of average water depth at the upper Negro River based on satellite altimeter data and modeled discharges, Journal of Hydrology, 328, 481-496, 10.1016/j.jhydrol.2005.12.006, 2006.

Liu, G., Schwartz, F. W., Tseng, K. H., and Shum, C. K.: Discharge and water-depth estimates for ungauged rivers: Combining hydrologic, hydraulic, and inverse modeling with stage and water-area measurements from satellites, Water Resources Research, 51, 6017-6035, 10.1002/2015WR016971, 2015.

Łyszkowicz, A. B., and Bernatowicz, A.: Current state of art of satellite altimetry, Geodesy and Cartography, 66, 259-270, https://doi.org/10.1515/geocart-2017-0016, 2017.

Manning, R.: On the flow of water in open channels and pipes, Transactions of the Institution of Civil Engineers of Ireland, 20, 161-207, 1891.

McCuen Richard, H., Knight, Z., and Cutter, A. G.: Evaluation of the Nash–Sutcliffe Efficiency Index, Journal of Hydrologic Engineering, 11, 597-602, 10.1061/(ASCE)1084-0699(2006)11:6(597), 2006.

McMillan, H., Westerberg, I., and Branger, F.: Five guidelines for selecting hydrological signatures, Hydrological Processes, 31, 4757-4761, 10.1002/hyp.11300, 2017.

McMillan, H. K., and Westerberg, I. K.: Rating curve estimation under epistemic uncertainty, Hydrological Processes, 29, 1873-1882, 10.1002/hyp.10419, 2015.

Michailovsky, C. I., McEnnis, S., Berry, P. A. M., Smith, R., and Bauer-Gottwein, P.: River monitoring from satellite radar altimetry in the Zambezi River basin, Hydrol. Earth Syst. Sci., 16, 2181-2192, 10.5194/hess-16-2181-2012, 2012.

Michailovsky, C. I., Milzow, C., and Bauer-Gottwein, P.: Assimilation of radar altimetry to a routing model of the Brahmaputra River, Water Resources Research, 49, 4807-4816, 10.1002/wrcr.20345, 2013.

Michailovsky, C. I., and Bauer-Gottwein, P.: Operational reservoir inflow forecasting with radar altimetry: the Zambezi case study, Hydrol. Earth Syst. Sci., 18, 997-1007, 10.5194/hess-18-997-2014, 2014.

Montanari, M., Hostache, R., Matgen, P., Schumann, G., Pfister, L., and Hoffmann, L.: Calibration and
sequential updating of a coupled hydrologic-hydraulic model using remote sensing-derived water stages, Hydrol. Earth Syst. Sci., 13, 367-380, 10.5194/hess-13-367-2009, 2009.

Nash, J. E., and Sutcliffe, J. V.: River flow forecasting through conceptual models part I — A discussion of principles, Journal of Hydrology, 10, 282-290, https://doi.org/10.1016/0022-1694(70)90255-6, 1970.

Nijzink, R. C., Samaniego, L., Mai, J., Kumar, R., Thober, S., Zink, M., Schäfer, D., Savenije, H. H. G., and
Hrachowitz, M.: The importance of topography-controlled sub-grid process heterogeneity and semi-quantitative prior constraints in distributed hydrological models, Hydrol. Earth Syst. Sci., 20, 1151-1176, 10.5194/hess-20-1151-2016, 2016.

Nijzink, R. C., Almeida, S., Pechlivanidis, I. G., Capell, R., Gustafssons, D., Arheimer, B., Parajka, J., Freer, J., Han, D., Wagener, T., van Nooijen, R. R. P., Savenije, H. H. G., and Hrachowitz, M.: Constraining Conceptual
Hydrological Models With Multiple Information Sources, Water Resources Research, 54, 8332-8362, 10.1029/2017WR021895, 2018.

Oubanas, H., Gejadze, I., Malaterre, P. O., Durand, M., Wei, R., Frasson, R. P. M., and Domeneghetti, A.: Discharge Estimation in Ungauged Basins Through Variational Data Assimilation: The Potential of the SWOT Mission, Water Resources Research, 54, 2405-2423, 10.1002/2017WR021735, 2018.

Pandya, U., Patel, A., and Patel, D.: RIVER CROSS SECTION DELINEATION FROM THE GOOGLE EARTH FOR DEVELOPMENT OF 1D HEC-RAS MODEL-A CASE OF SABARMATI RIVER, GUJARAT, INDIA, International Conference on Hydraulics, Water Resources & Coastal Engineering, Ahmedabad, India, 2017.

Papa, F., Bala, S. K., Pandey, R. K., Durand, F., Gopalakrishna, V. V., Rahman, A., and Rossow, W. B.: Ganga-
Brahmaputra river discharge from Jason-2 radar altimetry: An update to the long-term satellite-derived estimates of continental freshwater forcing flux into the Bay of Bengal, Journal of Geophysical Research: Oceans, 117, 10.1029/2012JC008158, 2012.

Paris, A., Dias de Paiva, R., Santos da Silva, J., Medeiros Moreira, D., Calmant, S., Garambois, P. A., Collischonn, W., Bonnet, M. P., and Seyler, F.: Stage-discharge rating curves based on satellite altimetry and
modeled discharge in the Amazon basin, Water Resources Research, 52, 3787-3814, 10.1002/2014WR016618, 2016.

Pechlivanidis, I. G., and Arheimer, B.: Large-scale hydrological modelling by using modified PUB recommendations: The India-HYPE case, Hydrology and Earth System Sciences, 19, 4559-4579, 10.5194/hess-19-4559-2015, 2015.

Pedinotti, V., Boone, A., Decharme, B., Crétaux, J. F., Mognard, N., Panthou, G., Papa, F., and Tanimoun, B. A.: Evaluation of the ISBA-TRIP continental hydrologic system over the Niger basin using in situ and satellite derived datasets, Hydrology and Earth System Sciences, 16, 1745-1773, 10.5194/hess-16-1745-2012, 2012.

Pekel, J.-F., Cottam, A., Gorelick, N., and Belward, A. S.: High-resolution mapping of global surface water and its long-term changes, Nature, 540, 418-422, 10.1038/nature20584, 2016.

Pereira-Cardenal, S. J., Riegels, N. D., Berry, P. A. M., Smith, R. G., Yakovlev, A., Siegfried, T. U., and Bauer-Gottwein, P.: Real-time remote sensing driven river basin modeling using radar altimetry, Hydrol. Earth Syst. Sci., 15, 241-254, 10.5194/hess-15-241-2011, 2011.

Pramanik, N., Panda, R. K., and Sen, D.: One Dimensional Hydrodynamic Modeling of River Flow Using DEM Extracted River Cross-sections, Water Resour Manage, 24, 835-852, 10.1007/s11269-009-9474-6, 2010.

Prenner, D., Kaitna, R., Mostbauer, K., and Hrachowitz, M.: The Value of Using Multiple Hydrometeorological Variables to Predict Temporal Debris Flow Susceptibility in an Alpine Environment, Water Resources Research, 54, 6822-6843, 10.1029/2018WR022985, 2018.

Rakovec, O., Kumar, R., Attinger, S., and Samaniego, L.: Improving the realism of hydrologic model functioning through multivariate parameter estimation, Water Resources Research, 52, 7779-7792, 10.1002/2016WR019430, 2016.

Rantz, S. E.: Measurement and computation of streamflow: Volume 2, Computation of Discharge, Report 2175, 1982.

Renard, B., Kavetski, D., Kuczera, G., Thyer, M., and Franks, S. W.: Understanding predictive uncertainty in hydrologic modeling: The challenge of identifying input and structural errors, Water Resources Research, 46, 10.1029/2009WR008328, 2010.

Rennó, C. D., Nobre, A. D., Cuartas, L. A., Soares, J. V., Hodnett, M. G., Tomasella, J., and Waterloo, M. J.: HAND, a new terrain descriptor using SRTM-DEM: Mapping terra-firme rainforest environments in Amazonia, Remote Sensing of Environment, 112, 3469-3481, https://doi.org/10.1016/j.rse.2008.03.018, 2008.

Revilla-Romero, B., Beck, H. E., Burek, P., Salamon, P., de Roo, A., and Thielen, J.: Filling the gaps: Calibrating a rainfall-runoff model using satellite-derived surface water extent, Remote Sensing of Environment, 171, 118-131, http://dx.doi.org/10.1016/j.rse.2015.10.022, 2015.

Riegger, J., Tourian, M. J., Devaraju, B., and Sneeuw, N.: Analysis of grace uncertainties by hydrological and hydro-meteorological observations, Journal of Geodynamics, 59-60, 16-27, https://doi.org/10.1016/j.jog.2012.02.001, 2012.

SADC: Integrated Water Resources Management Strategy and Implementation Plan for the Zambezi River Basin, Euroconsult Mott MacDonald, 2008.

Santhi, C., Kannan, N., Arnold, J. G., and Di Luzio, M.: Spatial Calibration and Temporal Validation of Flow for Regional Scale Hydrologic Modeling, JAWRA Journal of the American Water Resources Association, 44, 829-846, 10.1111/j.1752-1688.2008.00207.x, 2008.

Savenije, H. H. G.: Equifinality, a blessing in disguise?, Hydrological Processes, 15, 2835-2838, 10.1002/hyp.494, 2001.

Savenije, H. H. G.: Topography driven conceptual modelling (FLEX-Topo), Hydrol. Earth Syst. Sci., 14, 2681-2692, 2010.

Sawicz, K., Wagener, T., Sivapalan, M., Troch, P. A., and Carrillo, G.: Catchment classification: Empirical analysis of hydrologic similarity based on catchment function in the eastern USA, Hydrology and Earth System Sciences, 15, 2895-2911, 10.5194/hess-15-2895-2011, 2011.

Schleiss, A. J., and Matos, J. P.: Chapter 98: Zambezi River Basin, in: Chow's Handbook of Applied Hydrology, edited by: Singh, V. P., McGraw-Hill Education - Europe, United States, 2016.

Schneider, R., Godiksen, P. N., Villadsen, H., Madsen, H., and Bauer-Gottwein, P.: Application of CryoSat-2 altimetry data for river analysis and modelling, Hydrol. Earth Syst. Sci., 21, 751-764, 10.5194/hess-21-751-2017, 2017.

Schoups, G., Lee Addams, C., and Gorelick, S. M.: Multi-objective calibration of a surface water-groundwater flow model in an irrigated agricultural region: Yaqui Valley, Sonora, Mexico, Hydrol. Earth Syst. Sci., 9, 549-568, 10.5194/hess-9-549-2005, 2005.

Schumann, G., Kirschbaum, D., Anderson, E., and Rashid, K.: Role of Earth Observation Data in Disaster Response and Recovery: From Science to Capacity Building, in: Earth Science Satellite Applications edited by: Hossain, F., Springer International Publishing, Seattle, USA, 2016.

Schwatke, C., Dettmering, D., Bosch, W., and Seitz, F.: DAHITI – an innovative approach for estimating water level time series over inland waters using multi-mission satellite altimetry, Hydrol. Earth Syst. Sci., 19, 4345-
4364, 10.5194/hess-19-4345-2015, 2015.

Seibert, J., and Beven, K. J.: Gauging the ungauged basin: how many discharge measurements are needed?, Hydrol. Earth Syst. Sci., 13, 883-892, 10.5194/hess-13-883-2009, 2009.

Seibert, J., and Vis, M. J. P.: How informative are stream level observations in different geographic regions?, Hydrological Processes, 30, 2498-2508, 10.1002/hyp.10887, 2016.

Seyler, F., Calmant, S., Silva, J. S. d., Moreira, D. M., Mercier, F., and Shum, C. K.: From TOPEX/Poseidon to Jason-2/OSTM in the Amazon basin, Advances in Space Research, 51, 1542-1550, https://doi.org/10.1016/j.asr.2012.11.002, 2013.

Sichangi, A. W., Wang, L., Yang, K., Chen, D., Wang, Z., Li, X., Zhou, J., Liu, W., and Kuria, D.: Estimating continental river basin discharges using multiple remote sensing data sets, Remote Sensing of Environment, 179,
36-53, https://doi.org/10.1016/j.rse.2016.03.019, 2016.

Sikorska, A. E., and Renard, B.: Calibrating a hydrological model in stage space to account for rating curve uncertainties: general framework and key challenges, Advances in Water Resources, 105, 51-66, https://doi.org/10.1016/j.advwatres.2017.04.011, 2017.

Smith, B., and Sandwell, D.: Accuracy and resolution of shuttle radar topography mission data, Geophysical
Research Letters, 30, 10.1029/2002GL016643, 2003.

Spearman, C.: The proof and measurement of association between two things, The American Journal of Psychology, 15, 72-101, 1904.

Sulistioadi, Y. B., Tseng, K. H., Shum, C. K., Hidayat, H., Sumaryono, M., Suhardiman, A., Setiawan, F., and Sunarso, S.: Satellite radar altimetry for monitoring small rivers and lakes in Indonesia, Hydrol. Earth Syst. Sci.,
19, 341-359, 10.5194/hess-19-341-2015, 2015.

Sun, W., Ishidaira, H., and Bastola, S.: Calibration of hydrological models in ungauged basins based on satellite radar altimetry observations of river water level, Hydrological Processes, 26, 3524-3537, 10.1002/hyp.8429, 2012.

Sun, W., Ishidaira, H., Bastola, S., and Yu, J.: Estimating daily time series of streamflow using hydrological
model calibrated based on satellite observations of river water surface width: Toward real world applications, Environmental Research, 139, 36-45, http://dx.doi.org/10.1016/j.envres.2015.01.002, 2015.

Sun, W., Fan, J., Wang, G., Ishidaira, H., Bastola, S., Yu, J., Fu, Y. H., Kiem, A. S., Zuo, D., and Xu, Z.: Calibrating a hydrological model in a regional river of the Qinghai–Tibet plateau using river water width determined from high spatial resolution satellite images, Remote Sensing of Environment, 214, 100-114,
https://doi.org/10.1016/j.rse.2018.05.020, 2018.

Swenson, S. C., and Wahr, J.: Post-processing removal of correlated errors in GRACE data, Geophys. Res. Lett., 33, doi:10.1029/2005GL025285, 2006.

Swenson, S. C.: GRACE monthly land water mass grids NETCDF RELEASE 5.0, in, PO.DAAC, CA, USA, 2012.

Tang, Y., Hooshyar, M., Zhu, T., Ringler, C., Sun, A. Y., Long, D., and Wang, D.: Reconstructing annual groundwater storage changes in a large-scale irrigation region using GRACE data and Budyko model, Journal of Hydrology, 551, 397-406, https://doi.org/10.1016/j.jhydrol.2017.06.021, 2017.

Tarpanelli, A., Barbetta, S., Brocca, L., and Moramarco, T.: River discharge estimation by using altimetry data and simplified flood routing modeling, Remote Sensing, 5, 4145-4162, 10.3390/rs5094145, 2013.

Tarpanelli, A., Amarnath, G., Brocca, L., Massari, C., and Moramarco, T.: Discharge estimation and forecasting by MODIS and altimetry data in Niger-Benue River, Remote Sensing of Environment, 195, 96-106, 10.1016/j.rse.2017.04.015, 2017.

The World Bank: The Zambezi River Basin: A Multi-Sector Investment Opportunities Analysis, in, 2010.

Tourian, M. J., Sneeuw, N., and Bárdossy, A.: A quantile function approach to discharge estimation from
satellite altimetry (ENVISAT), Water Resources Research, 49, 2013.

Tourian, M. J., Tarpanelli, A., Elmi, O., Qin, T., Brocca, L., Moramarco, T., and Sneeuw, N.: Spatiotemporal densification of river water level time series by multimission satellite altimetry, Water Resources Research, 52, 1140-1159, 10.1002/2015WR017654, 2016.

Tourian, M. J., Schwatke, C., and Sneeuw, N.: River discharge estimation at daily resolution from satellite
altimetry over an entire river basin, Journal of Hydrology, 546, 230-247, https://doi.org/10.1016/j.jhydrol.2017.01.009, 2017.

University of East Anglia Climatic Research Unit, Harris, I. C., and Jones, P. D.: CRU TS4.01: Climatic Research Unit (CRU) Time-Series (TS) version 4.01 of high-resolution gridded data of month-by-month variation in climate (Jan. 1901- Dec. 2016), Centre for Environmental Data Analysis,,
doi:10.5285/58a8802721c94c66ae45c3baa4d814d0, 2017.

Vatanchi, S. M., and Maghrebi, M. F.: Uncertainty in Rating-Curves Due to Manning Roughness Coefficient, Water Resour Manage, 33, 5153-5167, 10.1007/s11269-019-02421-6, 2019.

Velpuri, N. M., Senay, G. B., and Asante, K. O.: A multi-source satellite data approach for modelling Lake Turkana water level: calibration and validation using satellite altimetry data, Hydrol. Earth Syst. Sci., 16, 1-18,
10.5194/hess-16-1-2012, 2012.

Vishwakarma, D. B., Devaraju, B., and Sneeuw, N.: What Is the Spatial Resolution of grace Satellite Products for Hydrology?, Remote Sensing, 10, 10.3390/rs10060852, 2018.

Wahr, J., Molenaar, M., and Bryan, F.: Time variability of the Earth's gravity field: Hydrological and oceanic effects and their possible detection using GRACE, Journal of Geophysical Research: Solid Earth, 103, 30205-
30229, 10.1029/98JB02844, 1998.

Winsemius, H. C., Savenije, H. H. G., and Bastiaanssen, W. G. M.: Constraining model parameters on remotely sensed evaporation: justification for distribution in ungauged basins?, Hydrol. Earth Syst. Sci., 12, 1403-1413, 10.5194/hess-12-1403-2008, 2008.

ZAMCOM, SADC, and SARDC: Zambezi Environment Outlook 2015, Harare, Gaborone, 2015.

Zhou, X., and Wang, H.: Application of Google Earth in Modern River Sedimentology Research, 1-8 pp., 2015.

Zink, M., Mai, J., Cuntz, M., and Samaniego, L.: Conditioning a Hydrologic Model Using Patterns of Remotely Sensed Land Surface Temperature, Water Resources Research, 54, 2976-2998, 10.1002/2017WR021346, 2018.