# Peer review of "Using altimetry observations combined with GRACE to select parameter sets of a hydrological model in data scarce regions"

_Hydrology and Earth System Sciences, 2019_

## Referee Comment (RC1) · Anonymous Referee #1 · 7 Oct 2019

This was an interesting m/s to read, and I believe this study has some interesting insights worthy of publication. However I do believe it needs some improvements. More detail below.

Comment 1) I got lost in some of the technical detail around the various alternative calibration strategies tested for calibration. As the abstract seems to suggest insights relating to calibration strategy are the main contribution of this m/s, so I think this needs some more attention. For example, the research hypotheses (l. 109-11) do not address this aspect. In the introduction, can you provide some discussion around the rationale for the different experiments? In fact, to support that, it would be helpful if the authors

[Figure]

could provide a table listing, for each variant, the objective function, any transformation of model or observation data (i.e. the observation model), the potential benefits of the variant (i.e., why was it tested), and the empirically-found pros and cons.

Comment 2) Please consult Domeneghetti (2016) and Oubanas et al. (2018) and consider whether they may be relevant to your discussion.

Comment 3) With the caveat that I did not understand all details, I seem to gather that one of the main conclusions of this m/s is that selecting parameters based on rank correlation between discharge and altimetry water level is not sufficient to constrain model parameters, and that altimetry levels need to be converted to actual discharge to provide an appropriate constraint. Is that correct? If so, then that would be expected when evaluating against a performance measure that is extremely bias-sensitive, like Nash-Sutcliffe efficiency (NSE). However, while I know NSE is religiously adhered to by some hydrologists, it is not a relevant performance indicator for all possible uses of river discharge modelling (and indeed many hydrologists have already found a new religion in the more information-rich components of Kling-Gupta Efficiency, KGE). For many practical applications, a high correlation may well be more important than a bias-free estimate, for example in flood and drought applications. Even if volumetric accuracy is more important (e.g. in water resources volume management) then, in this case, you have some gauged data, so provided correlation is high a post-model bias correction would be straightforward. (Although of course station gauge data always have some bias of their own against the unknown truth!). Furthermore, given the almost certainly large uncertainty and bias in the CHIRPS rainfall data for this region, it is likely that a parameter set minimising bias will compensate for the biases and errors I the rainfall data. (Perhaps there are some rain gauge data to test this). In summary, I would recommend not relying on NSE nearly as much, and also considering correlation measures, perhaps by using the KGE breakdown. At the very least, more discussion is needed.

Comment 4) Please add some discussion about the performance of the different variants against the different flow signatures introduced in l. 276-280. Rather than referring to Euser et al., why not include the formula in a table and list the performance of each model variant? I note that most of the signatures are sensitive to bias (see below) and the runoff coefficients also to bias in rainfall. That means that the potential bias in the spatial rainfall estimates and station discharge records needs to be discussed.

Comment 5) I would like to see some comparison of model vs remotely sensed GRACE and altimetry data, and the performance of the different calibrated variants against it.

Comment 6) GRACE observations are coarse and subject to various uncertainties. To better understand uncertainty relating to calibration to GRACE, can you discuss the contributions of the different storage terms to the temporal variation? This would help to understand where the main uncertainties might be, e.g., how important surface water storage variations are. Also, given the proximity of lakes, dams and wetlands (Cahora Bassa, Lake Malawi, Bangwelu wetlands), they may well have had an influence on GRACE water storage variations. There is no question they are sufficiently close to affect the signal, but perhaps their water level variations haven't been very large during the analysis period. Please discuss this and provide some evidence. For example, you could look at their water level changes (e.g. from altimetry) and you could map the temporal correlation of each GRACE pixel to the respective pixels over each of these 3 areas. Finally, please discuss the SEE between model and GRACE water storage in comparison to the random noise in the GRACE solutions.

Comment 7) The apparent benefit of having accurate river cross-section data along with the altimetry data is an interesting one, and could be the most important contribution of this m/s. Can you explore opportunities to build on this insight a bit more please? For example, it is my understanding that profiles can be derived from the altimetry measurements. I am not a radar altimetry specialist and appreciate the authors are not either, but I am sure insights can be found in the literature. Secondly, given the importance of river geometry, can you discuss whether river width and pseudo-bathymetry from optical remote sensing might help you (see Sichangi et al., 2016; Hou

et al., 2018), particularly now there are such data globally at Landsat resolution. In fact, a simple and useful addition would be to add a map of each virtual and actual gauge derived from the Global Surface Water Dataset which is a great resource (Pekel et al., 2016; https://global-surface-water.appspot.com/map). Finally, one of the other reviewers will probably already suggest you mention the SWOT mission. While not seeing inherent merit in arm-waving, in this case, it is interesting to discuss to what extent the SWOT observations might provide richer and/or more accurate data (e.g. on river cross-section and profile) than the current crop of altimeters.

References

Domeneghetti, A. (2016). On the use of SRTM and altimetry data for flood modeling in data‐sparse regions. Water Resources Research, 52, 2901-2918

Hou, J., Van Dijk, A.I.J.M., Renzullo, L.J., & Vertessy, R.A. (2018). Using modelled discharge to develop satellite-based river gauging: a case study for the Amazon Basin. Hydrology and Earth System Science, 22, 6435-6448

Oubanas, H., Gejadze, I., Malaterre, P.O., Durand, M., Wei, R., Frasson, R.P.M., & Domeneghetti, A. (2018). Discharge estimation in ungauged basins through variational data assimilation: the potential of the SWOT mission. Water Resources Research, 54, 2405-2423

Pekel, J.-F., Cottam, A., Gorelick, N., & Belward, A.S. (2016). High-resolution mapping of global surface water and its long-term changes. Nature, 540, 418-422

Sichangi, A.W., Wang, L., Yang, K., Chen, D., Wang, Z., Li, X., Zhou, J., Liu, W., & Kuria, D. (2016). Estimating continental river basin discharges using multiple remote sensing data sets. Remote Sensing of Environment, 179, 36-53

---

## Referee Comment (RC2) · Anonymous Referee #2 · 11 Oct 2019

This paper by Hulsman et al., presents and compares several strategies for using remote sensing data for the calibration of a distributed hydrological model. It is an interesting contribution, as the poor availability of insitu data is certainly is a problem for the application of hydrological models in many regions in the world as the African test-case presented here. + the strategy of gradually reducing the parameter space of plausible simulations is also quite interesting. However in its present form the manuscript raises several questions that are listed below:

1/ My main comment is that the general strategy followed for the model calibration lacks legibility. The 7 (?) calibration strategies tested are explained at various places

in the manuscript (3.1, 3.3.1-4, again in 4.1.1-4) with a lot of redundancy and at the same time partial information here and there. We don't really understand how the strategies interact (are they all independent from each other). For example it is not clear in section 3 whether the altimetry and water level strategies were applied after the GRACE strategy or independently. Is there a reference strategy to which all other strategies are compared? We lack also information about the objectives behind the technical setup of each strategy (what are the assumptions tested, why)? I think that a synthetic table presenting the strategies and how they are linked to each other would be very informative.

2/ I have doubts on the interest of the "water level" strategies presented in the paper. They don't correspond to the title of the paper that mentions only GRACE and altimetry data. If I understood correctly, these strategies correspond to using the water level time series of the gauging station instead of the discharge data. Since the discharge data are available, what is the interest of these strategies? Is it just about reconstructing a rating curve using Google Earth cross – sections? Why not, but there is really no need to involve a hydrological model in that case. I think that the authors should question the interest of presenting these strategies in the paper, and if yes explain how they relate to the other strategies and what they bring for the use of satellite altimetry data.

3/ About water level based calibration : as shown by the results (Altimetry strategies 1 and 2) and discussed by the authors (p 25, l. 620-625; p26 l. 649-653), calibration of models directly on water level data generates additional uncertainties associated to the level – discharge transformation. Have the authors considered separating the problems by 1/ tackling the altimetry water level – discharge transformation issue (without hydrological model) 2/ considering the model multi-station calibration on discharge. It would bring a clearer theoretical framework, by separating the uncertainty sources (see for example Renard et al., 2010). Moreover, there is already a rich literature corpus on each subject, to which the authors could relate. I think this could be worth a discussion. Renard, B.; Kavetski, D.; Kuczera, G.; Thyer, M. & Franks, S. W. (2010), 'Understanding predictive uncertainty in hydrologic modeling: The challenge of identifying input and structural errors', Water Resources Research 46(5), W055521.

4/ More information should be provided in the paper about GRACE, for the readers not familiar with satellite products. In particular, readers need to understand how the GRACE water storage anomalies (what is it exactly)? can be compared to total water storage in the model (not even speaking about calibration).

5/ Many performance indicators are used in the paper and not always explained / justified. The use of NSE on variables like water storage of flow duration curve seems a bit strange, as these variables behave very differently from discharge time series for which NSE is defined. Similarly, the general performance indicator for signatures combines NSE values and relative error values. Again, it is not clear to me how this indicator can be interpreted. What is the added value of using such complex indicators instead of more direct relative errors?

6/ In the model presentation it is not clear how the flow routing in the hydrographic network is computed – or is there any channel routing at all? This is quite important to know in the context of calibration with water level data (see also remark 3/).

Minor comments: - A table of presenting the parameters (+ how many parameters and which ones were calibrated for each strategy) would be useful in the main text, instead of the detail of all model equations - Provide a table with a clear list of signatures + associated performance criteria – the reader is left to guess what goes with what when it comes to presentation and interpretation of results. - p 11 l 253: what are type II errors? - p 13 l345-350: the authors present a Distance as performance criterion like Eq 3, but there are only water levels in this strategy? Were signatures calculated here as well? - Table 4 is confusing. Why are the criteria different for each strategy in the "model efficiency" column?

---

## Referee Comment (RC3) · Anonymous Referee #3 · 15 Oct 2019

This study explores the feasibility of using water level derived from satellite observations combined with GRACE observations and other information about river cross-section shape for the calibration of a distributed hydrological model in ungauged basins. The results are interesting and potentially valuable to improve our understanding about how to more effectively use radar altimetry observations from space to trace streamflow. The paper can be published in HESS after addressing the following comments:

1.Using GRACE observation to constrain parameter space is definitely worthy to be evaluated. However, the uncertainty of GRACE observations in the model calibration process need to be considered. The parameter sets reproduce GRACE observation

well may not be reasonably reflect hydrological process of the basin. It is in doubt whether it is reasonable to discard 75% of the parameter set only based on their poor ability to reproduce GRACE observations. It is recommended to calibrate the model based on radar altimetry firstly and then based on GRACE observation. The differences between the two cases may give some new insights about amount of information contained in the two types of satellite observations for hydrological model calibration.

2.Table 4 shows that the parameter set has the highest model efficiency in calibration based on satellite observation is not necessarily to perform best in simulating streamflow. To judge which strategy is more effective in model calibration, it is suggested to show the correlations between model efficiency in simulating the satellite observations and streamflow corresponding to each parameter set.

3.The discussions about the influences of number of virtual stations on model simulation should be extended to exam its influences on streamflow estimation.

4.The spatial resolution of GRACE observations and hydrological simulation are different. How did you treat this difference in model calibration?

5.In the results and discussion section, it is expected to get more understanding about the implications for the future studies in this research field from the findings of the current study, rather than limitation and comparison with previous studies, for which the relevance to the simulation results is not very high and therefore the content need to be reduced. Also the length of abstract need to be reduced.

---

## Author Response (AR1)

We would like to thank the editor and all referees for their comments to improve our paper. Based on these comments the following main changes were applied:

- The abstract and discussion (Section 4.3) was edited to make it more concise as suggested by Referee #3.
- The introduction was changed such that the different altimetry based calibration strategies were introduced there and included in the hypothesis as suggested by Referee #1.
- In the method section (Section 3), a table was added to create an overview of the different calibration strategies (Table 4 in the manuscript) as suggested by Referee #1 and #2. With this table, we hope that it becomes clearer for the reader how the different calibration strategies build on each other and interact with each other.
- In the results and discussion section, several aspects were included, for example discussing the model performance with respect to the individual flow signatures (suggested by Referee #1), the influence of the number of virtual stations on the discharge estimation (suggested by Referee #3), GRACE uncertainties (suggested by Referee #1 and #3), and model performance metrics (suggested by Referee #1 and #2).
- A section was added to describe opportunities for future studies (Section 4.5) as suggested by Referee #1 and #3.

More details on the individual changes can be found in the responses to the reviewers and the marked-up revised manuscript. In the responses to the reviewers, a reference to the individual changes was included (page and line number) based on the marked-up revised manuscript.

Dear Anonymous Referee #1,

Thank you very much for your feedback on our paper on "Using altimetry observations combined with GRACE to select parameter sets of a hydrological model in data scarce regions". Hereby we would like to respond to your comments:

*Comment 1: I got lost in some of the technical detail around the various alternative calibration strategies tested for calibration. As the abstract seems to suggest insights relating to calibration strategy are the main contribution of this m/s, so I think this needs some more attention. For example, the research hypotheses (l. 109-11) do not address this aspect. In the introduction, can you provide some discussion around the rationale for the different experiments? In fact, to support that, it would be helpful if the authors could provide a table listing, for each variant, the objective function, any transformation of model or observation data (i.e. the observation model), the potential benefits of the variant (i.e., why was it tested), and the empirically-found pros and cons.*

Response: This is an excellent suggestion - we agree, the different calibration strategies with respect to altimetry were only introduced in the methods section (Section 3.3.3), but not mentioned in the introduction nor the hypothesis. We changed this in the manuscript by adding a section in the introduction (p. 3, l. 91) and adjusting the hypothesis (p. 4, l. 120). In addition, we agree it would be helpful for the readers to include an overview of the different calibration strategies including their objective functions, discharge – water level conversion techniques and benefits/drawbacks. Therefore, the table shown below was included (Table 4 in the manuscript).

*Comment 2: Please consult Domeneghetti (2016) and Oubanas et al. (2018) and consider whether they may be relevant to your discussion.*

Response: Thank you for pointing out these interesting papers! We included them in the manuscript (p. 31, l. 761 and 763).

*Comment 3: With the caveat that I did not understand all details, I seem to gather that one of the main conclusions of this m/s is that selecting parameters based on rank correlation between discharge and altimetry water level is not sufficient to constrain model parameters, and that altimetry levels need to be converted to actual discharge to provide an appropriate constraint. Is that correct? If so, then that would be expected when evaluating against a performance measure that is extremely bias-sensitive, like Nash-Sutcliffe efficiency (NSE). However, while I know NSE is religiously adhered to by some hydrologists, it is not a relevant performance indicator for all possible uses of river discharge modelling (and indeed many hydrologists have already found a new religion in the more information-rich components of Kling-Gupta Efficiency, KGE). For many practical applications, a high correlation may well be more important than a bias-free estimate, for example in flood and drought applications. Even if volumetric accuracy is more important (e.g. in water resources volume management) then, in this case, you have some gauged data, so provided correlation is high a post-model bias correction would be straightforward. (Although of course station gauge data always have some bias of their own against the unknown truth!). Furthermore, given the almost certainly large uncertainty and bias in the CHIRPS rainfall data for this region, it is likely that a parameter set minimising bias will compensate for the biases and errors I the rainfall data. (Perhaps there are some rain gauge data to test this). In summary, I would recommend*

*not relying on NSE nearly as much, and also considering correlation measures, perhaps by using the KGE breakdown. At the very least, more discussion is needed.*

Response: We agree the Nash-Sutcliffe efficiency has its limitations just as any other metric. Nevertheless, it still can provide us with valuable information. For many applications in water resources management, it is important to capture both the flow dynamics and volumes correctly; for instance for the management of a dam in the context of flood/drought protection. In that case, a bias-sensitive performance metric such as the Nash-Sutcliffe becomes quite useful.

The Spearman Rank Correlation function only accounts for the dynamical changes and not for the volume which indeed could be taken into account by using information available through gauged data as Referee #1 suggested. This would be possible for this study, but hypothetically, what if no gauged information would be available (which is the reality for the vast majority of river basins world-wide)? This was the assumption made in this study, to answer the question of how well can we do to reproduce river flow in a basin where no flow observations are available and only altimetry data are used for model calibration. Then this study illustrated the added value of converting the water level to discharge using the Strickler-Manning equation to capture the volume better. We agree that the bias in for example the rainfall data can then be compensated through the additional calibration parameters which therefore need to be constrained as much as possible. A section on the model performance metric was added in the discussion (p.30, l.720).

*Comment 4: Please add some discussion about the performance of the different variants against the different flow signatures introduced in l. 276-280. Rather than referring to Euser et al., why not include the formula in a table and list the performance of each model variant? I note that most of the signatures are sensitive to bias (see below) and the runoff coefficients also to bias in rainfall. That means that the potential bias in the spatial rainfall estimates and station discharge records needs to be discussed.*

Response: Thank you for pointing this out. The performance of the different variants with respect to the different flow signatures is visualized in Figure 7 in the original manuscript, but not discussed explicitly as the focus was on improving the over model performance with respect to the flow. We included this in the manuscript (p. 24, l. 519; p.25, l. 551, 566 and 580; p. 26, l. 600 and 610) together with a table summarizing the formulas for the different flow signatures (Table 5 in the manuscript). In addition, a detailed table summarizing the model performance for each calibration strategy with respect to each flow signatures (as shown in Figure 7) was added in the supplementary material for the interested reader (Table S4).

**Table 1: Overview of the calibration strategies applied in this study**

| Calibration strategy name | Calibration data | Objective function | Nr. of calibration parameters | Comments | Discharge – water level conversion method | Benefits (+) & limitations (-) |
|---|---|---|---|---|---|---|
| **Discharge (reference)** | Discharge (at basin outlet) | $D_E$ | 17 | Traditional model calibration on observed flow data. Combination of 8 different flow signatures | - | - |
| **Seasonal water storage** | GRACE | $E_{NS,Stot}$ | 17 | No discharge data used | - | - |
| **Altimetry Strategy 1** | Altimetry (at 18 virtual stations) & GRACE | Altimetry: $D_{E,R,WL}$ GRACE: $E_{NS,Stot}$ | 17 | No discharge data used. Combination of 18 virtual stations. Combined with GRACE | - | + No extra parameters or data needed + Assumption: monotonic relation between discharge and river water level - Focus on dynamics only, not volume |
| **Altimetry Strategy 2** | Altimetry (at 18 virtual stations) & GRACE | Altimetry: $D_{E,NS,RC}$ GRACE: $E_{NS,Stot}$ | 25 | No discharge data used. Combination of 18 virtual stations. Combined with GRACE | Calibrated Rating curve | + No extra data needed - Two extra parameters per cross-section |
| **Altimetry Strategy 3** | Altimetry (at 18 virtual stations) & GRACE | Altimetry: $D_{E,NS,SM}$ GRACE: $E_{NS,Stot}$ | 18 | No discharge data used. Combination of 18 virtual stations. Combined with GRACE | Strickler-Manning | + Only 1 extra parameter - Cross-section data needed - Assumption: constant roughness in space and time |
| **Water level Strategy 1** | Water level (at basin outlet) & GRACE | Altimetry: $E_{NS,SM,GE}$ GRACE: $E_{NS,Stot}$ | 18 | No discharge data used. Combined with GRACE | Strickler-Manning | + Only 1 extra parameter - Cross-section data needed - Assumption: constant roughness in space and time |
| **Water level Strategy 2** | Water level (at basin outlet) & GRACE | Altimetry: $E_{NS,SM,GE}$ GRACE: $E_{NS,Stot}$ | 18 | No discharge data used. Combined with GRACE | Strickler-Manning | + Only 1 extra parameter - Cross-section data needed - Assumption: constant roughness in space and time |

*Comment 5: I would like to see some comparison of model vs remotely sensed GRACE and altimetry data, and the performance of the different calibrated variants against it.*

Response: In the manuscript, we wanted to find out whether accurate discharge simulations can be obtained when calibrating to altimetry which is why supporting graphs visualizing the flow were mainly shown.

But with Figure 1, we would like to illustrate the difference between the following two strategies with respect to the water level: converting the simulated discharge to observed water levels using 1) calibrated rating curves and 2) the Strickler-Manning equation. Both strategies were applied to Virtual Station 4 (see Figure 2 for its location) as an example. This graph shows that the water level simulations improved significantly when applying the Strickler-Manning equation and including cross-section information. In addition, Figure 3 shows the model simulation results with respect to the total water storage compared to GRACE. This was added in the manuscript in the results section (p. 24, l. 513; p. 25, l. 558 and 577; Figure 9) and in the supplements (Figure S6).

[Figure]

**Figure 1: Range of model solutions for Virtual Station 4 (see Figure 2 for its location). The left panel shows the time series and the right panel the exceedance probability graph of the recorded (black) and modelled water level: the line indicates the solution with the highest calibration objective function and the shaded area the envelope of the solutions retained as feasible. Solutions retained as feasible based on altimetry observations using all virtual stations within the basin and 1) calibrated rating curves for the discharge – water level conversion or 2) the Strickler-Manning equation with cross-section information retrieved from Google Earth.**

[Figure]

**Figure 2: Map of the Luangwa River Basin illustrating the location of Virtual Station 4**

[Figure]

**Figure 3: Range of random model realizations with respect to the total water storage (grey) including the observation according to GRACE (black).**

*Comment 6: GRACE observations are coarse and subject to various uncertainties. To better understand uncertainty relating to calibration to GRACE, can you discuss the contributions of the different storage terms to the temporal variation? This would help to understand where the main uncertainties might be, e.g., how important surface water storage variations are. Also, given the proximity of lakes, dams and wetlands (Cahora Bassa, Lake Malawi, Bangwelu wetlands), they may well have had an influence on GRACE water storage variations. There is no question they are sufficiently close to affect the signal, but perhaps their water level variations haven't been very large during the analysis period. Please discuss this and provide some evidence. For example, you could look at their water level changes (e.g. from altimetry) and you could map the temporal correlation of each GRACE pixel to the respective pixels over each of these 3 areas. Finally, please discuss the SEE between model and GRACE water storage in comparison to the random noise in the GRACE solutions.*

Response: Thank you for this comment. There are indeed quite some uncertainties and random noise in the GRACE observations; this was included in the discussion as it was missing (p. 29, l. 679). Within the hydrological system, there are several components that contribute to the total water storage such as the water stored on the surface, in the shallow subsurface zone and in the groundwater. The temporal variation of the first two components is relatively high whereas groundwater levels change slowly. The temporal variation in the

monthly total water storage is dominated by the slow variations in the groundwater level. In addition, there are strong seasonal variations in this region due to a very clear wet and dry season which is reflected in all storage components.

As pointed out by Referee #1, there are several lakes/reservoirs and wetlands in the area that could affect GRACE observations. For example the water level variations at Cahora Bassa are significantly larger than the variations in GRACE focusing on the pixel where the virtual station is located (Figure 4A). This influence decreases when focusing on a larger area for instance the area within a 300 km radius of the virtual station (Figure 4B) which is the same distance used to smooth the data and filter out noise. Similar results were found for the other open water bodies and swamps mentioned by Referee #1. In this study, the smallest distance between the basin and a large open water body or swamp was 51 km for Lake Malawi, 72 km for Kafue Flats, 74 km for Cahora Bassa, 135 km for Kariba, 173 for Bangweulu and 210 km for Tanganyika (Figure 5). Hence, large open water bodies and swamps can indeed affect GRACE observations considerably especially for small areas.

[Figure]

**Figure 4: Altimetry observations at Cahora Bassa (blue) and average total water storage according to GRACE (black) for the following areas of interest: A) GRACE cell in which the virtual station is located, and B) area surrounding the virtual station with a radius of 3 degree (GRACE area of influence). The altimetry was multiplied with the area-weighted contribution of open water within the area of interest.**

[Figure]

**Figure 5: Map of virtual stations**

The temporal correlation of each GRACE cell relative to a specific cell is illustrated in Figure 6. For this purpose, the GRACE observations for the cell in which the virtual station for Cahora Bassa is located is plotted

against all cells within an area surrounding the virtual station with a radius of 3 degree (GRACE area of influence). This figure shows that there is a relatively strong temporal correlation between the GRACE cells which could be a result of for example the strong seasonality in this area. However, the temporal correlation between GRACE and the altimetry observation is significantly weaker for this example (blue dots in Figure 6). This indicates that the Cahora Bassa reservoir had a limited impact on the GRACE observations within its representative cell despite the large fluctuations.

Unfortunately, we are not sure what Referee #1 is referring to when mentioning the abbreviation "SEE".

[Figure]

**Figure 6: Temporal correlation of the GRACE observations for the cell in which the virtual station for Cahora Bassa is located (horizontal axis) and for A) all cells within an area surrounding the virtual station with a radius of 3 degree (GRACE area of influence, vertical axis, black), and B) the altimetry observation at Cahora Bassa (vertical axis, blue). The 1:1 line is visualised in red.**

*Comment 7: The apparent benefit of having accurate river cross-section data along with the altimetry data is an interesting one, and could be the most important contribution of this m/s. Can you explore opportunities to build on this insight a bit more please? For example, it is my understanding that profiles can be derived from the altimetry measurements. I am not a radar altimetry specialist and appreciate the authors are not either, but I am sure insights can be found in the literature. Secondly, given the importance of river geometry, can you discuss whether river width and pseudobathymetry from optical remote sensing might help you (see Sichangi et al., 2016; Hou et al., 2018), particularly now there are such data globally at Landsat resolution. In fact, a simple and useful addition would be to add a map of each virtual and actual gauge derived from the Global Surface Water Dataset which is a great resource (Pekel et al., 2016; https://global-surface-water.appspot.com/map). Finally, one of the other reviewers will probably already suggest you mention the SWOT mission. While not seeing inherent merit in arm-waving, in this case, it is interesting to discuss to what extent the SWOT observations might provide richer and/or more accurate data (e.g. on river cross-section and profile) than the current crop of altimeters.*

Response: We agree, showing the benefit of detailed cross-section information when combining it with altimetry observations is the main major finding of this manuscript. This approach has a lot of potential. For example, it would be very interesting to combine altimetry observations with river width estimates derived from Landsat or Sentinel-1/2 (Huang et al., 2018). Alternatively, altimetry observations could be combined with CryoSat based altimetry observations which provide water level information at lower temporal resolution (every 369 days), but higher spatial resolution (equatorial inter-track distance of 7.5 km) providing valuable information to estimate the river slope (Schneider et al., 2017;Jiang et al., 2017). In addition, with the upcoming SWOT (Surface Water Ocean Topography) mission, more accurate altimetry observations should be available as also river slope observations and cross-sections; the repeat cycle will be 21 days and across-track resolution between 10 m and

60 m increasing the number of observation points available within a specific area (Biancamaria et al., 2016;Langhorst et al., 2019). Also, it would be very useful to improve cross-section estimates with respect to the submerged part as already explored in previous studies (Domeneghetti, 2016). Furthermore, drone observations could be used to obtain more accurate cross-section information and estimates of the river slope and roughness (Entwistle and Heritage, 2019). We added a more extensive discussion on these points in the revised manuscript (Section 4.5 was added).

**Literature**

Biancamaria, S., Lettenmaier, D. P., and Pavelsky, T. M.: The SWOT Mission and Its Capabilities for Land Hydrology, Surveys in Geophysics, 37, 307-337, 10.1007/s10712-015-9346-y, 2016.

Domeneghetti, A.: On the use of SRTM and altimetry data for flood modeling in data-sparse regions, Water Resources Research, 52, 2901-2918, 10.1002/2015WR017967, 2016.

Entwistle, N. S., and Heritage, G. L.: Small unmanned aerial model accuracy for photogrammetrical fluvial bathymetric survey, Journal of Applied Remote Sensing, 13, 1-19, 19, 2019.

Huang, Q., Long, D., Du, M., Zeng, C., Qiao, G., Li, X., Hou, A., and Hong, Y.: Discharge estimation in high-mountain regions with improved methods using multisource remote sensing: A case study of the Upper Brahmaputra River, Remote Sensing of Environment, 219, 115-134, https://doi.org/10.1016/j.rse.2018.10.008, 2018.

Jiang, L., Schneider, R., Andersen, O. B., and Bauer-Gottwein, P.: CryoSat-2 altimetry applications over rivers and lakes, Water (Switzerland), 9, 10.3390/w9030211, 2017.

Langhorst, T., Pavelsky, T. M., Frasson, R. P. d. M., Wei, R., Domeneghetti, A., Altenau, E. H., Durand, M. T., Minear, J. T., Wegmann, K. W., and Fuller, M. R.: Anticipated Improvements to River Surface Elevation Profiles From the Surface Water and Ocean Topography Mission, Frontiers in Earth Science, 7, 102, 2019.

Schneider, R., Godiksen, P. N., Villadsen, H., Madsen, H., and Bauer-Gottwein, P.: Application of CryoSat-2 altimetry data for river analysis and modelling, Hydrol. Earth Syst. Sci., 21, 751-764, 10.5194/hess-21-751-2017, 2017.

Dear Anonymous Referee #2,

Thank you very much for your feedback on our paper "Using altimetry observations combined with GRACE to select parameter sets of a hydrological model in data scarce regions". To respond to your comments:

*Comment 1: My main comment is that the general strategy followed for the model calibration lacks legibility. The 7 (?) calibration strategies tested are explained at various places in the manuscript (3.1, 3.3.1-4, again in 4.1.1-4) with a lot of redundancy and at the same time partial information here and there. We don't really understand how the strategies interact (are they all independent from each other). For example it is not clear in section 3 whether the altimetry and water level strategies were applied after the GRACE strategy or independently. Is there a reference strategy to which all other strategies are compared? We lack also information about the objectives behind the technical setup of each strategy (what are the assumptions tested, why)? I think that a synthetic table presenting the strategies and how they are linked to each other would be very informative.*

Response: This is an excellent suggestion - we agree that including a table presenting the strategies and their links would be very helpful; therefore Table 1, as shown below, was added (Table 4 in the revised manuscript). Each strategy was explained in detail in the methods section (Section 3), but we adjusted these descriptions and directly linked them to the new table. When explaining the results for each strategy, the individual strategies were briefly summarized in the results section to help the reader. However, this might not be necessary anymore when including the new table. We hope that with this table it becomes clearer how the different calibration strategies build on each other and interact with each other: the overall objective of this paper is to explore how well we can select parameter sets for hydrological models in catchments when \*no\* flow observations are available. The sequence of strategies is therefore meant to follow the potential thought process of a modeler in such an ungauged situation: first remove parameter sets that cannot reproduce the seasonal signal as indicated by GRACE. As this set of solutions still (at least in our case) contains many solutions that cannot reproduce river flow in a reasonable way, the set is subsequently further constrained by water level data from altimetry observations. This, in itself, has similarly little additional constraining power. Thus, water levels were converted to flow using different methods, including calibrated rating curves and the Strickler-Manning formula.

*Comment 2: I have doubts on the interest of the "water level" strategies presented in the paper. They don't correspond to the title of the paper that mentions only GRACE and altimetry data. If I understood correctly, these strategies correspond to using the water level time series of the gauging station instead of the discharge data. Since the discharge data are available, what is the interest of these strategies? Is it just about reconstructing a rating curve using Google Earth cross – sections? Why not, but there is really no need to involve a hydrological model in that case. I think that the authors should question the interest of presenting these strategies in the paper, and if yes explain how they relate to the other strategies and what they bring for the use of satellite altimetry data.*

Response: Thank you for this comment. The "water level" strategies indeed used water level time series at the gauge station. The objective of including this strategy was to illustrate the importance of incorporating more accurate cross-section information. At the locations where altimetry observations were available, cross-section information was extracted from high-resolution terrain maps available on Google Earth. This, unfortunately, has

a low accuracy, leaving us with inaccurate cross-section information at these locations. Unfortunately, accurate cross-section information from in-situ surveys was only available at the gauging station where, in turn, no altimetry observations are available. That is why water level time series were used to illustrate the importance of using more accurate cross-section information. We clarified that in the manuscript (p. 18, l. 413).

*Comment 3: About water level based calibration : as shown by the results (Altimetry strategies 1 and 2) and discussed by the authors (p 25, l. 620-625; p26 l. 649-653), calibration of models directly on water level data generates additional uncertainties associated to the level – discharge transformation. Have the authors considered separating the problems by 1/ tackling the altimetry water level – discharge transformation issue (without hydrological model) 2/ considering the model multi-station calibration on discharge. It would bring a clearer theoretical framework, by separating the uncertainty sources (see for example Renard et al., 2010). Moreover, there is already a rich literature corpus on each subject, to which the authors could relate. I think this could be worth a discussion. Renard, B.; Kavetski, D.; Kuczera, G.; Thyer, M. & Franks, S. W. (2010), 'Understanding predictive uncertainty in hydrologic modeling: The challenge of identifying input and structural errors', Water Resources Research 46(5), W055521.*

Response: It would indeed be interesting to separate the uncertainties related to the discharge – water level conversion from the hydrological model. For this conversion, there are several calibration parameters. So when disentangling this from the model, alternative information sources would be needed to estimate these parameters. Unfortunately, there was no further really useful information available at the virtual stations to estimate these parameters. This would be very interesting for a follow-up study doing exactly that in a more data rich region to look into these uncertainties more detailed. This aspect was added to the discussion section (p. 31, l. 764).

*Comment 4: More information should be provided in the paper about GRACE, for the readers not familiar with satellite products. In particular, readers need to understand how the GRACE water storage anomalies (what is it exactly)? can be compared to total water storage in the model (not even speaking about calibration).*

Response: Thank you for pointing this out. A section explaining GRACE more detailed for those not familiar with this product was indeed missing and was added to Section 2.1.2 (p. 5, l. 155).

*Comment 5: Many performance indicators are used in the paper and not always explained / justified. The use of NSE on variables like water storage of flow duration curve seems a bit strange, as these variables behave very differently from discharge time series for which NSE is defined. Similarly, the general performance indicator for signatures combines NSE values and relative error values. Again, it is not clear to me how this indicator can be interpreted. What is the added value of using such complex indicators instead of more direct relative errors?*

Response: In this paper we indeed used the Nash-Sutcliffe efficiency for the discharge time series, but also for other signatures such as the flow duration curve, and other variables such as total water storage or water level. Even though the Nash-Sutcliffe efficiency was originally defined for discharge time series as pointed out by Referee #2, it was assumed this performance metric can also provide valuable information for other signatures or variables. We agree additional study is required to confirm this assumption and to assess which performance metric would be most suitable, but this was beyond the scope of this study. We included this issue in the

discussion (p. 30, l. 720). Furthermore, we wanted to incorporate multiple signatures of the discharge in the performance metric, instead of focusing on only part of the information available in the discharge time series. That is why these performance metrics for each signature were combined, in a similar way as in many earlier studies doing multi-objective calibration. However, we also agree that in the choice of objective functions there is always a strong subjective component and one error metric may be able to capture some aspects of the response better than another one. The reason we did not only show the individual performance indicators, but also decided to provide the combined ones is that we think that a good model should be able to reproduce all indicators simultaneously as well as possible. As there is quite some difference between the performance levels of different indicators, it is difficult to see the overall effect, when only analyzing their individual values.

*Comment 6: In the model presentation it is not clear how the flow routing in the hydrographic network is computed – or is there any channel routing at all? This is quite important to know in the context of calibration with water level data (see also Comment 3).*

Response: The flow routing scheme was indeed explained only briefly in the manuscript. For the flow routing, the mean flow length of each model gird cell to the outlet was derived based on the topography using a digital elevation map. In addition, it was assumed the flow velocity was constant in space and time; this velocity was calibrated. With this information on the flow path length and velocity, the accumulated flow in each grid cell was calculated at the end of each time step. This explained this in more detail in the manuscript (p. 9, l. 242).

*Minor comments: - A table of presenting the parameters (+ how many parameters and which ones were calibrated for each strategy) would be useful in the main text, instead of the detail of all model equations - Provide a table with a clear list of signatures + associated performance criteria – the reader is left to guess what goes with what when it comes to presentation and interpretation of results. - p 11 l 253: what are type II errors? - p 13 l345-350: the authors present a Distance as performance criterion like Eq 3, but there are only water levels in this strategy? Were signatures calculated here as well? - Table 4 is confusing. Why are the criteria different for each strategy in the "model efficiency" column?*

Response: We included a table to create an overview of the different calibration strategies and performance metrics (Table 5); we hope that this will also make it clearer why there are different performance metrics for each calibration strategy in Table 4 in the manuscript. A table of the parameters was presented in the supplements (Table S1). A type I error is the rejection of a true hypothesis (e.g. a good parameter set that was supposed to be accepted got rejected), while a type II error is the non-rejection of a false hypothesis (e.g. a bad parameter set that was supposed to be rejected got accepted); this was changed in the manuscript to avoid any confusion (p. 12, l. 276). When calculating the model performance with respect to altimetry, the Euclidian distance was used to combine the model performance of each individual virtual station into one error metric; we agree though the reference to Eq. 3 is confusing; that is why Eq. 5 was introduced.

**Table 2: Overview of the calibration strategies applied in this study**

| Calibration strategy name | Calibration data | Objective function | Nr. of calibration parameters | Comments | Discharge – water level conversion method | Benefits (+) & limitations (-) |
|---|---|---|---|---|---|---|
| **Discharge (reference)** | Discharge (at basin outlet) | $D_E$ | 17 | Traditional model calibration on observed flow data  Combination of 8 different flow signatures | - | - |
| **Seasonal water storage** | GRACE | $E_{NS,Stot}$ | 17 | No discharge data used | - | - |
| **Altimetry Strategy 1** | Altimetry (at 18 virtual stations) & GRACE | Altimetry: $D_{E,R,WL}$  GRACE: $E_{NS,Stot}$ | 17 | No discharge data used  Combination of 18 virtual stations  Combined with GRACE | - | + No extra parameters or data needed  + Assumption: monotonic relation between discharge and river water level  - Focus on dynamics only, not volume |
| **Altimetry Strategy 2** | Altimetry (at 18 virtual stations) & GRACE | Altimetry: $D_{E,NS,RC}$  GRACE: $E_{NS,Stot}$ | 25 | No discharge data used  Combination of 18 virtual stations  Combined with GRACE | Calibrated Rating curve | + No extra data needed  - Two extra parameters per cross-section |
| **Altimetry Strategy 3** | Altimetry (at 18 virtual stations) & GRACE | Altimetry: $D_{E,NS,SM}$  GRACE: $E_{NS,Stot}$ | 18 | No discharge data used  Combination of 18 virtual stations  Combined with GRACE | Strickler-Manning | + Only 1 extra parameter  - Cross-section data needed  - Assumption: constant roughness in space and time |
| **Water level Strategy 1** | Water level (at basin outlet) & GRACE | Altimetry: $E_{NS,SM,GE}$  GRACE: $E_{NS,Stot}$ | 18 | No discharge data used  Combined with GRACE | Strickler-Manning | + Only 1 extra parameter  - Cross-section data needed  - Assumption: constant roughness in space and time |
| **Water level Strategy 2** | Water level (at basin outlet) & GRACE | Altimetry: $E_{NS,SM,GE}$  GRACE: $E_{NS,Stot}$ | 18 | No discharge data used  Combined with GRACE | Strickler-Manning | + Only 1 extra parameter  - Cross-section data needed  - Assumption: constant roughness in space and time |

Dear Anonymous Referee #3,

Thank you for your feedback on our paper "Using altimetry observations combined with GRACE to select parameter sets of a hydrological model in data scarce regions". We hereby would like to respond to your comments:

*Comment 1: Using GRACE observation to constrain parameter space is definitely worthy to be evaluated. However, the uncertainty of GRACE observations in the model calibration process need to be considered. The parameter sets reproduce GRACE observation well may not be reasonably reflect hydrological process of the basin. It is in doubt whether it is reasonable to discard 75% of the parameter set only based on their poor ability to reproduce GRACE observations. It is recommended to calibrate the model based on radar altimetry firstly and then based on GRACE observation. The differences between the two cases may give some new insights about amount of information contained in the two types of satellite observations for hydrological model calibration.*

Response: We agree there are quite some uncertainties related to the GRACE observations, especially when using it for a small river basin. This was included in the discussion as it was still missing (p. 29, l. 679). As Referee #3 suggested, we compared the following two calibration approaches: 1) calibrate based on GRACE first, then altimetry (done in the manuscript), 2) calibrate to altimetry first, then GRACE (new). This change of the order mostly affected the selection of the "best" parameter set, especially for Altimetry 1 and Water level 2 (Figure 1), but affected the selection of feasible parameter sets less when using altimetry data as can be seen by the similar ranges in the boxplots. This order had a larger effect when using water level time series at the gauge station. This was added in the manuscript (p. 24, l. 528 and Figure S9)

[Figure]

**Figure 7: Model performance with respect to discharge for each calibration strategy. Parameter sets were selected based on A) first GRACE, then (satellite based) river water level, or B) first (satellite based) river water level, then GRACE.**

*Comment 2: Table 4 shows that the parameter set has the highest model efficiency in calibration based on satellite observation is not necessarily to perform best in simulating streamflow. To judge which strategy is more effective in model calibration, it is suggested to show the correlations between model efficiency in simulating the satellite observations and streamflow corresponding to each parameter set.*

Response: Thank you for this interesting comment. As recommended by Referee #3, Figure 2 visualizes the correlation between the model efficiency with respect to (satellite based) river water levels and with respect to discharge. This figure shows that a high model performance with respect to the stream levels did not necessarily mean the model performance with respect to discharge was high. This was added in the manuscript (p. 25, l. 548, p. 26, l. 563 and Figure S8).

[Figure]

**Figure 8: Model performance with respect to discharge (horizontal axes) vs. model performance with respect to (satellite based) river water level (vertical axes) for each calibration strategy**

*Comment 3: The discussions about the influences of number of virtual stations on model simulation should be extended to exam its influences on streamflow estimation.*

Response: It is indeed a very interesting idea to extend the analysis such that the influence of the number of virtual stations on the streamflow simulation is included. As illustrated in Figure 3, the model performance with respect to discharge increased when using more virtual stations. However, at some point an optimum is reached where the model performance remained constant even when adding more virtual stations. The number of virtual stations where this optimum was reached varied per strategy. We added this point to the discussion of the results (p. 28, l. 644 and Figure S5).

[Figure]

[Figure]

**Figure 9: Model performance with respect to discharge (all signatures combined) using an increasing number of virtual stations for calibration**

*Comment 4: The spatial resolution of GRACE observations and hydrological simulation are different. How did you treat this difference in model calibration?*

Response: We agree, in the manuscript it was not explained how we dealt with the differences in the spatial resolution. The gridded information was rescaled to the model resolution of $0.1°$. If the resolution of the satellite product was higher than $0.1°$, then the area weighted average was taken of all cells located within each model cell. Otherwise, each cell of the satellite product was divided into multiple cells such that the model resolution is obtained, retaining the original value. This was included in Section 2.1.2 (p. 5, l. 159).

*Comment 5: In the results and discussion section, it is expected to get more understanding about the implications for the future studies in this research field from the findings of the current study, rather than limitation and comparison with previous studies, for which the relevance to the simulation results is not very high and therefore the content need to be reduced. Also the length of abstract need to be reduced.*

Response: Thank you for this comment. There are indeed many interesting opportunities for future studies that could be included in the manuscript; this was done by including a new section in the discussion (Section 4.5). For example, 
[revised manuscript text omitted]
_{\mathrm{NS,SM,GE,opt}}$) resulted in $E_{\mathrm{NS,Q,GE}}$ = 0.65 ($E_{\mathrm{NS,Q,5/95,GE}}$ = -0.48 – 0.60) and $D_{\mathrm{E,GE}}$ = 0.77 ($D_{\mathrm{E,GE,5/95}}$ = 0.28 – 0.70) with respect to discharge (Figure 7, Table 6); the model performance with respect to the remaining signatures as visualised in Figure 7 are tabulated in Table S4. As shown in Figure 8G, the discharge was overestimated in particular during intermediate and low flows.

**Water level Strategy 2: River geometry information obtained from a detailed field survey**

 The parameter identification strategy "Water level Strategy 2", using cross-section information obtained from a detailed field survey, resulted in improved river water level simulations (compare Figure 10A and B) with an optimal objective function value with respect to river water levels of $E_{\mathrm{NS,SM,ADCP,opt}}$ = 0.79 ($E_{\mathrm{NS,SM,ADCP,5/95}}$ = 0.60 – 0.74). The parameter set associated with the best performing model with respect to river water levels ($E_{\mathrm{NS,SM,ADCP,opt}}$) resulted in $E_{\mathrm{NS,Q,ADCP}}$ = 0.14 ($E_{\mathrm{NS,Q,5/95,ADCP}}$ = -1.1 – 0.50) and in $D_{\mathrm{E,ADCP}}$ = 0.55 ($D_{\mathrm{
[revised manuscript text omitted]

1205

---

## Author Response (AR2)

We would like to thank the editor and referees for their feedback on our paper. We are sorry if we did not completely satisfy the reviewers' requests. We are happy that we got a second chance to meet these demands. We hope that by this revision we completely take away the concerns raised. We have updated the paper based on these comments with the following main changes:

- The introduction was updated such that GRACE was introduced and included in the objective of the study.
- A list of parameters was included in Table 5 to avoid any confusion on the number of parameters for each calibration strategy.
- The methodology section was rearranged and shortened.
- The discussion was updated such that the implications of the points raised were included where that was missing.

More details on the changes can be found in the responses to the reviewer and the marked-up revised manuscript.

Dear Anonymous Referee #2,

Thank you for your feedback. We have revised the manuscript taking your comments into account.

*The authors do indeed give a few details on GRACE and discuss the uncertainties associated to GRACE products but a clear description of the GRACE product and how it was processed is still missing. The reader is still left to guess whether GRACE provides soil water storage in mm or storage variation in mm/time step? What are the "seasonal water storage anomalies" mentioned p 7 l 196 and p14 l 315?*

*p3 l 100-111: GRACE is not mentioned in the definition of the objectives of the study. It appears suddenly at section 3.1. There is also no reference to GRACE in the last parts of the discussion (4.4-4.5). What is the added value of GRACE + altimetry data instead of only altimetry. Could this be tested and discussed?*

GRACE was indeed not mentioned in the Introduction. In the revised manuscript, we described GRACE in the Introduction and included it in the study objectives. Altimetry observations only monitor water level dynamics, hence there is no information on the amounts of water flowing in a river; in other words absolute discharge magnitudes. By using GRACE observations, which describe the monthly total water storage anomalies, the hydrological model can be constrained further in the calibration procedure, leading to more accurate discharge estimates. With GRACE, improved simulation of the rainfall partitioning into runoff and evaporation can be attained, which was illustrated in previous studies (Rakovec et al., 2016; Bai et al., 2018).

In addition, section 2.1.2 was updated to inform the reader on more technical details including the following: Gravity Recovery and Climate Experiment (GRACE) observations describe the monthly total water storage anomalies. With two identical satellites, the variations in the Earth's gravity field were measured to detect regional mass changes which are dominated by variations in the terrestrial water storage after having accounted for atmospheric effects (Landerer and Swenson, 2012; Swenson, 2012). In this study, processed GRACE observations of Release 05 generated by CSR (Centre for Space Research), GFZ (GeoForschungsZentrum Potsdam) and JPL (Jet Propulsion Laboratory) were downloaded from the GRACE Tellus website (https://grace.jpl.nasa.gov/); the average of all three sources were used. The data processing included among others estimating terrestrial water storage variations from GRACE gravity field estimates; removing atmospheric mass changes using ECMWF (European Centre for Medium-Range Weather Forecasts) atmospheric pressure fields; removing systematic errors that cause north-south-oriented stripes; spatial smoothening to remove high frequency noise using a 300 km wide Gaussian filter; and subtracting the 2004 – 2009 time-mean from the observations to obtain water storage anomalies (Swenson and Wahr, 2006; Landerer and Swenson, 2012; Wahr et al., 1998). Processed GRACE observations describe the total terrestrial water storage "anomaly" in "equivalent water thickness" in [cm] relative to the 2004 – 2009 time-mean baseline. In other words, the "water storage anomaly" is the observed total water storage subtracted by the time-mean (Landerer and Swenson, 2012). GRACE describes the *total* water storage anomaly, which includes the variation of all terrestrial water stores present in the groundwater, soil moisture and surface water (the atmospheric water has been subtracted).

Also the discussion (section 4.3 and 4.4) has been updated by discussing the added value of including GRACE in the calibration procedure including the following: GRACE observations are prone to uncertainties as a result of data (post-) processing including for example data smoothening (Landerer and Swenson, 2012; Blazquez et al.,

2018; Riegger et al., 2012) causing neighbouring cells of 1° (≈ 111 km) not to be completely independent from each other. Additionally, GRACE observations are more accurate for large areas; depending on the applied processing scheme, the error is about 2 cm for basins with an area of around 63 000 km$^2$ (Landerer and Swenson, 2012; Vishwakarma et al., 2018). Also note that due to the coarse temporal resolution, monthly GRACE observations only provide information on slow changing processes such as the groundwater and the soil moisture; fast processes are missed. It is strong in monitoring the seasonal variations which is reflected in all storage components.

Strikingly, only limited studies combined altimetry with GRACE observations in the calibration procedure (Kittel et al., 2018). As altimetry observations only describe water level variations with no information on the flow amounts, GRACE provides additional valuable information to constrain flow volumes by improving the rainfall runoff partitioning as demonstrated in previous studies (Rakovec et al., 2016; Bai et al., 2018). Combining both data sources in the calibration procedure allowed for a more accurate identification of feasible parameter sets; the model performance range with respect to discharge improved from $D_{E,5/95}$ = -8.4 – 0.77, when using only altimetry, to $D_{E,5/95}$ = 0.19 – 0.75 when combining GRACE and altimetry for Altimetry Strategy 3 (see Figure S9). Unfortunately, GRACE observations are prone to several sources of uncertainties and limitations as explained in the previous section, which could result in inadvertently discarding behavioural parameter sets when calibrating with respect to altimetry and GRACE simultaneously.

*A clear Table with the list of model parameters was requested by the Reviewers but not provided. There are a Figure and Table in Supplementary Materials but no reference is made to them in the main text so they are useless. Besides, they are themselves quite confusing. The authors declare 17 to 25 calibration parameters (Table 4) but depending what Figure / Table the reader looks at, it shows between 9 and 27 parameters… In Table 3 there is also a list of parameters in the caption where I can count 11 parameters + a reference to Hydrological Response Units that comes out of nowhere.*

We apologize for not having added a clearer table in the previous version. The reviewer is correct in stating that the combination of different models and calibration strategies tested here, together with the varying numbers of free calibration parameters caused some confusion.   We have now adapted Table 5 (Table 4 in the original manuscript) in the revised manuscript such that the respective calibration parameters are listed for each calibration strategy. We hope this helps the reader in getting a clear image on the model parameters for each calibration approach. In total, there are 27 different calibration parameters, but none of the calibration strategies include them all simultaneously (see Table R1 here below). In the benchmark reference model, there were 18 parameter sets (instead of 17 as mentioned in the manuscript).

The schematization (Figure 2) and the description of the model structure (Section 3.2) in the previous version of the manuscript already included explicit reference to and explanation of the hydrological response units used. We have further clarified this in the newly revised version of the manuscript.

**Table R1: Overview of calibration parameters**

| Strategy | Parameter group | Calibration parameters |
|---|---|---|
| **Discharge** **(reference)** | Entire basin | $K_s$, $C_e$ |
| | Plateau & Terrace | $I_{max}$, $S_{umax}$, $K_f$, W |
| | Hillslope | $I_{max}$, $S_{umax}$, $K_f$, W, β, $T_{lag}$ |
| | Wetland | $I_{max}$, $S_{umax}$, $K_f$, W, $C_{max}$ |
| | River profile | v |
| | | Total: 18 |
| **Seasonal** **water storage** | Entire basin | $K_s$, $C_e$ |
| | Plateau & Terrace | $I_{max}$, $S_{umax}$, $K_f$, W |
| | Hillslope | $I_{max}$, $S_{umax}$, $K_f$, W, β, $T_{lag}$ |
| | Wetland | $I_{max}$, $S_{umax}$, $K_f$, W, $C_{max}$ |
| | River profile | v |
| | | Total: 18 |
| **Altimetry** **Strategy 1** | Entire basin | $K_s$, $C_e$ |
| | Plateau & Terrace | $I_{max}$, $S_{umax}$, $K_f$, W |
| | Hillslope | $I_{max}$, $S_{umax}$, $K_f$, W, β, $T_{lag}$ |
| | Wetland | $I_{max}$, $S_{umax}$, $K_f$, W, $C_{max}$ |
| | River profile | v |
| | | Total: 18 |
| **Altimetry** **Strategy 2** | Entire basin | $K_s$, $C_e$ |
| | Plateau & Terrace | $I_{max}$, $S_{umax}$, $K_f$, W |
| | Hillslope | $I_{max}$, $S_{umax}$, $K_f$, W, β, $T_{lag}$ |
| | Wetland | $I_{max}$, $S_{umax}$, $K_f$, W, $C_{max}$ |
| | River profile | v, $a_1$, $a_2$, $a_3$, $a_4$, $b_1$, $b_1$, $b_3$, $b_4$ |
| | | Total: 26 |
| **Altimetry** **Strategy 3** | Entire basin | $K_s$, $C_e$ |
| | Plateau & Terrace | $I_{max}$, $S_{umax}$, $K_f$, W |
| | Hillslope | $I_{max}$, $S_{umax}$, $K_f$, W, β, $T_{lag}$ |
| | Wetland | $I_{max}$, $S_{umax}$, $K_f$, W, $C_{max}$ |
| | River profile | v, k |
| | | Total: 18 |
| **Water level** **Strategy 1** | Entire basin | $K_s$, $C_e$ |
| | Plateau & Terrace | $I_{max}$, $S_{umax}$, $K_f$, W |
| | Hillslope | $I_{max}$, $S_{umax}$, $K_f$, W, β, $T_{lag}$ |
| | Wetland | $I_{max}$, $S_{umax}$, $K_f$, W, $C_{max}$ |
| | River profile | v, k |
| | | Total: 19 |
| **Water level** **Strategy 2** | Entire basin | $K_s$, $C_e$ |
| | Plateau & Terrace | $I_{max}$, $S_{umax}$, $K_f$, W |
| | Hillslope | $I_{max}$, $S_{umax}$, $K_f$, W, β, $T_{lag}$ |
| | Wetland | $I_{max}$, $S_{umax}$, $K_f$, W, $C_{max}$ |
| | River profile | v, k |
| | | Total: 19 |

*Remarks were done on the presentation of the study methodology (section 3.3). Besides the addition of a synthesis Table (which is welcome), very little modifications were done to the text. As a result, the methodology section is still very long (longer than the results & discussion section) and confusing. For example, the model evaluation criteria are described along with the first calibration strategy in 3.3.1 although they were also used for the other strategies (but not all of them). I really think that this section should be reworked in depth to make things clearer.*

Thank you for this comment. It is true that the methodology section is long and detailed. However, the multi-faceted complexity of the experiment requires a somewhat lengthy description. We have tried to make the section clearer and updated Section 3.3, by moving the part describing the model evaluation to Section 3.4 as it is not part of the parameter selection procedure itself. Further, a table was added with an overview of the objective functions used in this paper as some were used for multiple strategies. In addition, this section was rearranged as can be seen in the marked-up version of the manuscript. We hope these adjustments improve this section such that it is not confusing anymore.

*On this topic I'm still not convinced that is was necessary to use all these evaluation criteria to see differences between the strategies and draw conclusions. I'm not sure that each of these signatures does bring specific information that is not already covered by another one (see McMillan et al., 2017). Using less signatures/criteria would certainly make the paper more legible.*

We agree that some signatures do have some overlapping information content and we will acknowledge that in the revised version of the manuscript. However, each signature also provides additional information that cannot be provided by other signatures. Some signatures will add more, some other will add less information. However, in the absence of more detailed and suitable data to calibrate a model, as in this data scarce environment, it is still necessary to efficiently and effectively constrain the model parameter space. Here the use of as many signatures as possible has considerable value: even if a signature only identifies one unsuitable parameter set, this set can be removed, thereby reducing model uncertainty. In any case it is desirable for any model to reproduce as many signatures as well as possible, so as to give us more confidence in the model's skill to reproduce, at least to some degree, also the actual internal dynamics of the system.

*The discussion lists the points raised by the reviewers but does not really discuss them. It should be expanded to discuss really the implications of these points for the conclusions of the study.*

Thank you for this comment. In the manuscript, several discussion points were indeed added based on review comments; this included a discussion on GRACE uncertainties, choice of model performance measure and signatures, bias in precipitation and discharge observations, separating uncertainty sources (originating from the hydrological model or the discharge – water level conversion), and opportunities for future studies.
Sections 4.3 and 4.4 were updated to highlight implications where that was missing or unclear (see marked-up manuscript). Please note, that many discussion points explain uncertainties related to data, assumptions or simplifications. It is difficult to assess their exact implications on the conclusion without further studies which is

beyond the scope of this paper. That is why we decided to inform the reader on the existence of different uncertainty sources that could impact the results to increase the awareness. In addition, we tried to remain concise taking into account the comments of previous reviewers.

Unfortunately we were not sure to which discussion points exactly the reviewer was referring to, but hope these changes improve the discussion as recommended by the Referee.

[revised manuscript text omitted]

| Strategy | Calibration data | Objective function | Parameter group | Calibration parameters | Comments | Q – h conversion | Benefits (+) & limitations (-) |
|---|---|---|---|---|---|---|---|
| **Discharge (reference)** | Discharge (at basin outlet) | $E_{NS,Q}$ (Eq.1) | Entire basin
Plateau & Terrace
Hillslope
Wetland
River profile | $K_s$, $C_e$
$I_{max}$, $S_{umax}$, $K_f$, W
$I_{max}$, $S_{umax}$, $K_f$, W, β, $T_{lag}$
$I_{max}$, $S_{umax}$, $K_f$, W, $C_{max}$
v
Total: 18 | Traditional model calibration on observed flow data
Combination of 8 different flow signatures | - | - |
| **Seasonal water storage** | GRACE | $E_{NS,Stot}$ (Eq.1) | Entire basin
Plateau & Terrace
Hillslope
Wetland
River profile | $K_s$, $C_e$
$I_{max}$, $S_{umax}$, $K_f$, W
$I_{max}$, $S_{umax}$, $K_f$, W, β, $T_{lag}$
$I_{max}$, $S_{umax}$, $K_f$, W, $C_{max}$
v
Total: 18 | No discharge data used | - | - |
| **Altimetry Strategy 1** | Altimetry (at 18 virtual stations) & GRACE | Altimetry: $D_{E,R,WL}$ (Eq.2,4)
GRACE: $E_{NS,Stot}$ (Eq.1) | Entire basin
Plateau & Terrace
Hillslope
Wetland
River profile | $K_s$, $C_e$
$I_{max}$, $S_{umax}$, $K_f$, W
$I_{max}$, $S_{umax}$, $K_f$, W, β, $T_{lag}$
$I_{max}$, $S_{umax}$, $K_f$, W, $C_{max}$
v
Total: 18 | No discharge data used
Combination of 18 virtual stations
Combined with GRACE | - | + No extra parameters or data needed
+ Assumption: monotonic relation between discharge and river water level
- Focus on dynamics only, not volume |
| **Altimetry Strategy 2** | Altimetry (at 18 virtual stations) & GRACE | Altimetry: $D_{E,NS,RC}$ (Eq.1,4)
GRACE: $E_{NS,Stot}$ (Eq.1) | Entire basin
Plateau & Terrace
Hillslope
Wetland
River profile | $K_s$, $C_e$
$I_{max}$, $S_{umax}$, $K_f$, W
$I_{max}$, $S_{umax}$, $K_f$, W, β, $T_{lag}$
$I_{max}$, $S_{umax}$, $K_f$, W, $C_{max}$
v, $a_1$, $a_2$, $a_3$, $a_4$, $b_1$, $b_1$, $b_3$, $b_4$
Total: 26 | No discharge data used
Combination of 18 virtual stations
Combined with GRACE | Calibrated Rating curve | + No extra data needed
- Two extra parameters per cross-section |
| **Altimetry Strategy 3** | Altimetry (at 18 virtual stations) & GRACE | Altimetry: $D_{E,NS,SM}$ (Eq.1,4)
GRACE: $E_{NS,Stot}$ (Eq.1) | Entire basin
Plateau & Terrace
Hillslope
Wetland
River profile | $K_s$, $C_e$
$I_{max}$, $S_{umax}$, $K_f$, W
$I_{max}$, $S_{umax}$, $K_f$, W, β, $T_{lag}$
$I_{max}$, $S_{umax}$, $K_f$, W, $C_{max}$
v, k
Total: 18 | No discharge data used
Combination of 18 virtual stations
Combined with GRACE | Strickler-Manning | + Only 1 extra parameter
- Cross-section data needed
- Assumption: constant roughness in space and time |
| **Water level Strategy 1** | Water level (at basin outlet) & GRACE | Altimetry: $E_{NS,SM,GE}$ (Eq.1)
GRACE: $E_{NS,Stot}$ (Eq.1) | Entire basin
Plateau & Terrace
Hillslope
Wetland
River profile | $K_s$, $C_e$
$I_{max}$, $S_{umax}$, $K_f$, W
$I_{max}$, $S_{umax}$, $K_f$, W, β, $T_{lag}$
$I_{max}$, $S_{umax}$, $K_f$, W, $C_{max}$
v, k
Total: 19 | No discharge data used
Combined with GRACE | Strickler-Manning | + Only 1 extra parameter
- Cross-section data needed
- Assumption: constant roughness in space and time |
| **Water level Strategy 2** | Water level (at basin outlet) & GRACE | Altimetry: $E_{NS,SM,ADCP}$ (Eq.1)
GRACE: $E_{NS,Stot}$ (Eq.1) | Entire basin
Plateau & Terrace
Hillslope
Wetland
River profile | $K_s$, $C_e$
$I_{max}$, $S_{umax}$, $K_f$, W
$I_{max}$, $S_{umax}$, $K_f$, W, β, $T_{lag}$

[revised manuscript text omitted]

---

## Author Response (AR3)

Dear Editor,

Thank you for your feedback and for accepting the manuscript. Several minor typos have been corrected as requested and as shown in the marked-up version below.

[revised manuscript text omitted]
_{NS,\theta} = 1 - \dfrac{\sum_t (\theta_{mod}(t) - \theta_{obs}(t))^2}{\sum_t (\theta_{obs}(t) - \overline{\theta_{obs}})^2}$ | $\theta$: variable | (1) |
| Spearman-Rank correlation coefficient | $E_{R,WL} = \dfrac{\text{cov}(r_{Q_{mod}}, r_{WL_{obs}})}{\sigma(r_{Q_{mod}}) \cdot \sigma(r_{WL_{obs}})}$ | $r_{Q,mod}$: ranks of the modelled discharge  $r_{WL,obs}$: ranks of the observed water levels | (2) |
| Relative error | $E_{R,\theta} = 1 - \dfrac{\lvert \theta_{mod} - \theta_{obs} \rvert}{\theta_{obs}}$ | $\theta$: variable | (3) |
| Euclidian distance over multiple virtual stations | $D_{E,\beta,\gamma} = 1 - \sqrt{\left( \sum_i w_i \cdot \left( 1 - E_{\beta,\gamma} \right)^2 \right)}$ | $w_i$: relative weight of virtual station $i$  $\beta$: model performance metric  $\gamma$: parameter selection method | (4) |
| Euclidian distance over multiple signatures | $D_E = 1 - \sqrt{\dfrac{1}{(N+M)} \left( \sum_n \left( 1 - E_{NS,\theta_n} \right)^2 + \sum_m \left( 1 - E_{R,\theta_m} \right)^2 \right)}$ | $\theta$: signature  $n$: signatures evaluated with Eq.1 with maximum $N$  $m$: signatures evaluated with Eq.3 with maximum $M$ | (5) |

**3.3.2 Parameter selection based on the seasonal water storage (GRACE)**

In a next step we assumed that discharge records in the Luangwa Basin were absent. The starting assumption
315 thus had to be that all model realizations, i.e. all sampled parameter sets, were equally likely to allow feasible representations of the hydrological system. In a stepwise approach, confronting these realizations with different types of data, we sequentially identified and discarded solutions that were least likely to provide meaningful system representations, thereby gradually narrowing down the feasible parameter space.

We first identified and discarded solutions that were least likely to preserve observed  seasonal water storage
320 ($S_{tot}$) fluctuations. To do so, the monthly modelled total water storage ($S_{tot,mod} = S_i + S_u + S_f + S_s$) relative to the 2004-2009 time-mean baseline in each grid cell was compared to water storage anomalies observed with GRACE where this same time-mean baseline was used (Tang et al., 2017; Fang et al., 2016; Forootan et al., 2019; Khaki and Awange, 2019).

The model's skill to reproduce the seasonal water storage, i.e. $S_{tot}$, was assessed using the Nash-Sutcliffe
325 efficiency $E_{NS,Stot}$ (Eq.1). Note that $E_{NS,Stot,j}$ was computed at first from the time series of $S_{tot}$ in each grid cell $j$ which were then averaged to obtain $E_{NS,Stot}$. If no additional data were available, a hypothetic modeller relying on $E_{NS,Stot}$ to calibrate a model, may choose only the solution with the highest $E_{NS,Stot}$ or allow for some uncertainty. To mimic this traditional approach and  balance it with a sufficient number of feasible solutions to be kept for the subsequent steps, we here identified and discarded the poorest performing 75% of all solutions in terms of
330 $E_{NS,Stot}$ as unfeasible for the subsequent modelling steps.

**3.3.3 Parameter selection based on satellite altimetry data**

Next, the remaining feasible parameter sets were used to evaluate their potential to reproduce time series of observed altimetry applying three distinct parameter selection  strategies. Assuming again the situation of an ungauged basin (i.e. no time-series of river flow available), we kept for each strategy as

335 feasible the respective 1% best performing parameter sets according to the specific performance metric associated to that strategy.

**Altimetry Strategy 1: Direct comparison of altimetry data to modelled discharge**

In the simplest approach, we directly used altimetry data to correlate observed water levels with modelled

340 discharge based on the Spearman rank correlation coefficient ($E_{R,WL}$; Spearman, 1904) using Eq.2 (Table 4). This strategy, hereafter referred to with subscript WL, i.e. water level, requires the assumption that the relationship between water level and discharge is monotonic. The Spearman rank correlation was applied successfully in previous studies to calibrate a rainfall-runoff model to water level time series (Seibert and Vis, 2016). As there were multiple virtual stations with water level data available in this study, the $E_{R,WL}$ was

345 computed at each location simultaneously. The individual values $E_{R,WL}$ were weighted based on the record length of the corresponding virtual stations and then combined into the Euclidean distance as aggregate metric $D_{E,R,WL}$ with Eq.4.

**Table 5: Overview of the parameter identification strategies applied in this study**

| Strategy | Calibration data | Objective function | Parameter group | Calibration parameters | Comments | Q – h conversion | Benefits (+) & limitations (-) |
|---|---|---|---|---|---|---|---|
| **Discharge (reference)** | Discharge (at basin outlet) | $E_{NS,Q}$ (Eq.1) | Entire basin
Plateau & Terrace
Hillslope
Wetland
River profile | $K_s$, $C_e$
$I_{max}$, $S_{umax}$, $K_f$, W
$I_{max}$, $S_{umax}$, $K_f$, W, $\beta$, $T_{lag}$
$I_{max}$, $S_{umax}$, $K_f$, W, $C_{max}$
v
Total: 18 | Traditional model calibration on observed flow data
Combination of 8 different flow signatures | - | - |
| **Seasonal water storage** | GRACE | $E_{NS,Stot}$ (Eq.1) | Entire basin
Plateau & Terrace
Hillslope
Wetland
River profile | $K_s$, $C_e$
$I_{max}$, $S_{umax}$, $K_f$, W
$I_{max}$, $S_{umax}$, $K_f$, W, $\beta$, $T_{lag}$
$I_{max}$, $S_{umax}$, $K_f$, W, $C_{max}$
v
Total: 18 | No discharge data used | - | - |
| **Altimetry Strategy 1** | Altimetry (at 18 virtual stations) & GRACE | Altimetry: $D_{E,R,WL}$ (Eq.2,4)
GRACE: $E_{NS,Stot}$ (Eq.1) | Entire basin
Plateau & Terrace
Hillslope
Wetland
River profile | $K_s$, $C_e$
$I_{max}$, $S_{umax}$, $K_f$, W
$I_{max}$, $S_{umax}$, $K_f$, W, $\beta$, $T_{lag}$
$I_{max}$, $S_{umax}$, $K_f$, W, $C_{max}$
v
Total: 18 | No discharge data used
Combination of 18 virtual stations
Combined with GRACE | - | + No extra parameters or data needed
+ Assumption: monotonic relation between discharge and river water level
- Focus on dynamics only, not volume |
| **Altimetry Strategy 2** | Altimetry (at 18 virtual stations) & GRACE | Altimetry: $D_{E,NS,RC}$ (Eq.1,4)
GRACE: $E_{NS,Stot}$ (Eq.1) | Entire basin
Plateau & Terrace
Hillslope
Wetland
River profile | $K_s$, $C_e$
$I_{max}$, $S_{umax}$, $K_f$, W
$I_{max}$, $S_{umax}$, $K_f$, W, $\beta$, $T_{lag}$
$I_{max}$, $S_{umax}$, $K_f$, W, $C_{max}$
v, $a_1$, $a_2$, $a_3$, $a_4$, $b_1$, $b_2$, $b_3$, $b_4$
Total: 26 | No discharge data used
Combination of 18 virtual stations
Combined with GRACE | Calibrated Rating curve | + No extra data needed
- Two extra parameters per cross-section |
| **Altimetry Strategy 3** | Altimetry (at 18 virtual stations) & GRACE | Altimetry: $D_{E,NS,SM}$ (Eq.1,4)
GRACE: $E_{NS,Stot}$ (Eq.1) | Entire basin
Plateau & Terrace
Hillslope
Wetland
River profile | $K_s$, $C_e$
$I_{max}$, $S_{umax}$, $K_f$, W
$I_{max}$, $S_{umax}$, $K_f$, W, $\beta$, $T_{lag}$
$I_{max}$, $S_{umax}$, $K_f$, W, $C_{max}$
v, k
Total: 18 | No discharge data used
Combination of 18 virtual stations
Combined with GRACE | Strickler-Manning | + Only 1 extra parameter
- Cross-section data needed
- Assumption: constant roughness in space and time |
| **Water level Strategy 1** | Water level (at basin outlet) & GRACE | Altimetry: $E_{NS,SM,GE}$ (Eq.1)
GRACE: $E_{NS,Stot}$ (Eq.1) | Entire basin
Plateau & Terrace
Hillslope
Wetland
River profile | $K_s$, $C_e$
$I_{max}$, $S_{umax}$, $K_f$, W
$I_{max}$, $S_{umax}$, $K_f$, W, $\beta$, $T_{lag}$
$I_{max}$, $S_{umax}$, $K_f$, W, $C_{max}$
v, k
Total: 19 | No discharge data used
Combined with GRACE | Strickler-Manning | + Only 1 extra parameter
- Cross-section data needed
- Assumption: constant roughness in space and time |
| **Water level Strategy 2** | Water level (at basin outlet) & GRACE | Altimetry: $E_{NS,SM,ADCP}$ (Eq.1)
GRACE: $E_{NS,Stot}$ (Eq.1) | Entire basin
Plateau & Terrace
Hillslope
Wetland
River profile | $K_s$, $C_e$
$I_{max}$, $S_{umax}$, $K_f$, W

[revised manuscript text omitted]